# Correlated Noise Provably Beats Independent Noise for Differentially Private Learning

**Christopher A. Choquette-Choo**[*]   **Krishnamurthy (Dj) Dvijotham**[*]   **Krishna Pillutla**[*]
**Arun Ganesh**   **Thomas Steinke**   **Abhradeep Guha Thakurta**
Google

## Abstract

Differentially private (DP) learning algorithms inject noise into the learning process. While the most common private learning algorithm, DP-SGD, adds independent Gaussian noise in each iteration, recent work on matrix factorization mechanisms has shown empirically that introducing correlations in the noise can greatly improve their utility. We characterize the asymptotic learning utility for any choice of the correlation function, giving precise analytical bounds for linear regression and as the solution to a convex program for general convex functions. We show, using these bounds, how correlated noise provably improves upon vanilla DP-SGD as a function of problem parameters such as the effective dimension and condition number. Moreover, our analytical expression for the near-optimal correlation function circumvents the cubic complexity of the semi-definite program used to optimize the noise correlation matrix in previous work. We validate our theory with experiments on private deep learning. Our work matches or outperforms prior work while being efficient both in terms of compute and memory.

## 1  Introduction

The broad adoption of deep learning using sensitive data has led to the increasing popularity of rigorous frameworks for privacy preservation, such as differential privacy (Dwork et al., 2006). The workhorse of private learning, a differentially private variant of stochastic gradient descent called DP-SGD (Song et al., 2013; Bassily et al., 2014; Abadi et al., 2016), clips per-example gradients to some $\ell_2$ norm and adds *independent* Gaussian noise. DP-SGD has been used in a range of applications from learning with medical images (Adnan et al., 2022) to finetuning large language models with $O(100B)$ parameters (He et al., 2023).

A recent line of work instead proposes to add *correlated* Gaussian noise to each clipped gradient (Smith & Thakurta, 2013; Kairouz et al., 2021a; Denisov et al., 2022; Choquette-Choo et al., 2023b). This class of algorithms called DP-FTRL, has been used for private federated learning at industrial scale (Xu et al., 2023). By solving an expensive semi-definite program to find the noise correlations, Choquette-Choo et al. (2023a) demonstrated *empirically* that DP-FTRL is never worse and often *much better* than DP-SGD in its privacy-utility tradeoff across multiple modalities like images and text.

However, several questions remain open. Does DP-FTRL **provably improve** over DP-SGD in its expected utility? Further, can we design a more **computationally efficient** procedure to find the noise correlations for DP-FTRL without significantly worsening the privacy-utility tradeoff?

We answer both questions affirmatively by (1) providing a sharp theoretical characterization of the noisy training dynamics of DP-FTRL, and (2) leveraging these analytical tools to circumvent the semi-definite program required in past work.

---

[*]Equal contribution; alphabetical ordering.

---

**Algorithm 1** The DP-FTRL/Noisy-FTRL algorithms with a noise correlation matrix $\boldsymbol{B} \in \mathbb{R}^{T \times T}$

---

**Input:** $\boldsymbol{B} \in \mathbb{R}^{T \times T}$, initial iterate $\boldsymbol{\theta}_0 \in \mathbb{R}^d$, $\ell_2$ clip norm $G$, noise multiplier $\sigma_{\mathsf{dp}}$, learning rate $\eta$, dataset $\mathcal{D}$

1: **for** $t = 0, \ldots, T-1$ **do**

2:      Obtain the next datapoint $\boldsymbol{z}_t$ and compute $\boldsymbol{g}_t = \begin{cases} \nabla f(\boldsymbol{\theta}_t; \boldsymbol{z}_t) + \nabla r(\boldsymbol{\theta}) & \text{for Noisy-FTRL,} \\ \mathsf{clip}\left(\nabla f(\boldsymbol{\theta}_t; \boldsymbol{z}_t), G\right) + \nabla r(\boldsymbol{\theta}) & \text{for DP-FTRL} \end{cases}$

3:      Sample noise $\boldsymbol{w}_t \sim \mathcal{N}(0, \sigma_{\mathsf{dp}}^2 G^2 \boldsymbol{I}_d)$ and calculate the correlated noise $\widetilde{\boldsymbol{w}}_t = \sum_{\tau=0}^{t} \boldsymbol{B}_{t,\tau} \boldsymbol{w}_\tau$

4:      Update $\boldsymbol{\theta}_{t+1} = \boldsymbol{\theta}_t - \eta \widetilde{\boldsymbol{g}}_t$ for the noisy gradient $\widetilde{\boldsymbol{g}}_t = \boldsymbol{g}_t + \widetilde{\boldsymbol{w}}_t$

**Return** $\boldsymbol{\theta}_T$

---

## 1.1 PROBLEM SETUP AND BACKGROUND

Let $\mathcal{D} = \{\boldsymbol{z}_0, \ldots, \boldsymbol{z}_{T-1}\}$ be a dataset of $T$ datapoints, where each datapoint is sampled i.i.d. from an underlying distribution $\mathbb{P}_{\mathsf{data}}$. Our learning objective is to minimize:

$$F(\boldsymbol{\theta}) = \mathbb{E}_{\boldsymbol{z} \sim \mathbb{P}_{\mathsf{data}}}\left[f(\boldsymbol{\theta}; \boldsymbol{z})\right] + r(\boldsymbol{\theta}) \ , \tag{1}$$

where $f(\boldsymbol{\theta}; \boldsymbol{z})$ is the loss incurred by model parameters $\boldsymbol{\theta} \in \mathbb{R}^d$ on a datapoint $\boldsymbol{z}$, and $r(\cdot)$ is data-independent regularization. We aim to minimize $F$ while satisfying differential privacy with respect to the dataset $\mathcal{D}$. We assume that $F$ has a unique minimizer denoted $\boldsymbol{\theta}_\star$.

We focus on variants of stochastic gradient descent with a batch size of $1$ for data arriving in a stream. The learning algorithms we study are presented in Algorithm 1; we assume throughout that the dataset $\mathcal{D}$ is randomly shuffled before running the algorithm so that each datapoint $\boldsymbol{z}_t$ is an i.i.d. sample from $\mathbb{P}_{\mathsf{data}}$. DP-FTRL with a noise coefficient matrix $\boldsymbol{B} \in \mathbb{R}^{T \times T}$ (which is lower triangular) performs the updates[1]

$$\boldsymbol{\theta}_{t+1} = \boldsymbol{\theta}_t - \eta \left( \mathsf{clip}\left(\nabla f(\boldsymbol{\theta}_t; \boldsymbol{z}_t), G\right) + \nabla r(\boldsymbol{\theta}_t) + \sum_{\tau=0}^{t} \boldsymbol{B}_{t,\tau} \boldsymbol{w}_\tau \right) \tag{2}$$

for Gaussian noise $\boldsymbol{w}_t \sim \mathcal{N}(\boldsymbol{0}, \sigma_{\mathsf{dp}}^2 G^2 \boldsymbol{I}_d)$, where $\mathsf{clip}(\cdot, G)$ denotes projection onto an $\ell_2$ ball of radius $G$. We define Noisy-FTRL to be DP-FTRL without clipping. Taking $\boldsymbol{B} = \boldsymbol{I}$ as the identity matrix recovers DP-SGD (with clipping) and Noisy-SGD (without clipping), and other choices give rise to alternate algorithms.

We restate a result from prior work showing that DP-FTRL is differentially private for any choice of $\boldsymbol{B}$, provided the noise multiplier is scaled up appropriately.

**Theorem 1.1** (Denisov et al. (2022); Bun & Steinke (2016)). *DP-FTRL (Algorithm 1 with the clipping enabled) satisfies $\rho$-zero concentrated differential privacy (zCDP) if the noise multiplier is taken as $\sigma_{\mathsf{dp}}^2 = \gamma_T^2(\boldsymbol{B})/(2\rho)$ where $\gamma_T(\boldsymbol{B}) = \max_{t<T} \|(\boldsymbol{B}^{-1})_{:,t}\|_2$ is the sensitivity of $\boldsymbol{B}^{-1}$.*[2]

**Remark 1.2.** *Although Noisy-FTRL is* not *differentially private, it lets us analyze the noise dynamics of DP-FTRL without technicalities associated with clipping. We sharply characterize the asymptotic utility of Noisy-FTRL for linear regression and show later that this analysis extends to DP-FTRL under appropriate assumptions. For mean estimation and learning with Lipschitz convex losses, we directly analyze DP-FTRL.*

## 1.2 MOTIVATION

This work is motivated by two open questions in particular.

---

[1] Matrices (e.g. $\boldsymbol{B} = [\boldsymbol{B}_{t,\tau}]_{t,\tau \geq 0}$) and vectors (e.g. $\boldsymbol{\beta} = (\beta_0, \beta_1, \ldots)$) are zero-indexed and bold-faced.

[2] We give DP guarantees w.r.t. the "zero-out" notion of neighborhood (Kairouz et al., 2021a); see Appendix A for a review. Further, a $\rho$-zCDP guarantee can be readily translated into $(\varepsilon, \delta)$-DP (Bun & Steinke, 2016, Prop. 1.3).

**Provable separation between DP-SGD and DP-FTRL:** The best-known separation between DP-SGD and DP-FTRL in the literature is due to Kairouz et al. (2021a). For $G$-Lipschitz convex losses, DP-FTRL at a privacy level of $\rho$-zCDP achieves a suboptimality of $O(Gd^{1/4}/\sqrt{\rho T})$ compared to DP-SGD's $O(Gd^{1/4}/\sqrt{\rho^2 T})$. The only improvement here is in terms of the privacy parameter $\rho$. More recently, Koloskova et al. (2023) analyze Noisy-FTRL but *without* normalizing for the sensitivity $\gamma_T(\boldsymbol{B})$ as in Theorem 1.1. Thus, the existing theory fails to reflect the large margin by which DP-FTRL empirically outperforms DP-SGD across the board (Choquette-Choo et al., 2023a), and a precise characterization is missing.

**Computationally efficient DP-FTRL:** Prior work on DP-FTRL utilizes the noise correlation matrix $\boldsymbol{B}$ that minimizes the squared error in the *gradient prefix sums* (Kairouz et al., 2021a; Denisov et al., 2022):

$$\varphi(\boldsymbol{B}) = \sum_{t=0}^{T-1} \mathbb{E}\left\|\sum_{\tau=0}^{t} \widetilde{\boldsymbol{g}}_t - \sum_{\tau=0}^{t} \boldsymbol{g}_t\right\|_2^2 \tag{3}$$

where $\boldsymbol{g}_t$ is the clipped gradient applied in iteration $t$ and $\widetilde{\boldsymbol{g}}_t$ is its noisy counterpart (cf. Algorithm 1). This was, in turn, obtained as an upper bound on the regret in an adversarial online learning setting (Kairouz et al., 2021a, Thm. C.1). The most potent algorithm from the previous work gave $\boldsymbol{B}$ as the solution of a semidefinite program with matrix variables of size $O(T^2)$, requiring $O(T^3)$ time (Denisov et al., 2022, Eq. 4). This cost is prohibitive for large learning problems. Moreover, there is a mismatch between the objective (3) used to find the noise correlations and the final learning objective $F(\boldsymbol{\theta}_T)$. In particular, there exist matrices $\boldsymbol{B}_1, \boldsymbol{B}_2$ with equal squared error $\varphi(\boldsymbol{B}_1) = \varphi(\boldsymbol{B}_2)$ and equal sensitivities $\gamma_T(\boldsymbol{B}_1) = \gamma_T(\boldsymbol{B}_2)$ such that DP-FTRL with $\boldsymbol{B}_1$ diverges while DP-FTRL with $\boldsymbol{B}_2$ converges (Koloskova et al., 2023).

**Our approach:** We study the suboptimality in the final objective $\mathbb{E}[F(\boldsymbol{\theta}_T) - F(\boldsymbol{\theta}_\star)]$. We work in the asymptotic $T \to \infty$ regime to allow the use of analytic tools, but also to derive results that apply regardless of the dataset size.[3] Second, we restrict the search over $\boldsymbol{B}$ to *Toeplitz* matrices $\boldsymbol{B}_{t,\tau} = \beta_{t-\tau}$ generated by a sequence $\boldsymbol{\beta} = (\beta_0, \beta_1, \ldots)$ of reals, but a stronger motivation is that they are **anytime**, i.e., they do not be recomputed for each value of $T$ and easily apply as $T \to \infty$. Toeplitz $\boldsymbol{B}$ were previously considered for their computational efficiency in learning (Choquette-Choo et al., 2023b) and their near-optimal rates in linear counting queries (Henzinger et al., 2024). Thus, our goal is to characterize the **asymptotic suboptimality**

$$F_\infty(\boldsymbol{\beta}) := \lim_{T \to \infty} \mathbb{E}\left[F(\boldsymbol{\theta}_T) - F(\boldsymbol{\theta}_\star)\right] \tag{4}$$

for $\boldsymbol{\theta}_T$ produced by Noisy-FTRL or DP-FTRL under noise correlation weights $\boldsymbol{\beta}$ where $\boldsymbol{\theta}_\star = \arg\min F$ is assumed unique. This limit turns out to be well-defined and finite for the settings we consider as long as $\|\boldsymbol{\beta}\|_2$ is finite. We analyze $F_\infty$ in the frequency domain using the **discrete-time Fourier transform** $B(\omega) = \sum_{t=0}^{\infty} \beta_t \exp(i\omega t)$, with $i$ the imaginary unit. Further, we define the limiting sensitivity associated with $B$ as the limiting value of $\gamma_T$, which, using standard Fourier analysis tools, equals

$$\gamma_\infty(B) := \lim_{T \to \infty} \gamma_T(B) = \left(\frac{1}{2\pi} \int_{-\pi}^{\pi} |B(\omega)|^{-2} \, d\omega\right)^{1/2}. \tag{5}$$

### 1.3 OUR CONTRIBUTIONS

The concrete contributions of this work are as follows.

**$\nu$-DP-FTRL: Analytically optimal DP-FTRL for mean estimation:** We give analytical expressions for the asymptotic suboptimality $F_\infty$ for mean estimation and the noise correlations $\boldsymbol{\beta}$ that minimize $F_\infty$ as a function of the learning rate $\eta$ (§2.1). We find that the optimal noise is *anti*-correlated, so the algorithm *subtracts out* previously added noise. Inspired by the analytical expression for the optimal noise correlations $\boldsymbol{\beta}_\star$ for mean estimation, we propose a single-parameter family of choices for $\boldsymbol{\beta}$, which we call $\nu$-DP-FTRL. We show its favorable theoretical and empirical properties for a broader range of problems.

---

[3]Note that the DP noise multiplier $\sigma_{\mathsf{dp}}$ remains finite in the asymptotic $T \to \infty$ regime as we consider the streaming setting: each example is processed once and the number of examples also grows to infinity.

Table 1: Asymptotic suboptimality of Noisy-SGD/Noisy-FTRL for linear regression with Gaussian inputs $\boldsymbol{x} \sim \mathcal{N}(\boldsymbol{0}, \boldsymbol{H})$ and noise multiplier $\sigma_{\text{dp}}^2 = \gamma_\infty(\boldsymbol{\beta})^2/(2\rho)$ based on the limiting sensitivity (5). We give the bounds in terms of the learning rate $\eta$, dimension $d$, the effective dimension $d_{\text{eff}} = \frac{\text{Tr}[\boldsymbol{H}]}{\lambda_{\max}(\boldsymbol{H})}$ and the noise variance $\rho^{-1}$ representing the privacy level. We take $G = 1$ and $\lambda_{\max}(\boldsymbol{H}) = 1$ w.l.o.g. and only show the term depending on $\rho$. Since $1 \leq d_{\text{eff}} \leq d$, Noisy-FTRL is better than Noisy-SGD at smaller $\eta$ or when $d_{\text{eff}}$ is small (e.g., when $\boldsymbol{H}$ is close to low rank).

| Algorithm | Asymptotic Suboptimality $F_\infty$ | Ratio w/ Lower Bound | Remark |
|---|---|---|---|
| Lower Bound | $\Omega\left(\eta^2 \rho^{-1} d_{\text{eff}}\right)$ | $1$ | for all $\boldsymbol{\beta}$ with finite $\|\boldsymbol{\beta}\|_1$ |
| Noisy-SGD | $\Theta\left(\eta\rho^{-1}d\right)$ | $\frac{d}{\eta d_{\text{eff}}}$ | $\Theta(\cdot)$ denotes matching upper & lower bounds |
| $\nu$-Noisy-FTRL | $O\left(\eta^2 \rho^{-1} d_{\text{eff}} \log^2 \frac{1}{\eta\mu}\right)$ | $\log^2 \frac{1}{\eta\mu}$ | Here, $\mu = \lambda_{\min}(\boldsymbol{H})$ and we use weights $\boldsymbol{\beta}$ from (7) |

**Strict separation for linear regression:** We establish sharp bounds on Noisy-FTRL (i.e., DP-FTRL without gradient clipping) for linear regression. Summarized in Table 1 and stated formally in §2.2, we show:

(a) $\nu$-Noisy-FTRL, with analytical closed-form correlations, matches the lower bound up to log factors. Both of these bounds scale with the effective dimension $d_{\text{eff}}$ of the problem, which is no greater than the dimension $d$ but can be much smaller when the data is approximately low rank.

(b) $\nu$-Noisy-FTRL is provably better than Noisy-SGD by a factor that can be as large as $d/\log d$ (when $d_{\text{eff}}$ is a constant). This shows an exponential separation between Noisy-FTRL and Noisy-SGD.

Our bounds quantitatively show how the anti-correlations of $\nu$-Noisy-FTRL help prevent noise accumulation along eigen-directions of the Hessian with small eigenvalues. The gradients have a weak signal along these directions and are unable to undo the effect of the previous noise and move the iterates back towards the minimizer; the anti-correlations are essential to obtain near-optimal asymptotic suboptimality. We also leverage these asymptotics to give bounds on the utility of $\nu$-DP-FTRL and DP-SGD for finite $T$.

**Numerical separation for general strongly convex functions:** We bound the asymptotic suboptimality $F_\infty$ for any noise correlation weights $\boldsymbol{\beta}$ as the optimal value of a convex program. We use this to show that DP-FTRL achieves a tighter bound particularly when the condition number is large (Figure 3 in §3).

**Experiments with private deep learning:** We show the proposed $\nu$-DP-FTRL outperforms other efficient differentially private algorithms on image and text classification tasks. We also find that our approach is competitive even with inefficient approaches that require $O(T^3)$ computation and $O(T^2)$ memory.

## 2 ANALYSIS FOR QUADRATIC OBJECTIVES

For quadratic objectives, Algorithm 1 (with no clipping) corresponds to a linear dynamical system (Gray & Davisson, 2004), allowing us to use analytical tools. We give an exact analysis of DP-FTRL for mean estimation and Noisy-FTRL for linear regression. The analysis of Noisy-FTRL also lets us derive guarantees for DP-FTRL for linear regression. We do not aim to achieve the best possible rates in these stylized models. Rather, our goal is to understand the noise dynamics of DP-FTRL and show a separation with DP-SGD.

### 2.1 CONCEPTUAL OVERVIEW: PRIVATE MEAN ESTIMATION IN ONE DIMENSION

We begin with the simplest objective function, the squared error for a mean estimation problem on the real line. This setting captures the core intuition and ideas used to derive further results.

Consider a distribution $\mathbb{P}_{\text{data}}$ with $|z - \mathbb{E}[z]| \leq \sigma_{\text{sgd}}$ and $|z| \leq 1$ a.s. for $z \sim \mathbb{P}_{\text{data}}$. Our objective now is

$$F(\theta) = \frac{1}{2}\mathbb{E}_{z\sim\mathbb{P}_{\text{data}}}(\theta - z)^2 \quad \text{with} \quad f(\theta; z) = \frac{z^2}{2} - z\theta, \quad \text{and} \quad r(\theta) = \frac{\theta^2}{2}. \tag{6}$$

We show a strict separation between DP-FTRL and DP-SGD for this simple minimization problem.

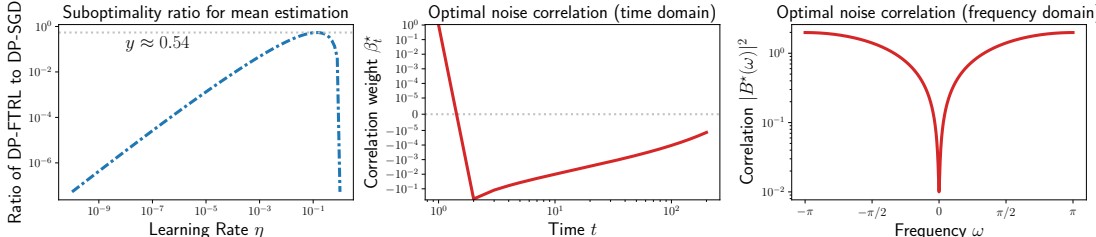

Figure 1: **Left**: The ratio of the asymptotic suboptimalities of DP-FTRL to DP-SGD for mean estimation vs. the learning rate $\eta$. DP-FTRL is never worse but is orders of magnitude better at $\eta \to 0$ or $\eta \to 1$. **Middle & Right**: Time- and frequency-domain descriptions of the optimal noise correlations for mean estimation (defined in Theorem 2.1).

**Theorem 2.1.** *Consider the setting above with learning rate $\eta \le 1$ and clip norm $G = 1$ and $\sigma_{\mathsf{dp}}^2 = \frac{\gamma_\infty(B)^2}{2\rho}$. Then, the asymptotic suboptimality of a $\rho$-zCDP sequence $(\theta_t)_{t=0}^\infty$ obtained via DP-SGD is $F_\infty(\boldsymbol{\beta}_{\mathsf{dpsgd}}) = \Theta(\eta\rho^{-1} + \eta\sigma_{\mathsf{sgd}}^2)$. Further, the asymptotic suboptimality of any $\rho$-zCDP sequence $(\theta_t)_{t=0}^\infty$ from DP-FTRL is*

$$\inf_{\boldsymbol{\beta}} F_\infty(\boldsymbol{\beta}) = F_\infty(\boldsymbol{\beta}^\star) = \Theta\left(\eta^2\rho^{-1}\log^2(1/\eta) + \eta\sigma_{\mathsf{sgd}}^2\right) \ .$$

*The infimum above is attained by $\beta_t^\star = (-1)^t \binom{1/2}{t}(1-\eta)^t$, where $\binom{1/2}{t} = \prod_{k=0}^{t-1}\frac{1/2-k}{t-k}$.*

*Proof Sketch.* Using tools from frequency-domain analysis of linear time-invariant systems (Oppenheim et al., 1997), we show that the asymptotic variance is an integral of $|B(\omega)|^2$. The sensitivity (5) is an integral of $|B(\omega)|^{-2}$ so that $F_\infty$ is a product of these integrals. Its minimizer $B^\star$ can be analytically computed in the Fourier domain (Fig. 1, right), which yields the expression for $\boldsymbol{\beta}^\star$ (Fig. 1, center). See §B for details. □

The optimal $\rho^{-1}$ coefficient $\eta^2\log^2(1/\eta)$ is better than DP-SGD's $\eta$. Note that $\beta_t^\star < 0$ for $t \ge 1$: the noise is *anti*-correlated and it helps by *subtracting out* the previously added noise. We also recover the correlations of (Fichtenberger et al., 2023) as $\eta \to 0$; these were shown to be near-optimal for linear counting queries.

$\nu$**-DP-FTRL/$\nu$-Noisy-FTRL:** Theorem 2.1 gives an analytical expression for the optimal noise correlation weights for DP-FTRL for this simplified setting. We parameterize it with a parameter $0 < \nu < 1$ to define

$$\hat{\beta}_t^\nu := (-1)^t\binom{1/2}{t}(1-\nu)^t \ . \tag{7}$$

We analyze this choice theoretically for the setting of Noisy-FTRL and demonstrate near-optimality for appropriate $\nu$. Later, for our experiments with DP-FTRL, we tune $\nu$ as a hyperparameter to tune. We call this approach (with clipping) $\nu$-DP-FTRL and (without clipping) $\nu$-Noisy-FTRL.

### 2.2 Asymptotic Suboptimality for Linear Regression

We now give a precise analysis of $F_\infty$ for linear regression with $\nu$-Noisy-FTRL. We will use this to derive non-asymptotic privacy-utility bounds for DP-FTRL at the end of this section.

We consider (unregularized) linear regression with loss function $f(\boldsymbol{\theta}; (\boldsymbol{x}, y)) = \frac{1}{2}(y - \langle\boldsymbol{\theta}, \boldsymbol{x}\rangle)^2$ so that

$$F(\boldsymbol{\theta}) = \frac{1}{2}\,\mathbb{E}_{(\boldsymbol{x}, y)\sim\mathbb{P}_{\mathsf{data}}}(y - \langle\boldsymbol{\theta}, \boldsymbol{x}\rangle)^2 \ . \tag{8}$$

We assume $d$-dimensional Gaussian covariates $\boldsymbol{x} \sim \mathcal{N}(\boldsymbol{0}, \boldsymbol{H})$ and independent Gaussian residuals $y - \langle\boldsymbol{\theta}_\star, \boldsymbol{x}\rangle \sim \mathcal{N}(0, \sigma_{\mathsf{sgd}}^2)$ where $\boldsymbol{\theta}_\star = \arg\min F$. We make these assumptions for ease of presentation; we state

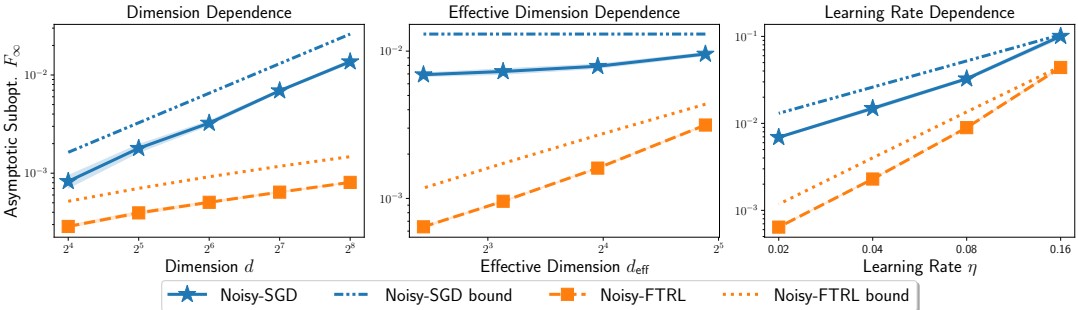

Figure 2: **Linear regression simulations**: We plot the empirically observed asymptotic suboptimality of $\nu$-Noisy-FTRL/Noisy-SGD and their theoretical bounds with $d = 128$ (varied in the left plot) where the Hessian $\boldsymbol{H}$ has eigenvalues $\lambda_k = 1/k$ (varied as $k^{-\alpha}$ for $\alpha \in [0.4, 1]$ in the middle plot), and learning rate $\eta = 0.02$ (varied in the right plot). The **slope** of the corresponding empirical and theoretical lines are nearly equal, showing the **tightness of the theory**. In particular, we observe that Noisy-SGD has a linear dependence on the dimension (slope 1.00) and is nearly constant w.r.t. the effective dimension (slope 0.18) while Noisy-FTRL has a near-linear dependence on the effective dimension (slope 0.94). Noisy-FTRL (slope 2.03) also has a better dependence on the learning rate than Noisy-SGD (slope 1.27).

and prove our results under weaker assumptions in the supplement. Further, we assume that $F$ is $L$-smooth and $\mu$-strongly convex (equivalently, $\mu \boldsymbol{I} \preceq \boldsymbol{H} \preceq L\boldsymbol{I}$ since the input covariance $\boldsymbol{H}$ is also the Hessian of $F$).

We express the bounds on $F_\infty$ in terms of the correlation weights $\boldsymbol{\beta}$ and the problem parameters $\rho, G$ which, for DP-FTRL, denote the target privacy level and the gradient clip norm respectively. See §C for proofs.

**Theorem 2.2.** *Let $c, C_1, C_2$ denote universal constants. For $\eta \leq c/\mathsf{Tr}\,[\boldsymbol{H}]$, we have*

$$\textit{(Noisy-SGD)} \qquad F_\infty(\boldsymbol{\beta}^{\mathsf{sgd}}) = \Theta\left(\eta d G^2 \rho^{-1} + \eta \sigma_{\mathsf{sgd}}^2 \mathsf{Tr}\,[\boldsymbol{H}]\right) \quad \textit{with } \boldsymbol{\beta}^{\mathsf{sgd}} = (1, 0, \ldots),$$

$$\textit{($\nu$-Noisy-FTRL)} \qquad F_\infty(\hat{\boldsymbol{\beta}}^\nu) \leq C_1 \left(\eta^2 G^2 \rho^{-1} \log^2 \tfrac{1}{\nu} + \eta \sigma_{\mathsf{sgd}}^2\right) \mathsf{Tr}\,[\boldsymbol{H}] \quad \textit{with } \nu \leq \eta\mu, \textit{ and}$$

$$\textit{(Lower bound)} \qquad F_\infty(\boldsymbol{\beta}) \geq C_2 \left(\eta^2 G^2 \rho^{-1} + \eta \sigma_{\mathsf{sgd}}^2\right) \mathsf{Tr}\,[\boldsymbol{H}] \quad \textit{for all } \boldsymbol{\beta} \textit{ with } \|\boldsymbol{\beta}\|_1 < \infty.$$

*This shows the near-optimality of $\nu$-Noisy-FTRL and a provable gap between Noisy-FTRL and Noisy-SGD.*

Observe that our bounds separate the contributions arising from correlated noise ($\rho^{-1}$ term) and those from the inherent noise in the linear model ($\sigma_{\mathsf{sgd}}^2$ term). We focus on the effect of correlation because the effect of the latter noise is the same across all choices of $\boldsymbol{\beta}$. We plot the differences in Figure 2.

**Exponential separation between Noisy-SGD and Noisy-FTRL:** Noisy-SGD's stationary error depends on the ambient dimension $d$, while the lower bound depends on the *effective dimension* $d_{\mathsf{eff}} = \mathsf{Tr}\,[\boldsymbol{H}]/\|\boldsymbol{H}\|_2$ of the covariance $\boldsymbol{H}$. We have, $d_{\mathsf{eff}} \leq d$ with equality when all the eigenvalues of $\boldsymbol{H}$ are equal but $d_{\mathsf{eff}} \ll d$ when the eigenvalues of $\boldsymbol{H}$ decay rapidly or it is nearly low rank. This is true particularly for overparameterized models where the features may be highly correlated resulting in an approximately low-rank covariance. The effective dimension is closely related to the stable rank (Rudelson & Vershynin, 2007); cf. §C.6.

For instance, if the eigenvalues of $\boldsymbol{H}$ are $(1, 1/d, \ldots, 1/d)$, then $d_{\mathsf{eff}} \leq 2$. Then, Noisy-FTRL's error of $O(\eta^2 \rho^{-1} \log^2(d/\eta))$ is exponentially better than Noisy-SGD's $\Theta(\eta \rho^{-1} d)$. The learning rate dependence of Noisy-SGD is also suboptimal, similar to §2.1. This result is also confirmed empirically in Figure 2 (right).

Assuming $\lambda_{\max}(\boldsymbol{H}) = 1$, the $d_{\mathsf{eff}}$-dependence comes from the contribution of eigen-direction $j$ of $\boldsymbol{H}$ to the asymptotic suboptimality improving from $\Theta(1)$ for Noisy-SGD to scale with the corresponding eigenvalue $\lambda_j$ of $\boldsymbol{H}$ for $\nu$-Noisy-FTRL. Thus, the anti-correlated noise particularly helps in the tail eigen-directions of $\boldsymbol{H}$. We discuss this further in Remark C.16 of §C.

Table 2: **Comparison to prior work**: We apply our theory to compute $F_\infty$ for linear regression given choices of $\boldsymbol{B}$ used in prior work. Though certain choices of the noise correlation $\boldsymbol{\beta}$ may be optimal for finite linear counting queries (Fichtenberger et al., 2023), our results show that they have $F_\infty = \infty$ because the sensitivity diverges as $T \to \infty$. $\nu$-Noisy-FTRL effectively introduces an additional damping term $(1-\nu)^t$ in the correlations of (Fichtenberger et al., 2023) to achieve near-optimality for linear regression. Damping similarly helps for anti-PGD (Orvieto et al., 2022), where the resulting error is the geometric mean of the lower bound and the bound of Noisy-SGD from Theorem 2.2.

| Algorithm | Noise Correlation Weights $\boldsymbol{\beta}$ | Sensitivity in $T$ steps $\gamma_T(\boldsymbol{\beta})^2$ | Asymptotic Suboptimality $F_\infty(\boldsymbol{\beta})$ |
|---|---|---|---|
| (Fichtenberger et al., 2023) | Eq. (7) with $\nu = 0$ | $\log T$ | $\infty$ |
| $\nu$-Noisy-FTRL (Ours) | Eq. (7) with $0 < \nu \leq \eta\mu$ | $\log(1/\nu)$ | $\eta^2 G^2 \rho^{-1} \mathsf{Tr}\,[\boldsymbol{H}] \log^2(1/\nu)$ |
| Anti-PGD (Orvieto et al., 2022) | $(1, -1, 0, \ldots)$ | $T$ | $\infty$ |
| Anti-PGD + Damping | $(1, -(1-\nu), 0, \ldots)$ | $1/\nu$ | $\eta^{3/2} G^2 \rho^{-1} \sqrt{d\mathsf{Tr}\,[\boldsymbol{H}]}$ |

## 2.3 Finite-time Privacy-Utility Bounds for Linear Regression

Noisy-FTRL, which we analyzed so far, is not differentially private. Differential privacy requires gradient clipping which significantly complicates the analysis. However, for a finite time horizon $T$, we can argue using concentration that $\nabla f(\boldsymbol{\theta}; \boldsymbol{z})$ is bounded with high probability, and clipping can be avoided. Formal statements and proofs for the finite-time analysis are given in §D.

Consider DP-FTRL with noise correlation $\hat{\boldsymbol{\beta}}^\nu$ from (7) with $\nu = \eta\mu$ and gradients clipped to any $\ell_2$-norm $G$. As mentioned in §1.1, the outputs $(\boldsymbol{\theta}_1, \ldots, \boldsymbol{\theta}_T)$ of DP-FTRL are $\rho$-zCDP. For an appropriate choice of $\eta$, we give utility bounds in terms of the effective dimension $d_{\mathsf{eff}}$ and the condition number $\kappa = L/\mu$:

(a) For $\eta$ small enough, we have with probability at least $1 - p$ that

$$\max_{t<T} \|\boldsymbol{g}_t\|_2 \leq c \max\left\{\mathsf{Tr}\,[\boldsymbol{H}]\,\|\boldsymbol{\theta}_0 - \boldsymbol{\theta}_\star\|_2, \sigma_{\mathsf{sgd}}\sqrt{\mathsf{Tr}\,[\boldsymbol{H}]}\right\} \mathrm{polylog}\,(T/p) =: G\,. \tag{9}$$

Let $\mathcal{E}$ denote this event. If $\mathcal{E}$ holds, no gradients are clipped and DP-FTRL coincides with Noisy-FTRL.

(b) For $T \geq \widetilde{\Omega}(\kappa^2 d_{\mathsf{eff}}^2 d/\rho)$, we have (omitting log factors and $o(1/T^2)$ terms and taking $\|\boldsymbol{H}\|_2 = 1$):

$$\mathbb{E}\left[(F(\boldsymbol{\theta}_t) - F(\boldsymbol{\theta}_\star)) \cdot \mathbb{1}\,(\mathcal{E})\right] \lesssim \begin{cases} \kappa\,d_{\mathsf{eff}}\left(\frac{d d_{\mathsf{eff}}\|\boldsymbol{\theta}_0 - \boldsymbol{\theta}_\star\|_2^2}{\rho T} + \frac{d\sigma_{\mathsf{sgd}}^2}{\rho T} + \frac{\sigma_{\mathsf{sgd}}^2}{T}\right) & \text{for DP-SGD,} \\ \kappa d_{\mathsf{eff}}\left(\frac{\kappa d_{\mathsf{eff}}^2\|\boldsymbol{\theta}_0 - \boldsymbol{\theta}_\star\|_2^2}{\rho T^2} + \frac{\kappa d_{\mathsf{eff}}\sigma_{\mathsf{sgd}}^2}{\rho T^2} + \frac{\sigma_{\mathsf{sgd}}^2}{T}\right) & \text{for } \nu\text{-DP-FTRL.} \end{cases}$$

Thus, the dimension $d$ in DP-SGD's bound effectively becomes $\kappa d_{\mathsf{eff}}/T$ for DP-FTRL, leading to a better dimension dependence. While faster $1/(\rho T^2)$ rates are known for DP-SGD-style algorithms for linear regression (Varshney et al., 2022; Liu et al., 2023), such algorithms require sophisticated adaptive clipping strategies. Our algorithms use a fixed clipping norm $G$ and a fixed noise multiplier $\sigma_{\mathsf{dp}}$ independent of $T$; the bounds presented above are, to the best of our knowledge, the best known in the literature for DP-SGD in this setting. We leave the exploration of combining adaptive clipping with correlated noise for future work.

## 3 Asymptotic Suboptimality for General Strongly Convex Functions

We now generalize §2.2 to general strongly convex problems. Here, we bound the asymptotic suboptimality of DP-FTRL and DP-SGD by the value of a convex program.

**Theorem 3.1.** *Suppose $f(\cdot; \boldsymbol{z})$ is $G$-Lipschitz, and the stochastic gradients are uniformly bounded as $\|\nabla_\theta f(\boldsymbol{\theta}; \boldsymbol{z}) - \mathbb{E}_{\boldsymbol{z}' \sim \mathbb{P}_{\mathsf{data}}}\,[\nabla_\theta f(\boldsymbol{\theta}; \boldsymbol{z}')]\|_2 \leq \sigma_{\mathsf{sgd}}$. Then, if $F$ is $\mu$-strongly convex and $L$-smooth, the asymptotic suboptimality $F_\infty$ is bounded for any noise correlation $B(\omega)$ in the frequency domain by:*

$$\inf\left\{\frac{Ld}{2\pi}\int_{-\pi}^\pi \left(G^2\rho^{-1}|B(\omega)|^2\gamma_\infty(B)^2 + \sigma_{\mathsf{sgd}}^2\right)\psi(\omega)\,\mathrm{d}\omega \,\bigg|\, \psi : [-\pi, \pi] \to \mathbb{R}_+,\ \psi \in \mathcal{C}\,(\eta, L, \mu)\right\}\,, \tag{10}$$

| DP-FTRL Variant | Citation | Corr. matrix $B$ | Anytime? | Computation Cost | |
| --- | --- | --- | --- | --- | --- |
| | | | | Generation | Training (per step) |
| DP-SGD | (Abadi et al., 2016) | Identity | ✓ | $O(1)$ | $O(1)$ |
| Honaker/TreeAgg | (Kairouz et al., 2021a) | Lower-Triangular (LT) | ✓ | $O(1)$ | $O(\log T)$ |
| Optimal CC | (Fichtenberger et al., 2023) | Toeplitz & LT | ✓ | $O(1)$ | $O(T)$ |
| $\nu$-DP-FTRL | Ours | Toeplitz & LT | ✓ | $O(1)$ | $O(T)$ |
| FFT | (Choquette-Choo et al., 2023b) | Toeplitz | - | $O(1)$ | $O\left(T\log^2 T\right)$ |
| Full Honaker | (Honaker, 2015) | Arbitrary | - | $O(T^2)$ | $O(T^2)$ |
| Multi-Epoch (ME) | (Choquette-Choo et al., 2023b) | Arbitrary | - | $O\left(T^3\right)$ | $O\left(T^2\right)$ |

Table 3: **Variants of DP-FTRL**: the noise correlation matrix $B$ and whether the correlation matrix $B$ can be created/optimized agnostic to the time horizon $T$ (denoted as "**Anytime**"), and the computation cost.

*where $\gamma_\infty(B)$ is the limiting sensitivity from Eq. (5), and $\mathcal{C}(\eta, \mu, L)$ is a convex set (details and proof in §E).*

While technically an infinite-dimensional optimization problem over the function $\psi$, we can approximate the solution by discretizing $\psi$ into $k$ points uniformly over $[-\pi, \pi]$. Further, if we discretize $B$ similarly, we can obtain a **second-order cone program** with $k$ conic constraints and $O(k)$ decision variables. As $k \to \infty$, the solution approaches the solution to (10). Empirically, we observe that the values stabilize quickly as $k$ increases. We stop the computation when the change in bound as a function of $k$ drops below a threshold — this gives $k = 1000$.

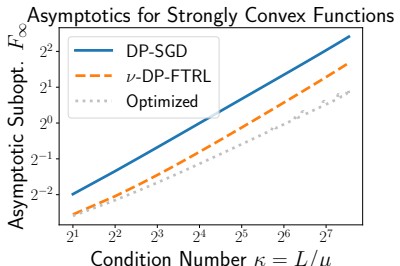

Further, given the optimal $\psi = \psi^\star$, we can run an alternating minimization where we minimize the objective of (10) with respect to $\psi$ for fixed $B$ and with respect to $B$ for fixed $\psi$. This leads to an iteratively improving choice of $B$. We find empirically that this iterative procedure converges quickly and leads to a provable theoretical gap between the upper bounds on $F_\infty$ achievable by DP-SGD and DP-FTRL.

We numerically compare the bound (10) for DP-SGD and $\nu$-DP-FTRL. Figure 3 shows that the gap between DP-SGD and $\nu$-DP-FTRL is multiplicative: the absolute gap grows with the increasing condition number $\kappa = L/\mu$. The suboptimality of "Optimized" DP-FTRL (optimized as described above) grows even more slowly with $\kappa$.

Figure 3: **DP-FTRL attains a tighter bound on** $F_\infty$ with the growing condition number. Here, "Optimized" approximately minimizes (10). The plots hold for smooth and strongly convex functions ($L = 1 = G$, $\sigma_{\text{sgd}} = 0$).

Overall, $\nu$-DP-FTRL significantly improves upon DP-SGD and has only a single tunable parameter $\nu$ and no expensive computation to generate the noise correlations. We focus on $\nu$-DP-FTRL for experiments in this paper but leave the possibility of improving results further based on Optimized DP-FTRL for future work.

## 4 EXPERIMENTS

We demonstrate the practical benefits of $\nu$-DP-FTRL for deep learning tasks. This approach has a single tunable parameter $\nu$ that can easily be tuned based on minimizing the squared error (3) as in prior work.

**Comparing Computation (Table 3):** While optimized matrices (e.g. "ME" in Table 3) have the state-of-the-art privacy-utility tradeoffs in private learning (without amplification), their computational cost scales as

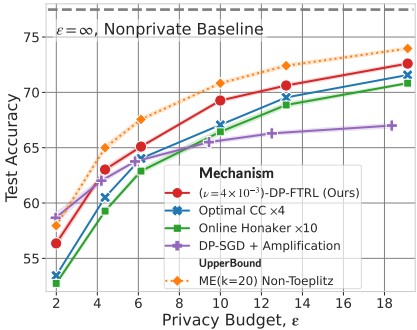 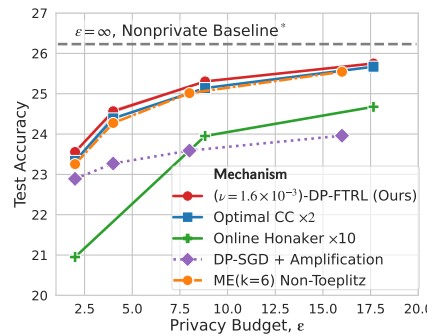

(a) Example-level DP on CIFAR-10 (image classification).  (b) User-level DP on StackOverflow (language modeling).

Figure 4: **The proposed $\nu$-DP-FTRL outperforms all other efficient and anytime mechanisms.** It also nearly equal or slightly outperform the state-of-the-art "ME" mechanism that requires significantly more compute (cf. Table 3). *The non-private baseline for StackOverflow uses per-user clipping as this improves performance by $\approx 0.5\%$ pp.

$O(T^3)$.[4] For example, generating the correlation matrix $\boldsymbol{B}$ for $T = 10^4$ takes around 24 hours (Choquette-Choo et al., 2023b). Moreover, it has a $O(T^2)$ cost per step. We find in this section that $\nu$-DP-FTRL achieves near state-of-the-art privacy-utility tradeoffs at a much smaller computational cost of $O(T)$ per iteration.

We compare with other *anytime* approaches for which the matrices $\boldsymbol{B}$ can extended to any time horizon $T$. The practitioner then need not specify $T$ in advance, but rather, can train for as long as necessary to achieve minimal model loss—it is common to, e.g., let algorithms run until certain conditions, like a maximum difference on the train-test loss, are met (Morgan & Bourlard, 1989). Moreover, general matrices $\boldsymbol{B}$ become prohibitive in terms of compute/memory as models scale up (Kaplan et al., 2020; Anil et al., 2023).

**Experiment Setup:** We use two standard benchmarks: example-level DP for image classification on the CIFAR-10 dataset and user-level DP for language modeling on the StackOverflow dataset. We use the same setup as (Kairouz et al., 2021a). We also stamp/restart all baselines as suggested in (Choquette-Choo et al., 2023b). This gives the baselines the advantage of an additional tuning parameter (tuned to minimize the squared error (3)), but does not affect their per-step training cost. We denote this by the suffix "$\times S$" for $S > 1$ in the plot. We tune all CIFAR-10 hyperparameters with a grid search, while we use hyperparameters reported from previous works for StackOverflow. Appendix G gives the full setup.

**Main Results:** Across both datasets, $\nu$-DP-FTRL outperforms all existing anytime mechanisms by a significant margin (Figure 4a). We find an average 3pp improvement that grows as $\varepsilon$ becomes small. Indeed, the proposed $\nu$-DP-FTRL makes up 30-80% of the gap between previous efficient approaches and the state-of-the-art and computationally intense ME approach. For instance, at $\varepsilon = 10$, we have $\nu$-DP-FTRL at $69.26\%$ nearly matches ME at $70.83\%$. In particular, $\nu$-DP-FTRL outperforms Optimal CC (Fichtenberger et al., 2023), which is equivalent to $\nu$-DP-FTRL with $\nu = 0$; this shows the practical importance of the exponential decay parameter $\nu$ in Eq. (7). For StackOverflow, we find that $\nu$-DP-FTRL outperforms the state-of-the-art ME across all $\varepsilon$ (Figure 4b) by $\approx 0.3\%$-points while requiring significantly less computation.

As $\varepsilon$ becomes small, DP-SGD can outperform DP-FTRL due to privacy amplification. We find that $\nu$-DP-FTRL outperforms DP-SGD for $\varepsilon \geq 4$ on CIFAR-10 ($63.02\%$ vs. $62.02\%$) and around $\varepsilon \approx 2$ for StackOverflow ($23.6\%$ versus $22.6\%$), showing its broad applicability. Finally, we observe that our mechanism achieves near non-private baselines on StackOverflow. A model trained via $\nu$-DP-FTRL gets $25.3\%$ validation accuracy at $\varepsilon = 8$, a mere $1\%$-point off from the non-private baseline.

---

[4]Note that in practice we take $T$ to be the number of steps of minibatch gradient descent, effectively doing several epochs over the data which differs from the theoretical setting considered in previous sections.

ACKNOWLEDGEMENTS

The authors thank H. Brendan McMahan, Fabian Pedregosa, Ian R. Manchester, Keith Rush, and Rahul Kidambi for fruitful discussions and helpful comments.

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

# Appendix

## Table of Contents

## A    FURTHER BACKGROUND ON DP-FTRL

In this appendix, we give a more detailed background of DP-FTRL, and its exact notion of differential privacy.

### A.1    DP-FTRL: THE MATRIX MECHANISM FOR PRIVATE LEARNING

The DP-FTRL algorithm (Kairouz et al., 2021a; Denisov et al., 2022) is obtained by adapting the matrix mechanism, originally designed for linear counting queries (Li et al., 2015), to optimization with a sequence $(\boldsymbol{g}_0, \ldots, \boldsymbol{g}_{T-1})$ of gradient vectors.

Algorithm 1 gives a detailed description of DP-FTRL. We give an alternate description of DP-FTRL with an invertible lower-triangular noise coefficient matrix $\boldsymbol{B} \in \mathbb{R}^{T \times T}$. Denoting $\boldsymbol{C} = \boldsymbol{B}^{-1}$, the iterates of DP-FTRL are generated by the update

$$\begin{pmatrix} \boldsymbol{\theta}_1 \\ \vdots \\ \boldsymbol{\theta}_T \end{pmatrix} = \begin{pmatrix} \boldsymbol{\theta}_0 \\ \vdots \\ \boldsymbol{\theta}_{T-1} \end{pmatrix} - \eta \boldsymbol{B} \left( \boldsymbol{C} \begin{pmatrix} \boldsymbol{g}_0 \\ \vdots \\ \boldsymbol{g}_{T-1} \end{pmatrix} + \begin{pmatrix} \boldsymbol{w}_0 \\ \vdots \\ \boldsymbol{w}_{T-1} \end{pmatrix} \right) \tag{11}$$

where $\eta$ is a learning rate and $\boldsymbol{w}_t \sim \mathcal{N}(\boldsymbol{0}, G^2 \sigma_{\mathsf{dp}}^2 \boldsymbol{I}_d)$ is i.i.d. Gaussian noise with a noise multiplier $\sigma_{\mathsf{dp}}$ and $G$ is the $\ell_2$ clip norm.

Following prior work, we also refer to $\boldsymbol{B}$ as the *noise correlation matrix* as the effective noise that is added to the optimization is the i.i.d. noise $(\boldsymbol{w}_0, \ldots, \boldsymbol{w}_{T-1})$ which are linearly correlated by the rows of the matrix $\boldsymbol{B}$. It is also common in the literature to refer to $\boldsymbol{C}$ as the *encoder*, while $\boldsymbol{B}$ is referred to as the *decoder*.

This privacy of (11) can be seen as a postprocessing of a single application of the Gaussian mechanism. Let $\boldsymbol{G}, \boldsymbol{W} \in \mathbb{R}^{T \times d}$ denote the matrix where each row is the gradient $\boldsymbol{g}_t$ (and respectively the noise $\boldsymbol{w}_t$). Then, (11) is effectively the postprocessing of one run of the Gaussian mechanism $\boldsymbol{C}\boldsymbol{G} + \boldsymbol{W}$. Under a neighborhood model that can change one row of $\boldsymbol{G}$, it can be seen that the maximum sensitivity of this operation is $\max_t \|\boldsymbol{C}_{:,t}\|_2^2$ (Denisov et al., 2022). This sensitivity logic also holds for adaptively chosen gradients; we postpone a formal description to Appendix A.2.

**Connection to the exposition in prior work:** Prior work introduced DP-FTRL differently. Letting $\boldsymbol{A} \in \mathbb{R}^{T \times T}$ denote the lower triangular matrix of all ones, update (11) can also be written as

$$\begin{pmatrix} \boldsymbol{\theta}_1 - \boldsymbol{\theta}_0 \\ \vdots \\ \boldsymbol{\theta}_T - \boldsymbol{\theta}_0 \end{pmatrix} = -\eta \widetilde{\boldsymbol{B}} \left( \boldsymbol{C} \begin{pmatrix} \boldsymbol{g}_0 \\ \vdots \\ \boldsymbol{g}_{T-1} \end{pmatrix} + \begin{pmatrix} \boldsymbol{w}_0 \\ \vdots \\ \boldsymbol{w}_{T-1} \end{pmatrix} \right), \tag{12}$$

where $\widetilde{\boldsymbol{B}} = \boldsymbol{A}\boldsymbol{B}$. The equivalence between (11) and (12) can be seen by multiplying (11) by $\boldsymbol{A}$, which is also equivalent to taking the cumulative sum of the rows of a matrix. In this notation, the objective from (3) used in previous work to find the matrix $\boldsymbol{B}$ can equivalently be written as

$$\varphi(\boldsymbol{B}) = \|\widetilde{\boldsymbol{B}}\|_F^2 = \|\boldsymbol{A}\boldsymbol{B}\|_F^2 .$$

**DP-FTRL with Toeplitz matrices:** We focus on the class of lower-triangular and Toeplitz matrices $\boldsymbol{B}$. That is, $[\boldsymbol{B}]_{t,t'} = \beta_{t-t'}$ for all $t \geq t'$ where $\boldsymbol{\beta} = (\beta_0, \ldots, \beta_{T-1})$ is the first column of $\boldsymbol{B}$.[5] In this case, (11)

---

[5]This implies that $\boldsymbol{C} = \boldsymbol{B}^{-1}$ is also lower-triangular and Toeplitz (Kucerovsky et al., 2016, Prop. 2.2 & Rem. 2.3).

reduces to this simple update:

$$\boldsymbol{\theta}_{t+1} = \boldsymbol{\theta}_t - \eta \left( \boldsymbol{g}_t + \sum_{\tau=0}^{t} \beta_\tau \boldsymbol{w}_{t-\tau} \right) . \tag{13}$$

This lets us study DP-FTRL as a time-invariant stochastic process and characterize its stationary behavior.

## A.2 DIFFERENTIAL PRIVACY IN ADAPTIVE STREAMS

**Neighboring streams:** We consider learning algorithms as operating over streams of gradients $\boldsymbol{g}_0, \boldsymbol{g}_1, \ldots \in \mathbb{R}^d$. We consider differential privacy (DP) under the "zero-out" notion of neighborhood (Kairouz et al., 2021a). Two streams $\boldsymbol{G} = (\boldsymbol{g}_0, \ldots, \boldsymbol{g}_{T-1})$ and $\boldsymbol{G}' = (\boldsymbol{g}'_0, \ldots, \boldsymbol{g}'_{T-1})$ of length $T$ are said to be neighbors if $\boldsymbol{g}_\tau = \boldsymbol{g}'_\tau$ for all positions $\tau \leq T-1$ except possibly one position $t$ where one of $\boldsymbol{g}_t$ or $\boldsymbol{g}'_t$ is the zero vector.

The zero-out neighborhood is standard in prior works on DP-FTRL (e.g. Kairouz et al., 2021a; Denisov et al., 2022). For a further discussion of different notions of neighborhood, we refer to (Ponomareva et al., 2023, Sec. 2.1.1). This guide suggests that the semantics of the zero-out neighborhood are roughly the same as that of the usual add/remove notion of neighborhood.

**DP with adaptive continual release:** It is customary to formalize DP with adaptive streams as a privacy game between a mechanism $\mathcal{M}$ and a privacy adversary $\mathcal{A}$. This is known as the *adaptive continual release setting* (Jain et al., 2023). The game makes a binary choice $b \in \{0, 1\}$ ahead of time — this remains fixed throughout and is not revealed to either $\mathcal{M}$ or $\mathcal{A}$. Each round $t$ consists of four steps:

- $\mathcal{M}$ sends the current model parameters $\boldsymbol{\theta}_t$ to the adversary $\mathcal{A}$;
- $\mathcal{A}$ generates two gradient vectors $\boldsymbol{g}_t, \boldsymbol{g}'_t$ (e.g. as $\nabla f(\boldsymbol{\theta}_t; \boldsymbol{z}_t)$ for $\boldsymbol{z}_t \sim \mathbb{P}_{\mathsf{data}}$ or simply the zero vector);
- the game accepts these inputs if the partial streams $(\boldsymbol{g}_0, \ldots, \boldsymbol{g}_t)$ and $(\boldsymbol{g}'_0, \ldots, \boldsymbol{g}'_t)$ are neighbors;
- $\mathcal{M}$ receives $\boldsymbol{g}_t$ if $b = 0$ else $\boldsymbol{g}'_t$.

DP in this setting requires that the adversary cannot infer the value of $b$, i.e., the distribution of $\boldsymbol{\theta}_{0:T}|b=0$ to be "close" to that of $\boldsymbol{\theta}_{0:T}|b=1$ (where the definition of "closeness" depends on the DP variant). For instance, $(\varepsilon, \delta)$-DP (Dwork et al., 2006) requires for each $b \in \{0, 1\}$ and any outcome set $S$ that

$$\mathbb{P}(\boldsymbol{\theta}_{0:T} \in S \,|\, b) \leq \exp(\varepsilon)\,\mathbb{P}(\boldsymbol{\theta}_{0:T} \in S \,|\, 1-b) + \delta \,.$$

Similarly, $\rho$-zCDP (Bun & Steinke, 2016) in this setting requires that the Rényi $\alpha$-divergence between the distribution $P_0$ of $\boldsymbol{\theta}_{0:T}|b=0$ and the distribution $P_1$ of $\boldsymbol{\theta}_{0:T}|b=1$ are close:

$$D_\alpha(P_0\|P_1) \leq \rho\alpha$$

for all $\alpha \in (0, \infty)$. Following standard arguments (e.g. Balle et al., 2020), $\rho$-zCDP in this setting implies $(\varepsilon_\delta, \delta)$-DP with

$$\varepsilon_\delta \leq \inf_{\alpha>1} \left\{ \rho\alpha + \frac{1}{\alpha-1} \log\left(\frac{1}{\alpha\delta}\right) + \log(1-\alpha^{-1}) \,. \right\}$$

DP-FTRL satisfies a zCDP guarantee as described in Theorem 1.1 in §1. This guarantee is equivalent to the one obtained by interpreting (11) as the postprocessing of one run of the Gaussian mechanism $\boldsymbol{CG} + \boldsymbol{W}$.

## B ASYMPTOTICS OF DP-FTRL FOR MEAN ESTIMATION

We now prove Theorem 2.1 on mean estimation.

*Proof of Theorem 2.1.* We rewrite the iterates of DP-FTRL as a linear time-invariant (LTI) dynamical system, whose stationary variance can be analyzed in the Fourier domain directly.

**Notation:** Since $|\nabla f(\theta; z)| = |z| \leq 1$ and $G \geq 1$, there is no gradient clipping. We consider a mean-adjusted version of the learning dynamics: let $\delta_t = \theta_t - \mathbb{E}[z]$ and $u_t = \frac{z_t - \mathbb{E}[z]}{\sigma_{\mathsf{sgd}}}$. This allows us to reason about the deviation of the parameters $\theta_t$ from the true mean $\mathbb{E}[z]$; indeed, it turns out that $\lim_{t \to \infty} \mathbb{E}[\delta_t] = 0$. The objective we optimize for can now be succinctly written as $\lim_{t \to \infty} \mathbb{E}[\delta_t^2]$.

**LTI System:** Our next step is to write this as an LTI system (see Appendix F.1 for a review). Thus, the sequence $(\delta_t)_{t=0}^{\infty}$ produced by (2) evolves as

$$\delta_{t+1} = (1 - \eta)\delta_t + \eta\sigma_{\mathsf{sgd}}u_t - \eta\sigma_{\mathsf{dp}}G\sum_{\tau=0}^{t}\beta_\tau w_{t-\tau} \quad t = 0, 1, \dots. \tag{14}$$

This is an LTI system with input $\boldsymbol{x}_t = (u_t; w_t) \in \mathbb{R}^2$ and output $\boldsymbol{y}_t = [\delta_t] \in \mathbb{R}^1$. We can verify its asymptotic stability by examining the dynamics under zero inputs: $u_t = 0$ and $w_t = 0$ for all $t$. This gives $\delta_t = (1 - \eta)^t\delta_0 \to 0$ as $t \to \infty$. Thus, this system is asymptotically stable. Further, we can also get from taking expectations that $\mathbb{E}[\delta_t] = (1 - \eta)^t\delta_0 \to 0$. Thus, our objective $F_\infty(B) = \lim_{t \to \infty} \mathbb{E}[\delta_t^2]$ is the limiting (stationary) variance of $\delta_t$.

To invoke results from the LTI literature, it is convenient to re-index time to start from $t = -\infty$ so that the behavior at $t = 0$ describes the stationary behavior. Hence, the dynamics can be replaced by

$$\delta_{t+1} = (1 - \eta)\delta_t + \eta\sigma_{\mathsf{sgd}}u_t - \eta\sigma_{\mathsf{dp}}G\sum_{\tau=0}^{\infty}\beta_\tau w_{t-\tau} \quad \forall t \in \mathbb{Z} \tag{15}$$

where $\mathbb{Z}$ denotes the set of integers and the objective can be taken to be $F_\infty(B) = \mathbb{E}[\delta_0^2]$.

**Transfer function of the LTI system:** The transfer function $\boldsymbol{G}(\omega)$ of the LTI system (15) is a complex matrix of shape $1 \times 2$ (see §F.1 for definitions), which can be written as

$$\boldsymbol{G}(\omega) = \left( \frac{-\eta}{1 - \eta - \exp(i\omega)} \quad \frac{\eta\,B(\omega)}{1 - \eta - \exp(i\omega)} \right). \tag{16}$$

The transfer function has the property that for any input sequences $u_t$ and $w_t$ with DTFT $U(\omega)$ and $Z(\omega)$, the output sequence satisfies $Y(\omega) = \boldsymbol{G}(\omega)\begin{pmatrix} U(\omega) \\ Z(\omega) \end{pmatrix}$.

**Stationary variance of the LTI system:** The stationary variance $\lim_{t \to \infty} \mathbb{E}[\delta_t^2]$ admits a nice closed form expression in the Fourier domain since its inputs are white noise. In particular, $u_t$ is i.i.d. in each step and independent of the DP noise $w_t$, so that the power spectral density of the sum of these two noise sources is simply the sum of the power spectral densities of the individual sources; the resulting expression is summarized in Theorem F.2.

We first calculate the input covariance is

$$\boldsymbol{\Sigma} = \mathbb{E}[\boldsymbol{x}_t \otimes \boldsymbol{x}_t] = \begin{pmatrix} \sigma_{\mathsf{sgd}}^2 & 0 \\ 0 & G^2\sigma_{\mathsf{dp}}^2 \end{pmatrix}. \tag{17}$$

We can then use Theorem F.2 from §F.1 to obtain an expression for the stationary variance $F_\infty(B) = \mathbb{E}[\delta_0^2]$:

$$F_\infty(B) = \frac{1}{2\pi}\int_{-\pi}^{\pi}\boldsymbol{G}(\omega)\boldsymbol{\Sigma}\boldsymbol{G}(\omega)^* \,\mathrm{d}\omega = \frac{\eta^2}{2\pi}\int_{-\pi}^{\pi}\frac{|B(\omega)|^2\frac{G^2}{2\rho}\gamma_\infty^2(B) + \sigma_{\mathsf{sgd}}^2}{|1 - \eta - \exp(i\omega)|^2}\,\mathrm{d}\omega.$$

Note that above $\boldsymbol{G}(\omega)^*$ denotes the conjugate transpose of the complex matrix $\boldsymbol{G}(\omega)$.

**Optimizing for the correlation in frequency domain:** The dependence of $F_\infty$ on $B$ is via the first term:

$$\frac{\eta^2}{2\pi} \int_{-\pi}^{\pi} \frac{|B(\omega)|^2 \frac{G^2}{2\rho} \gamma_\infty^2(B)}{|1 - \eta - \exp(i\omega)|^2} \, d\omega \overset{(5)}{=} \frac{\eta^2 \frac{G^2}{2\rho}}{4\pi^2} \left( \int_{-\pi}^{\pi} \frac{|B(\omega)|^2}{|1 - \eta - \exp(i\omega)|^2} \, d\omega \right) \left( \int_{-\pi}^{\pi} \frac{d\omega}{|B(\omega)|^2} \right). \quad (18)$$

The stationary variance's dependence on $B$ in (18) is a product of a linear function of $|B|^2$ and $\frac{1}{|B|^2}$. The former comes via the variance and the latter through the sensitivity $\gamma_\infty(B)$ via (5). The optimal value of $B$ must balance these two considerations. By the Cauchy-Schwarz inequality, the product is minimized when

$$\frac{|B^\star(\omega)|^2}{|1 - \eta - \exp(i\omega)|^2} = \frac{1}{|B^\star(\omega)|^2} \iff |B^\star(\omega)| = |\sqrt{1 - \eta - \exp(i\omega)}|, \quad (19)$$

and the minimum value is equal to

$$\frac{\eta^2 G^2 \sigma_{\mathsf{dp}}^2}{4\pi^2} \left( \int_{-\pi}^{\pi} \frac{d\omega}{|1 - \eta - \exp(i\omega)|} \right)^2.$$

The proof of the error bound now follows by computing and bounding the integral $\int_{-\pi}^{\pi} d\omega/|1-\eta-\exp(i\omega)|$. This can be bounded via reductions to standard integrals whose asymptotics are known (see Lemma F.15 and Property F.10 from §F.4). Similarly, Corollary C.5 can be used to bound the $\sigma_{\mathsf{sgd}}^2$ term in (17).

**Optimal correlation in time-domain:** Next, we derive the time-domain description by taking $B^\star(\omega) = \sqrt{1 - (1 - \eta)\exp(-i\omega)}$ (which amounts to fixing a phase in (19) above). We use the Maclaurin series expansion $\sqrt{1 + z} = \sum_{t=0}^{\infty} \binom{1/2}{t} z^t$ of the square root function to get

$$B^\star(\omega) = \sum_{t=0}^{\infty} (-1)^t \binom{1/2}{t} (1 - \eta)^t \exp(-i\omega t).$$

Comparing this to the definition of the discrete-time Fourier transform $B^\star(\omega) = \sum_{t=0}^{\infty} \beta_t^\star \exp(-i\omega t)$ gives the claimed expression for $\boldsymbol{\beta}^\star$. $\qquad\square$

Note also that the optimal correlations scale as $|\beta_t^\star| = \Theta(t^{-3/2} \exp(-\eta t))$.

## C ASYMPTOTICS OF DP-FTRL FOR LINEAR REGRESSION

The goal of this section is to prove Theorem 2.2. The proof relies heavily on the following matching upper and lower bounds on the stationary error of Noisy-FTRL with any noise correlations $\boldsymbol{\beta}$ in the frequency domain using its discrete-time Fourier transform (DTFT) $B$ as:

$$F_\infty(B) = \Theta\left( \eta \sigma_{\mathsf{sgd}}^2 \mathsf{Tr}[\boldsymbol{H}] + \eta^2 G^2 \rho^{-1} \gamma_\infty^2(B) \int_{-\pi}^{\pi} |B(\omega)|^2 h(\omega) \, d\omega \right), \quad (20)$$

where the function $h : [-\pi, \pi] \to \mathbb{R}$ depends on the eigenvalues $\lambda_1, \ldots, \lambda_d$ of the input covariance $\boldsymbol{H}$:

$$h(\omega) = \sum_{j=1}^{d} \frac{\lambda_j}{|1 - \exp(i\omega) - \eta\lambda_j|^2}. \quad (21)$$

The outline of the section is

Table 4: Asymptotic suboptimality of Noisy-SGD and Noisy-FTRL for linear regression with Gaussian inputs based on the eigenvalues $\lambda_k$ of the Hessian $\boldsymbol{H}$. We give the bounds in terms of the learning rate $\eta$, dimension $d$, the effective dimension $d_{\text{eff}} = \text{Tr}\,[\boldsymbol{H}]\,/\|\boldsymbol{H}\|_2$, and the noise variance $\rho^{-1}$ representing the privacy level. We take $G = 1$ and $\|\boldsymbol{H}\|_2 = 1$ w.l.o.g. Noisy-FTRL is always better at large dimension $d$ or small learning rate $\eta$.

| Eigenvalues of $\boldsymbol{H}$ | Effective dim. $d_{\text{eff}}$ | Noisy-SGD | Noisy-FTRL | Ratio of $\frac{\text{Noisy-FTRL}}{\text{Noisy-SGD}}$ |
|---|---|---|---|---|
| $\lambda_k = 1$ | $d$ | $\eta d \rho^{-1}$ | $\eta^2 d \rho^{-1} \log^2(\frac{1}{\eta})$ | $\eta \log^2(\frac{1}{\eta})$ |
| $\lambda_k = 1/\sqrt{k}$ | $\sqrt{d}$ | $\eta d \rho^{-1}$ | $\eta^2 \sqrt{d} \rho^{-1} \log^2(\frac{d}{\eta})$ | $\frac{\eta}{\sqrt{d}} \log^2(\frac{d}{\eta})$ |
| $\lambda_k = k^{-a}\ (a < 1)$ | $\frac{d^{1-a}}{1-a}$ | $\eta d \rho^{-1}$ | $(1-a)^{-1}\eta^2 d^{1-a}\rho^{-1}\log^2(d/\eta)$ | $\frac{\eta}{(1-a)d^a}\log^2\left(\frac{d}{\eta}\right)$ |
| $\lambda_k = 1/k$ | $\log d$ | $\eta d \rho^{-1}$ | $\eta^2 \rho^{-1}\log^3(\frac{d}{\eta})$ | $\frac{\eta}{d}\log^3(\frac{d}{\eta})$ |
| $\lambda_k = 1/k^2$ | constant | $\eta d \rho^{-1}$ | $\eta^2 \rho^{-1}\log^2(\frac{d}{\eta})$ | $\frac{\eta}{d}\log^3(\frac{d}{\eta})$ |
| $\lambda_k = k^{-a}\ (a > 1)$ | $\frac{a}{a-1}$ | $\eta d \rho^{-1}$ | $\left(\frac{a^2}{a-1}\right)\eta^2\rho^{-1}\log^2\left(\frac{d}{\eta}\right)$ | $\left(\frac{a^2}{a-1}\right)\frac{\eta}{d}\log^2\left(\frac{d}{\eta}\right)$ |

- **Appendix C.1**: Setup, including notation, and assumptions.
- **Appendix C.2**: Proofs of the upper bound of (20), specifically Theorem C.15 (see also Theorem C.14 for the time-domain description).
- **Appendix C.3**: Proofs of the lower bound of (20), specifically Theorem C.18.
- **Appendix C.4**: Asymptotics of $\nu$-Noisy-FTRL.
- **Appendix C.5**: Asymptotics of anti-PGD (see Table 2).
- **Appendix C.6**: Effective Dimension and its Connection to the Stable Rank.
- **Appendix C.7**: Proofs of intermediate technical results.

The separation between Noisy-SGD and $\nu$-Noisy-FTRL is further illustrated in Table 4. Following common practice (e.g. Caponnetto & De Vito, 2007), we compare the rates for various regimes of eigenvalue decays for $\boldsymbol{H}$.

## C.1 SETUP, ASSUMPTIONS, AND NOTATION

### C.1.1 SETUP

Recall that we wish to minimize the objective

$$F(\boldsymbol{\theta}) = \mathbb{E}_{(\boldsymbol{x},y)\sim\mathbb{P}_{\text{data}}}\left[(y - \langle\boldsymbol{\theta}, \boldsymbol{x}\rangle)^2\right]. \tag{22}$$

**Stochastic gradients:** Given $(\boldsymbol{x}, y) \sim \mathbb{P}_{\text{data}}$, the vector

$$\boldsymbol{g} := (\boldsymbol{x} \otimes \boldsymbol{x})\,\boldsymbol{\theta} - y\boldsymbol{x} = (\boldsymbol{x} \otimes \boldsymbol{x})(\boldsymbol{\theta} - \boldsymbol{\theta}_\star) - \xi\boldsymbol{x}$$

is a stochastic gradient of $F$ at $\boldsymbol{\theta}$, i.e., $\mathbb{E}[\boldsymbol{g}] = \nabla F(\boldsymbol{\theta})$.

**Noisy-FTRL Iterations:** We specialize the Noisy-FTRL algorithm with Toeplitz noise correlations. Let $T$ denote the number of iterations and $\boldsymbol{\beta}_{:T} = (\beta_0, \ldots, \beta_{T-1})$ denote the first column of the Toeplitz matrix $\boldsymbol{B} = \text{Toeplitz}(\boldsymbol{\beta}_{:T}) \in \mathbb{R}^{T\times T}$. Starting from a given $\boldsymbol{\theta}_0 \in \mathbb{R}^d$, Noisy-FTRL samples a fresh input-output

pair $(\boldsymbol{x}_t, y_t) \sim \mathbb{P}_{\text{data}}$ and noise $\boldsymbol{w}_t$ to set

$$\boldsymbol{\theta}_{t+1} = \boldsymbol{\theta}_t - \eta\left((\boldsymbol{x}_t \otimes \boldsymbol{x}_t)\boldsymbol{\theta}_t - y_t\boldsymbol{x}_t\right) - \eta\sum_{\tau=0}^{t}\beta_\tau\boldsymbol{w}_{t-\tau}\,. \tag{23}$$

Recall that the sensitivity $\gamma_T(\boldsymbol{\beta})$ equals to the maximum columns norm of $\boldsymbol{B}^{-1} = (\text{Toeplitz}(\boldsymbol{\beta}))^{-1}$:

$$\gamma_T(\boldsymbol{\beta}) = \max_{\tau=0,\dots,T-1}\left\|\boldsymbol{B}^{-1}\boldsymbol{e}_\tau\right\|_2\,, \tag{24}$$

where $\boldsymbol{e}_\tau = \left(\mathbb{I}(j=\tau)\right)_{\tau=0}^{T-1} \in \mathbb{R}^T$ is a standard basis vector. Note that the submatrix $[\boldsymbol{B}^{-1}]_{0:m,0:m}$ of the first $m$ rows and columns of $\boldsymbol{B}^{-1}$ equals $(\text{Toeplitz}(\beta_0,\dots,\beta_{m-1}))^{-1}$. Thus, the sensitivity $\gamma_t(\boldsymbol{\beta})$ is an increasing function of $t$ always.

**Infinite-time limit of Noisy-FTRL:** We study the Noisy-FTRL error under the limit $T \to \infty$ with an infinite sequence $\boldsymbol{\beta} = (\beta_0, \beta_1, \dots)$ of weights.

It is also convenient to re-index time to start from $t = -\infty$ and consider the sequence $(\boldsymbol{\theta})_{t=-\infty}^{\infty}$ produced by analogue of Equation (23), which reads

$$\boldsymbol{\theta}_{t+1} = \boldsymbol{\theta}_t - \eta\left((\boldsymbol{x}_t \otimes \boldsymbol{x}_t)\boldsymbol{\theta}_t - y_t\boldsymbol{x}_t\right) - \eta\sum_{\tau=0}^{\infty}\beta_\tau\boldsymbol{w}_{t-\tau}\,. \tag{25}$$

Note that this includes a summation over all previous DP noise $(\boldsymbol{w}_\tau)_{\tau=-\infty}^{t}$. For this sum to have finite variance, we require $\sum_{\tau=0}^{\infty}\beta_\tau^2 < \infty$ or that $\boldsymbol{\beta} \in \ell^2$, the space of all square-summable infinite sequences. We will assume this holds throughout.

**Sensitivity in the infinite limit:** We define the sensitivity $\gamma_\infty(\boldsymbol{\beta})$ by considering the linear operator $\boldsymbol{B} = \text{Toeplitz}(\boldsymbol{\beta})$ as the convolution operator $[\boldsymbol{B}\boldsymbol{w}]_t = \sum_{\tau=0}^{\infty}\beta_\tau\boldsymbol{w}_{t-\tau}$ on input $\boldsymbol{w} = (\boldsymbol{w}_\tau)_{\tau=-\infty}^{\infty}$. Let $\boldsymbol{B}^{-1}$ be the inverse operator to $\boldsymbol{B}$, assuming it exists. Note that the column norms $\left\|\boldsymbol{B}^{-1}\boldsymbol{e}_\tau\right\|_2$ from (24) become equal for all $\tau$ as $T \to \infty$. Thus, we get that the limiting sensitivity in the infinite time limit equals

$$\gamma_\infty(\boldsymbol{\beta}) = \left\|\boldsymbol{B}^{-1}\boldsymbol{e}_0\right\|_2 \tag{26}$$

for $\boldsymbol{B} = \text{Toeplitz}(\boldsymbol{\beta})$ and $\boldsymbol{e}_0 = (\mathbb{1}\,(\tau=0))_{\tau=0}^{\infty} \in \ell^2$. If $\boldsymbol{e}_0 \notin \text{Range}(\boldsymbol{B})$, then we take $\gamma_\infty(\boldsymbol{\beta}) = \infty$.

**Frequency-domain description:** Our analysis relies on the frequency-domain representation $B : [-\pi, \pi] \to \mathbb{C}$ of $\boldsymbol{\beta}$ obtained via a discrete-time Fourier transform (DTFT) and defined as

$$B(\omega) = \sum_{t=0}^{\infty}\beta_t\,\exp(i\omega t)\,. \tag{27}$$

The sequence $\boldsymbol{\beta}$ can be recovered from $B(\omega)$ using the inverse Fourier transform. Note that $\beta \in \ell^2$ is equivalent to $B \in L^2$, the space of square-integrable functions, by Parseval's theorem. The sensitivity (26) can be defined in the Fourier domain as follows.

**Property C.1.** *Let $B(\omega)$ denote the DTFT of $\boldsymbol{\beta} \in \ell^2$. Then, we have*

$$\gamma_\infty^2(\boldsymbol{\beta}) = \gamma_\infty^2(B) := \frac{1}{2\pi}\int_{-\pi}^{\pi}\frac{\mathrm{d}\omega}{|B(\omega)|^2}\,. \tag{28}$$

*Proof.* Let $\boldsymbol{z} = \boldsymbol{B}^{-1}\boldsymbol{e}_0$ be the solution of the linear system $\boldsymbol{B}\boldsymbol{z} = \boldsymbol{e}_0$. Let $Z(\omega)$ denote the DTFT of $\boldsymbol{z}$. Since the linear operator $\boldsymbol{B}$ is a convolution with the weights of $\boldsymbol{\beta}$, this system can be expressed in the Fourier domain as

$$B(\omega)Z(\omega) = \sum_{\tau=0}^{\infty}[\boldsymbol{e}_0]_\tau\exp(-i\omega\tau) = 1\,.$$

Thus, $Z(\omega) = 1/B(\omega)$. We complete the proof with Parseval's theorem: $\|\boldsymbol{z}\|_2^2 = \frac{1}{2\pi}\int_{-\pi}^{\pi}|Z(\omega)|^2\,\mathrm{d}\omega$. $\qquad\square$

### C.1.2 ASSUMPTIONS

We prove the stationary error bounds under a relaxation of the assumptions in §2.2.

**Assumption C.2.** *The data distribution* $\mathbb{P}_{\mathsf{data}}$ *satisfies the following:*

(A1) **Input Mean and Covariance**: *The inputs have mean* $\mathbb{E}[\boldsymbol{x}] = \mathbf{0}$ *and covariance* $\mathbb{E}[\boldsymbol{x} \otimes \boldsymbol{x}] =: \boldsymbol{H}$. *Further,* $L = \lambda_1 \geq \cdots \geq \lambda_d =: \mu > 0$ *are the eigenvalues of* $\boldsymbol{H}$.

(A2) **Noise Mean and Variance**: *There exists a* $\boldsymbol{\theta}_\star \in \mathbb{R}^d$ *such that* $y = \langle \boldsymbol{\theta}_\star, \boldsymbol{x} \rangle + \xi$ *where* $\xi$ *is independent of* $\boldsymbol{x}$ *with* $\mathbb{E}[\xi] = 0$ *and* $\mathbb{E}[\xi^2] \leq \sigma_{\mathsf{sgd}}^2$.

(A3) **Input Kurtosis**: *There exists* $R^2 < \infty$ *such that* $\mathbb{E}\left[\|\boldsymbol{x}\|_2^2 (\boldsymbol{x} \otimes \boldsymbol{x})\right] \preceq R^2 \boldsymbol{H}$. *Moreover, for every PSD* $\boldsymbol{P} \in \mathbb{S}_+^d$ *that commutes with* $\boldsymbol{H}$ *(i.e.,* $\boldsymbol{P}\boldsymbol{H} = \boldsymbol{H}\boldsymbol{P}$*), we have* $\mathbb{E}\left[(\boldsymbol{x} \otimes \boldsymbol{x})\boldsymbol{H}^{-1/2}\boldsymbol{P}\boldsymbol{H}^{-1/2}(\boldsymbol{x} \otimes \boldsymbol{x})\right] \preceq C_{\mathsf{kurt}} \, \mathsf{Tr}\,[\boldsymbol{P}] \, \boldsymbol{H}$ *for some* $C_{\mathsf{kurt}} < \infty$.

These assumptions are fairly standard in the context of linear regression. Assumption (A1) implies that the Hessian matrix of objective $F(\boldsymbol{\theta})$ is $\boldsymbol{H} \succ 0$. Thus, $F$ is $L$-smooth and $\mu$-strongly convex. Assumption (A2) implies that $\boldsymbol{\theta}_\star$ is the unique global minimizer of $F$ and that the linear model is well-specified. The upper bounds we prove continue to hold in the case where the linear model is mis-specified (i.e. $\xi$ is not independent of $\boldsymbol{x}$) but we still have $\mathbb{E}[\xi^2 (\boldsymbol{x} \otimes \boldsymbol{x})] \preceq \sigma_{\mathsf{sgd}}^2 \boldsymbol{H}$.

Assumption (A3) is a kurtosis (i.e. 4th moment) assumption on the input distribution; we will momentarily show that it follows with absolute constants when $\boldsymbol{x} \sim \mathcal{N}(\mathbf{0}, \boldsymbol{H})$. More generally, by taking a trace, we get from Jensen's inequality that $\mathsf{Tr}\,[\boldsymbol{H}] \leq R^2$. The case of $\boldsymbol{P} = \boldsymbol{I}$ of the second part of Assumption (A3) has a special significance in the literature (e.g. Hsu et al., 2014; Jain et al., 2018) as $C_{\mathsf{kurt}}\mathsf{Tr}\,[\boldsymbol{I}] = C_{\mathsf{kurt}}d$ is the number of samples that allows the spectral concentration of the empirical covariance to the population covariance $\boldsymbol{H}$.

**Property C.3.** *if* $\boldsymbol{x} \sim \mathcal{N}(\mathbf{0}, \boldsymbol{H})$, *we have that Assumption (A3) holds with* $R^2 \leq 3\,\mathsf{Tr}\,[\boldsymbol{H}]$ *and* $C_{\mathsf{kurt}} \leq 3$.

*Proof.* Let $\boldsymbol{z} = \boldsymbol{H}^{-1/2}\boldsymbol{x}$ be element-wise independent and distributed as a standard Gaussian. For the first part, denote $\boldsymbol{M} = \boldsymbol{H}^{-1/2}\mathbb{E}[\|\boldsymbol{x}\|_2^2 \boldsymbol{x} \otimes \boldsymbol{x}]\boldsymbol{H}^{-1/2} = \mathbb{E}[\langle \boldsymbol{z}, \boldsymbol{H}\boldsymbol{z} \rangle \boldsymbol{z} \otimes \boldsymbol{z}]$. Elementary properties of the standard Gaussian distribution give

$$\mathbb{E}[z_k z_l z_j^2] = \begin{cases} 3, & \text{if } k = l = j \\ 1, & \text{if } k = l \neq i \\ 0, & \text{if } k \neq l, \end{cases} \quad \text{and} \quad \mathbb{E}[z_k z_l z_j z_{j'}] = \begin{cases} 1, & \text{if } k = j \text{ and } l = j' \\ 1, & \text{if } k = j' \text{ and } l = j \\ 0, & \text{else} \end{cases}$$

for $j \neq j'$. Thus, we have $\boldsymbol{M} = 2\boldsymbol{H} + \mathsf{Tr}\,[\boldsymbol{H}]\,\boldsymbol{I}$. This gives

$$\mathbb{E}[\|\boldsymbol{x}\|_2^2 \boldsymbol{x} \otimes \boldsymbol{x}] = \boldsymbol{H}^{1/2}\boldsymbol{M}\boldsymbol{H}^{1/2} = 2\boldsymbol{H}^2 + \mathsf{Tr}\,[\boldsymbol{H}]\,\boldsymbol{H} \preceq 3\mathsf{Tr}\,[\boldsymbol{H}]\,\boldsymbol{H}\,.$$

For the second part, let $\boldsymbol{H} = \boldsymbol{U}\boldsymbol{\Lambda}\boldsymbol{U}^\top$ and $\boldsymbol{P} = \boldsymbol{U}\boldsymbol{\Sigma}\boldsymbol{U}^\top$ be the eigenvalue decomposition of $\boldsymbol{H}, \boldsymbol{P}$ respectively (since they commute, they are simultaneously diagonalized in the same basis given by the columns of $\boldsymbol{U}$). Since $\boldsymbol{U}^\top \boldsymbol{z}$ has the same distribution as $\boldsymbol{z}$ by the spherical invariance of Gaussians, we have,

$$\boldsymbol{H}^{-1/2}\mathbb{E}\left[(\boldsymbol{x} \otimes \boldsymbol{x})\boldsymbol{H}^{-1/2}\boldsymbol{P}\boldsymbol{H}^{-1/2}(\boldsymbol{x} \otimes \boldsymbol{x})\right]\boldsymbol{H}^{-1/2} = \mathbb{E}\left[(\boldsymbol{z} \otimes \boldsymbol{z})\boldsymbol{P}(\boldsymbol{z} \otimes \boldsymbol{z})\right] = \boldsymbol{U}\,\mathbb{E}\left[(\boldsymbol{z} \otimes \boldsymbol{z})\boldsymbol{\Sigma}(\boldsymbol{z} \otimes \boldsymbol{z})\right]\boldsymbol{U}^\top.$$
(29)

Each off-diagonal entry of $\mathbb{E}\left[(\boldsymbol{z} \otimes \boldsymbol{z})\boldsymbol{\Sigma}(\boldsymbol{z} \otimes \boldsymbol{z})\right]$ is zero since it involves expected odd powers of Gaussians. Its $j^{\text{th}}$ diagonal entry equals (denoting $\sigma_j := [\boldsymbol{\Sigma}]_{j,j}$)

$$\mathbb{E}\left[z_j^2 \sum_{k=1}^d \sigma_k z_k^2\right] = \sigma_j \mathbb{E}[z_j^4] + \sum_{k \neq j} \sigma_k \, \mathbb{E}[z_j^2 z_k^2] = 2\sigma_j + \mathsf{Tr}\,[\boldsymbol{\Sigma}]\,.$$

This gives $\mathbb{E}\left[(z \otimes z)\Sigma(z \otimes z)\right] = 2\Sigma + \mathsf{Tr}\left[\Sigma\right]I \preceq 3\mathsf{Tr}\left[\Sigma\right]I$ since $\Sigma \succeq 0$. Plugging this back into (29) and rearranging completes the proof. $\qquad\square$

### C.1.3 NOTATION

We set up some notation, that we use throughout this section.

- It is convenient to rewrite the Noisy-FTRL recursion in terms of the difference $\theta'_t := \theta_t - \theta_\star$. We can rewrite the Noisy-FTRL recursion (25) as

$$\theta'_{t+1} = \left(I - \eta(x_t \otimes x_t)\right)\theta'_t + \eta\,\xi_t x_t - \eta\sum_{\tau=0}^{\infty}\beta_\tau w_{t-\tau}\,. \tag{30}$$

We will analyze this recursion.
- We describe the asymptotic suboptimality in terms of the self-adjoint linear operator $T : \ell^2 \to \ell^2$ defined by

$$[T\beta]_t = \sum_{\tau=0}^{\infty}\beta_\tau\sum_{j=1}^{d}(1-\eta\lambda_j)^{|t-\tau|}\,. \tag{31}$$

This operator is positive semi-definite, as we show in Lemma C.6 below. In the finite time setting, we could represent $T$ by the matrix

$$T = \begin{bmatrix} d & \sum_{j=1}^{d}(1-\eta\lambda_j) & \sum_{j=1}^{d}(1-\eta\lambda_j)^2 & \cdots \\ \sum_{j=1}^{d}(1-\eta\lambda_j) & d & \sum_{j=1}^{d}(1-\eta\lambda_j) & \cdots \\ \sum_{j=1}^{d}(1-\eta\lambda_j)^2 & \sum_{j=1}^{d}(1-\eta\lambda_j) & d & \cdots \\ \vdots & & & \vdots \end{bmatrix}$$

We only consider step-size $0 < \eta < 1/R^2$, which implies that $1 - \eta\lambda_j \in (0,1)$ for all $j$.
- For $j = 1, \ldots, d$, define $T_j : \ell^2 \to \ell^2$ as the linear operator

$$[T_j\beta]_t = \sum_{\tau=0}^{\infty}\beta_\tau(1-\eta\lambda_j)^{|t-\tau|}\,. \tag{32}$$

Note that $[T_j\beta]_t < \infty$ always since

$$\sum_{\tau=0}^{\infty}\beta_\tau(1-\eta\lambda_j)^{|t-\tau|} \leq \frac{2\|\beta\|_\infty}{\eta\lambda_j} < \infty\,,$$

since $0 < \eta\lambda < 1$. Thus, we have that $T = \sum_{j=1}^{d}T_j$ by the bounded convergence theorem. Further, we show in the upcoming Lemma C.6 that each $T_j$ is PSD.
- Define $\Sigma_\beta, P_\beta \in \mathbb{S}^d$ as

$$\Sigma_\beta := \mathsf{diag}\left((\langle\beta, T_j\beta\rangle)_{j=1}^{d}\right), \quad \text{and} \quad P_\beta = U\Sigma_\beta U^\top\,, \tag{33}$$

where $U$ is the eigen-basis of $H = U\Lambda U^\top$. By definition, $P_\beta$ commutes with $H$ since $P_\beta H = HP_\beta = U(\Lambda\Sigma_\beta)U^\top$. Further, since each $T_j$ is PSD (Lemma C.6), we have that $\Sigma_\beta$ and $P_\beta$ are PSD as well. We also have

$$\mathsf{Tr}\left[P_\beta\right] = \mathsf{Tr}\left[\Sigma_\beta\right] = \langle\beta, T\beta\rangle\,. \tag{34}$$

- Define the matrix $M_\omega \in \mathbb{C}^{d \times d}$ as

$$M_\omega = \left((1 - \exp(i\omega))I - \eta H\right)^{-1}. \tag{35}$$

Throughout, we assume that Assumption C.2 holds.

**Preliminary lemmas:** This lemma helps us move back and forth between the time-domain and frequency-domain representations. See Appendix C.7 for a proof.

**Lemma C.4.** *Consider $\beta \in \ell^2$ and its DTFT $B(\omega)$. If $0 < \eta < 1/\lambda_j$, we have*

$$\frac{1}{2} \langle \beta, T_j \beta \rangle \le \frac{\eta \lambda_j}{2\pi} \int_{-\pi}^{\pi} \frac{|B(\omega)|^2 \, \mathrm{d}\omega}{|1 - \eta \lambda_j - \exp(i\omega)|^2} \le \langle \beta, T_j \beta \rangle.$$

Setting $B(\omega) = 1$ and $\beta = (1, 0, \ldots)$ gives the next corollary.

**Corollary C.5.** *If $0 < \eta < 1/\lambda_j$, we have,*

$$\frac{1}{2} \le \frac{\eta \lambda_j}{2\pi} \int_{-\pi}^{\pi} \frac{\mathrm{d}\omega}{|1 - \eta \lambda_j - \exp(i\omega)|^2} \le 1.$$

**Lemma C.6.** *The operators $T_j$ defined in (32) and $T$ defined in (31) are both positive semi-definite for $\eta < 1/\max_{j \in [d]} \lambda_j$.*

*Proof.* Consider any $\beta \in \ell^2$ and its DTFT $B(\omega)$. We have from Lemma C.4 that

$$0 \le \int_{-\pi}^{\pi} \frac{|B(\omega)|^2 \, \mathrm{d}\omega}{|1 - \eta \lambda_j - \exp(i\omega)|^2} \le \frac{2\pi}{\eta \lambda_j} \langle \beta, T_j \beta \rangle,$$

or that $\langle \beta, T_j \beta \rangle \ge 0$. $\qquad \square$

## C.2 PROOF OF THE UPPER BOUND ON THE ASYMPTOTIC SUBOPTIMALITY

The key tool in the warm-up analysis of mean estimation (Appendix B) is the use of linear time-invariant (LTI) input-output systems to relate the output covariance to the input covariance using its transfer function (see Appendix F.1 for a summary). The Noisy-FTRL recursion is not trivial to characterize in this manner because the update (25) is not LTI. Instead, we decompose it into an infinite sequence of LTI systems and carefully analyze the error propagation.

This consists of the following steps:

Part 1: Decompose the Noisy-FTRL recursion into a sequence of LTI systems.
Part 2: Compute the transfer function of each LTI system.
Part 3: Compute the stationary covariance for each LTI system from the previous one.
Part 4: Combine the stationary covariances to get the stationary error of the original iterate.

### C.2.1 PART 1: DECOMPOSITION INTO A SEQUENCE OF LTI SYSTEMS

A challenge in analyzing the stationary error of Equation (30) in the frequency domain is that it is not an LTI system. Replacing $x_t \otimes x_t$ by $H$ in Equation (30) results in an LTI update; this system is quite similar to fixed design linear regression. However, this leads to an error in the general case, which satisfies a recursion of the same form as (30). We can repeat the same technique of replacing $x_t \otimes x_t$ by $H$ and repeat this process indefinitely. This proof technique has been used in (Aguech et al., 2000) to analyze stochastic tracking algorithms and (Bach & Moulines, 2013) to analyze iterate-averaged SGD for linear regression. We adopt this technique to analyze the stationary covariance of DP mechanisms with correlated noise.

We define sequences $(\boldsymbol{\theta}_t^{(r)})_{t=-\infty}^{\infty}$ and $(\boldsymbol{\delta}_t^{(r)})_{t=-\infty}^{\infty}$ for $r \geq 0$ as follows:

$$
\begin{aligned}
\boldsymbol{\theta}_{t+1}^{(0)} &= (\boldsymbol{I} - \eta \boldsymbol{H}) \boldsymbol{\theta}_t^{(0)} + \eta \xi_t \boldsymbol{x}_t - \eta \sum_{\tau=0}^{\infty} \beta_\tau \boldsymbol{w}_{t-k} \,, \\
\boldsymbol{\theta}_{t+1}^{(r)} &= (\boldsymbol{I} - \eta \boldsymbol{H}) \boldsymbol{\theta}_t^{(r)} + \eta (\boldsymbol{H} - \boldsymbol{x}_t \otimes \boldsymbol{x}_t) \boldsymbol{\theta}_t^{(r-1)} \text{ for } r > 0 \,, \\
\boldsymbol{\delta}_{t+1}^{(r)} &= (\boldsymbol{I} - \eta \boldsymbol{x}_t \otimes \boldsymbol{x}_t) \boldsymbol{\delta}_t^{(r)} + \eta (\boldsymbol{H} - \boldsymbol{x}_t \otimes \boldsymbol{x}_t) \boldsymbol{\theta}_t^{(r)} \,.
\end{aligned}
\tag{36}
$$

These recursions are assumed to start at $t = -\infty$ from $\boldsymbol{\theta}_t^{(0)} = \boldsymbol{\theta}_t'$, $\boldsymbol{\delta}_t^{(r)} = \mathbf{0}$ for $r \geq 0$ and $\boldsymbol{\theta}_t^{(r)} = \mathbf{0}$ for $r > 0$. These recursions are a decomposition of (30) as we define below.

**Property C.7.** *For each iteration $t$ and any integer $m \geq 0$, we have $\boldsymbol{\theta}_t' = \sum_{r=0}^{m} \boldsymbol{\theta}_t^{(r)} + \boldsymbol{\delta}_t^{(m)}$.*

*Proof.* We prove this by induction. The base case at $t = -\infty$ holds by definition. Assume that this is true for some integer $t$. Then, we have

$$
\begin{aligned}
\sum_{r=0}^{m} \boldsymbol{\theta}_{t+1}^{(r)} + \boldsymbol{\delta}_{t+1}^{(m)} &= (\boldsymbol{I} - \eta \boldsymbol{x}_t \otimes \boldsymbol{x}_t) \left( \sum_{r=0}^{m} \boldsymbol{\theta}_t^{(r)} + \boldsymbol{\delta}_t^{(m)} \right) + \eta \xi_t \boldsymbol{x}_t - \eta \sum_{\tau=0}^{\infty} \beta_\tau \boldsymbol{w}_{t-\tau} \\
&= (\boldsymbol{I} - \eta \boldsymbol{x}_t \otimes \boldsymbol{x}_t) \boldsymbol{\theta}_t' + \eta \xi_t \boldsymbol{x}_t - \eta \sum_{\tau=0}^{\infty} \beta_\tau \boldsymbol{w}_{t-\tau} = \boldsymbol{\theta}_{t+1}' \,.
\end{aligned}
$$

$\square$

The idea behind the proof is to show that $\mathbb{E}\left[ \boldsymbol{\delta}_0^{(m)} \otimes \boldsymbol{\delta}_0^{(m)} \right] \to \mathbf{0}$ as $m \to \infty$. Then, we can use the triangle inequality to bound

$$
\|\boldsymbol{\theta}_t'\| \leq \sum_{r=0}^{\infty} \left\| \boldsymbol{\theta}_t^{(r)} \right\| \,,
$$

where the stationary error of the right side can be obtained from analyzing the LTI systems defined in (36).

### C.2.2 PART 2: CHARACTERIZE THE TRANSFER FUNCTION OF EACH LTI SYSTEM

There are two LTI systems. First, $\boldsymbol{\theta}_t^{(r)}$ for $r > 0$ is an LTI system

$$
\boldsymbol{z}_{t+1} = (\boldsymbol{I} - \eta \boldsymbol{H}) \boldsymbol{z}_t + \eta \boldsymbol{u}_t
\tag{37}
$$

with input $\boldsymbol{u}_t \in \mathbb{R}^d$ and output $\boldsymbol{z}_t \in \mathbb{R}^d$. Second, $\boldsymbol{\theta}_t^{(0)}$ satisfies satisfies an LTI system

$$
\boldsymbol{z}_{t+1} = (\boldsymbol{I} - \eta \boldsymbol{H}) \boldsymbol{z}_t + \eta \boldsymbol{u}_t - \eta \sum_{\tau=0}^{\infty} \beta_t \boldsymbol{w}_{t-\tau}
\tag{38}
$$

with inputs $(\boldsymbol{u}_t, \boldsymbol{w}_t) \in \mathbb{R}^d \times \mathbb{R}^d$ and output $\boldsymbol{z}_t \in \mathbb{R}^d$ where the weights $\boldsymbol{\beta} \in \ell^2$ are assumed to be given.

We now characterize the transfer functions of these LTI systems; see Appendix F.1 for a review.

**Property C.8.** *The LTI system (37) is $\boldsymbol{G}(\omega) = -\eta \boldsymbol{M}_\omega \in \mathbb{C}^{d \times d}$, where $\boldsymbol{M}_\omega$ is defined in Equation (35). Moreover, this system is asymptotically stable as long as $\mathbf{0} \prec \eta \boldsymbol{H} \prec \boldsymbol{I}$.*

*Proof.* Let $\boldsymbol{U}(\omega) \in \mathbb{C}^d$ and $\boldsymbol{Z}(\omega) \in \mathbb{C}^d$ be the Fourier transforms of $\boldsymbol{u}_t$ and $\boldsymbol{z}_t$ respectively. The transfer function must hold for any input-output sequences, so we can choose some sequences and solve for the

transfer functions. It is convenient to consider the delta spike on a standard basis (up to scaling), i.e., $U = 2\pi\delta_\omega e_j$, where $\delta_\omega$ is the Dirac delta at $\omega$, and $e_j$ is the $j^{\text{th}}$ standard basis vector in $\mathbb{R}^d$. This gives $Z = 2\pi g_j \delta_\omega$ where $g_j(\cdot)$ is the $j^{\text{th}}$ column of $G(\cdot)$.

To move back to the time domain, we take an inverse Fourier transform to get $u_t = \exp(i\omega t)e_j$ and $z_t = g_j(\omega)\exp(i\omega t)$. Plugging this into the update (37) gives and solving for $g_j(\omega)$ gives $g_j(\omega) = -\eta M_\omega e_j$. Stacking these into a matrix gives the expression.

If $u_t \equiv 0$ for all $t$, then $\|z_{t+1}\|_2 \leq \|I - \eta H\|_2 \|z_t\|_2 < \|z_t\|_2$ since $\|I - \eta H\|_2 < 1$. Hence, $\|z_t\|_2 \to 0$, giving the asymptotic stability of the system. $\qquad\square$

**Property C.9.** *The transfer function of the LTI system* (38) *is*

$$\widetilde{G}(\omega) = [G(\omega) \quad G'(\omega)] \in \mathbb{C}^{d \times 2d}$$

*where $G(\omega) = -\eta M_\omega$ and $G'(\omega) = \eta B(\omega)M_\omega$ with $B(\omega)$ as the DTFT of $\beta$. Moreover, this system is asymptotically stable as long as $0 \prec \eta H \prec I$.*

*Proof.* The expression for $G(\omega)$ is the same as in Property C.8. To find $G'$, we set the Fourier transforms $U \equiv 0$, $W = 2\pi\delta_\omega e_j$ so that $Z = 2\pi\delta_\omega g'_j$, where $g'_j(\cdot)$ is the $j^{\text{th}}$ column of $G'(\cdot)$.

An inverse Fourier transform gives the time domain versions $w_t = \exp(i\omega t)$, $u_t \equiv 0$, $z_t = \exp(i\omega t)g'_j(\omega)$. Plugging these into (38) and plugging in the definition of $B(\omega)$ gives the expression for the transfer function. Its asymptotic stability holds similar to Property C.8. $\qquad\square$

### C.2.3  PART 3: COMPUTE THE STATIONARY COVARIANCE OF EACH LTI SYSTEM

The stationary covariance of an LTI system driven by white noise can be concisely described in the frequency domain. A sequence $(u_t)$ is said to be a white noise process if it is mean zero and $\mathbb{E}[u_t u_\tau] = 0$ for $t \neq \tau$. This is true for both $\theta_t^{(0)}$ as well $\theta_t^{(r)}$ for $r > 0$. Since we care about the stationary distribution and we start at $t = -\infty$, we have reached the steady state at $t = 0$. So, we compute $\mathbb{E}[\theta_0^{(r)} \otimes \theta_0^{(r)}]$.

**Stationary covariance of the base recursion:** We first start with $\theta_t^{(0)}$.

**Proposition C.10.** *We have that $\mathbb{E}\left[\theta_t^{(0)} \otimes \theta_t^{(0)}\right]$ is equal for all $t > -\infty$ and is bounded as*

$$\mathbb{E}\left[\theta_t^{(0)} \otimes \theta_t^{(0)}\right] \preceq \eta\sigma_{\text{sgd}}^2 I + \eta\sigma^2\, H^{-1/2} P_\beta H^{-1/2}\,,$$

*where $P_\beta$ is defined in Equation (33) and we denote $\sigma^2 = G^2\gamma_\infty^2(\beta)/(2\rho)$.*

*Proof.* The input $(\xi_t x_t, w_t)$ forms a white noise sequence, since for $t \neq \tau$, we have $\mathbb{E}[\xi_t x_t \xi_\tau x_\tau] = \mathbb{E}[\xi_t x_t]\,\mathbb{E}[\xi_\tau x_\tau] = 0$ (since $\xi_t x_t$ for each $t$ is i.i.d.) and $\mathbb{E}[w_t w_\tau] = 0$. The covariance of the input is

$$\mathbb{E}[(\xi_t x_t, w_t) \otimes (\xi_t x_t, w_t)] = \begin{bmatrix} \mathbb{E}[\xi_t^2 x_t x_t] & 0 \\ 0 & \mathbb{E}[w_t \otimes w_t] \end{bmatrix} = \mathbb{E}[(\xi_\tau x_\tau, w_\tau) \otimes (\xi_\tau x_\tau, w_\tau)]$$

for all $t, \tau$. This is further bounded by Assumption (A1) as

$$\mathbb{E}[(\xi_t x_t, w_t) \otimes (\xi_t x_t, w_t)] \preceq \begin{bmatrix} \sigma_{\text{sgd}}^2 H & 0 \\ 0 & \sigma^2 I \end{bmatrix}$$

The output covariance of the asymptotically stable LTI system (38) can be given in terms of the transfer function $\widetilde{\boldsymbol{G}}(\omega) = [\boldsymbol{G}(\omega) \quad \boldsymbol{G}'(\omega)]$ characterized in Property C.9 using Theorem F.2. This gives that $\mathbb{E}\left[\boldsymbol{\theta}_t^{(0)} \otimes \boldsymbol{\theta}_t^{(0)}\right]$ is equal for each $t > -\infty$ and is bounded as

$$\mathbb{E}\left[\boldsymbol{\theta}_t^{(0)} \otimes \boldsymbol{\theta}_t^{(0)}\right] \preceq \frac{1}{2\pi}\int_{-\pi}^{\pi}\left(\eta^2\sigma_{\mathsf{sgd}}^2\boldsymbol{M}_\omega\boldsymbol{H}\boldsymbol{M}_\omega^* + \eta^2\sigma^2|B(\omega)|^2\,\boldsymbol{M}_\omega\boldsymbol{M}_\omega^*\right)\,\mathrm{d}\omega\,. \tag{39}$$

With the eigenvalue decomposition $\boldsymbol{H} = \boldsymbol{U}\boldsymbol{\Lambda}\boldsymbol{U}^\top$, we get $\boldsymbol{M}_\omega = \boldsymbol{U}\big((1-\exp(i\omega))\boldsymbol{I} - \eta\boldsymbol{\Lambda}\big)^{-1}\boldsymbol{U}^\top$. This gives

$$\boldsymbol{M}_\omega\boldsymbol{H}\boldsymbol{M}_\omega^* = \boldsymbol{U}\,\mathsf{diag}\left(\left(\lambda_j/|1-\exp(i\omega)-\eta\lambda_j|^2\right)_{j=1}^d\right)\boldsymbol{U}^\top\,.$$

We invoke Corollary C.5 to say

$$\int_{-\pi}^{\pi}\boldsymbol{M}_\omega\boldsymbol{H}\boldsymbol{M}_\omega^*\mathrm{d}\omega = \boldsymbol{U}\,\mathsf{diag}\left(\left(\int_{-\pi}^{\pi}\mathrm{d}\omega\,\lambda_j/|1-\exp(i\omega)-\eta\lambda_j|^2\right)_{j=1}^d\right)\boldsymbol{U}^\top$$

$$\preceq \boldsymbol{U}\,\mathsf{diag}\left(\left(2\pi/\eta\right)_{j=1}^d\right)\boldsymbol{U}^\top = \frac{2\pi}{\eta}\boldsymbol{I}\,. \tag{40}$$

Similarly, we invoke Lemma C.4 to compute

$$\int_{-\pi}^{\pi}|B(\omega)|^2\boldsymbol{M}_\omega\boldsymbol{M}_\omega^*\mathrm{d}\omega = \boldsymbol{U}\,\mathsf{diag}\left(\left(\int_{-\pi}^{\pi}\mathrm{d}\omega\,|B(\omega)|^2/|1-\exp(i\omega)-\eta\lambda_j|^2\right)_{j=1}^d\right)\boldsymbol{U}^\top$$

$$\preceq \boldsymbol{U}\,\mathsf{diag}\left(\left(2\pi\langle\boldsymbol{\beta},\boldsymbol{T}_j\boldsymbol{\beta}\rangle/(\eta\lambda_j)\right)_{j=1}^d\right)\boldsymbol{U}^\top$$

$$= \frac{2\pi}{\eta}\boldsymbol{U}\boldsymbol{\Lambda}^{-1/2}\boldsymbol{\Sigma}_{\boldsymbol{\beta}}\boldsymbol{\Lambda}^{-1/2}\boldsymbol{U}^\top = \frac{2\pi}{\eta}\boldsymbol{H}^{-1/2}\boldsymbol{P}_{\boldsymbol{\beta}}\boldsymbol{H}^{-1/2}\,, \tag{41}$$

where $\boldsymbol{\Sigma}_{\boldsymbol{\beta}}$ and $\boldsymbol{P}_{\boldsymbol{\beta}}$ are defined in (33). Plugging in (40) and (40) into (39) completes the proof of the upper bound. $\qquad\square$

**Stationary covariance of the higher-order recursion:** Next, we turn to $\boldsymbol{\theta}_t^{(r)}$.

**Proposition C.11.** *For any $r \geq 1$, we have*

$$\mathbb{E}\left[\boldsymbol{\theta}_0^{(r)} \otimes \boldsymbol{\theta}_0^{(r)}\right] \preceq \eta\left(\eta R^2\right)^r\left(\sigma_{\mathsf{sgd}}^2 + \frac{C_{\mathsf{kurt}}\sigma^2}{R^2}\langle\boldsymbol{\beta},\boldsymbol{T}\boldsymbol{\beta}\rangle\right)\,.$$

*Proof.* Follows from combining Proposition C.10 with the more general Lemma C.12 below. $\qquad\square$

**Lemma C.12.** *For some $r \geq 1$, suppose that $\mathbb{E}\left[\boldsymbol{\theta}_t^{(r-1)} \otimes \boldsymbol{\theta}_t^{(r-1)}\right]$ is equal for each $t$ and is bounded as $\mathbb{E}\left[\boldsymbol{\theta}_t^{(r-1)} \otimes \boldsymbol{\theta}_t^{(r-1)}\right] \preceq a\boldsymbol{I} + b\boldsymbol{H}^{-1/2}\boldsymbol{P}_{\boldsymbol{\beta}}\boldsymbol{H}^{-1/2}$ for some scalars $a, b \geq 0$. Then, we have the following.*

*(a) We have that $\boldsymbol{\zeta}_t^{(r)} := (\boldsymbol{H} - \boldsymbol{x}_t \otimes \boldsymbol{x}_t)\,\boldsymbol{\theta}_t^{(r-1)}$ is a white-noise process with*

$$\mathbb{E}\left[\boldsymbol{\zeta}_t^{(r)} \otimes \boldsymbol{\zeta}_t^{(r)}\right] \preceq \left(aR^2 + bC_{\mathsf{kurt}}\langle\boldsymbol{\beta},\boldsymbol{T}\boldsymbol{\beta}\rangle\right)\boldsymbol{H}\,.$$

*(b) We have that $\mathbb{E}\left[\boldsymbol{\theta}_t^{(r)} \otimes \boldsymbol{\theta}_t^{(r)}\right]$ is equal for each $t$ and is bounded as*

$$\mathbb{E}\left[\boldsymbol{\theta}_t^{(r)} \otimes \boldsymbol{\theta}_t^{(r)}\right] \preceq \eta\left(aR^2 + bC_{\mathsf{kurt}}\langle\boldsymbol{\beta},\boldsymbol{T}\boldsymbol{\beta}\rangle\right)\boldsymbol{I}\,.$$

*Proof.* Note that $\mathbb{E}\left[\boldsymbol{\zeta}_t^{(r)} \otimes \boldsymbol{\zeta}_\tau^{(r)}\right] = \mathbf{0}$ for $t \neq \tau$ since $\boldsymbol{x}_t$ is independent of $\boldsymbol{x}_\tau$ and $\mathbb{E}[\boldsymbol{x}_t \otimes \boldsymbol{x}_t] = \boldsymbol{H}$. Since $\boldsymbol{x}_t$ is independent of $\boldsymbol{\theta}_t^{(r-1)}$, we get from the tower rule of expectations that

$$\mathbb{E}\left[\boldsymbol{\zeta}_t^{(r)} \otimes \boldsymbol{\zeta}_t^{(r)}\right] = \mathbb{E}\left[(\boldsymbol{H} - \boldsymbol{x}_t \otimes \boldsymbol{x}_t)\left(\boldsymbol{\theta}_t^{(r-1)} \otimes \boldsymbol{\theta}_t^{(r-1)}\right)(\boldsymbol{H} - \boldsymbol{x}_t \otimes \boldsymbol{x}_t)\right]$$
$$= \mathbb{E}\left[(\boldsymbol{H} - \boldsymbol{x}_t \otimes \boldsymbol{x}_t)\,\mathbb{E}\left[\boldsymbol{\theta}_t^{(r-1)} \otimes \boldsymbol{\theta}_t^{(r-1)}\right](\boldsymbol{H} - \boldsymbol{x}_t \otimes \boldsymbol{x}_t)\right],$$

or that $(\boldsymbol{\zeta}_t^{(r)})$ is a white noise process. Its covariance can further be bounded as

$$\mathbb{E}\left[\boldsymbol{\zeta}_t^{(r)} \otimes \boldsymbol{\zeta}_t^{(r)}\right] \preceq \mathbb{E}\left[(\boldsymbol{H} - \boldsymbol{x}_t \otimes \boldsymbol{x}_t)\left(a\boldsymbol{I} + b\boldsymbol{H}^{-1/2}\boldsymbol{P_\beta}\boldsymbol{H}^{-1/2}\right)(\boldsymbol{H} - \boldsymbol{x}_t \otimes \boldsymbol{x}_t)\right]$$
$$\preceq a\,\mathbb{E}\left[\|\boldsymbol{x}_t\|_2^2\,(\boldsymbol{x}_t \otimes \boldsymbol{x}_t)\right] + b\,\mathbb{E}\left[(\boldsymbol{x}_t \otimes \boldsymbol{x}_t)\boldsymbol{H}^{-1/2}\boldsymbol{P_\beta}\boldsymbol{H}^{-1/2}(\boldsymbol{x}_t \otimes \boldsymbol{x}_t)\right)$$
$$\preceq aR^2\boldsymbol{H} + bC_{\mathsf{kurt}}\,\mathsf{Tr}\left[\boldsymbol{P_\beta}\right]\boldsymbol{H},$$

where the last inequality followed from Assumption (A3). Further, note that $\mathsf{Tr}\left[\boldsymbol{P_\beta}\right] = \langle\boldsymbol{\beta}, \boldsymbol{T}\boldsymbol{\beta}\rangle$ from (34).

The output covariance of the asymptotically stable LTI system (37) can be given in terms of the transfer function $\boldsymbol{G}(\omega) = -\eta\boldsymbol{M}_\omega$ using Theorem F.2 as

$$\mathbb{E}\left[\boldsymbol{\theta}_t^{(r)} \otimes \boldsymbol{\theta}_t^{(r)}\right] \preceq \frac{\eta^2\left(aR^2 + bC_{\mathsf{kurt}}\langle\boldsymbol{\beta}, \boldsymbol{T}\boldsymbol{\beta}\rangle\right)}{2\pi}\int_{-\pi}^{\pi}\boldsymbol{M}_\omega\boldsymbol{H}\boldsymbol{M}_\omega^*\,\mathrm{d}\omega \overset{(40)}{\preceq} \eta\left(aR^2 + bC_{\mathsf{kurt}}\langle\boldsymbol{\beta}, \boldsymbol{T}\boldsymbol{\beta}\rangle\right)\boldsymbol{I}.$$

$\square$

**Remainder Term:** It remains to show that the remainder term $\boldsymbol{\delta}_t$ can be neglected by taking $m \to \infty$.

**Proposition C.13.** *We have* $\lim_{m\to\infty}\mathbb{E}\left[\boldsymbol{\delta}_t^{(m)} \otimes \boldsymbol{\delta}_t^{(m)}\right] = \mathbf{0}$.

*Proof.* Let $\boldsymbol{\zeta}_t^{(m+1)} := (\boldsymbol{H} - \boldsymbol{x}_t \otimes \boldsymbol{x}_t)\boldsymbol{\theta}_t^{(m)}$. By Lemma C.12 and Proposition C.11, we have $\boldsymbol{\zeta}_t$ is a white-noise process with

$$\mathbb{E}\left[\boldsymbol{\zeta}_t^{(m+1)} \otimes \boldsymbol{\zeta}_t^{(m+1)}\right] \preceq (\eta R^2)^{m+1}\left(\sigma_{\mathsf{sgd}}^2 + \frac{C_{\mathsf{kurt}}\sigma^2}{R^2}\langle\boldsymbol{\beta}, \boldsymbol{T}\boldsymbol{\beta}\rangle\right)\boldsymbol{H} \to \mathbf{0}$$

as $m \to \infty$ since $\eta < 1/R^2$. Note that the update for $\boldsymbol{\delta}_t^{(m)}$ exactly matches that of SGD (without added DP noise), and the noise covariance is $\mathbf{0}$. The statement of this result is equivalent to showing that the stationary covariance of SGD with zero residuals is zero. This observation is formalized in Lemma 4 of (Jain et al., 2017a) (see also Theorem F.3 of Appendix F), which gives for any $t$ that

$$\mathbf{0} \preceq \mathbb{E}[\boldsymbol{\delta}_t^{(m)} \otimes \boldsymbol{\delta}_t^{(m)}] \preceq \frac{\eta}{1 - \eta R^2}\left[(\eta R^2)^{m+1}\left(\sigma_{\mathsf{sgd}}^2 + \frac{C_{\mathsf{kurt}}\sigma^2}{R^2}\langle\boldsymbol{\beta}, \boldsymbol{T}\boldsymbol{\beta}\rangle\right)\right]\boldsymbol{I} \to \mathbf{0}$$

as $m \to \infty$. $\square$

### C.2.4 PART 4: COMBINING THE ERRORS

**Time-domain description:** We now state and prove a time-domain description of the upper bound of Equation (20).

**Theorem C.14.** *Suppose Assumption C.2 holds. Consider the sequence $(\boldsymbol{\theta}_t)_{t=-\infty}^{\infty}$ produced by the Noisy-FTRL update in Equation (25) with some given weights $\boldsymbol{\beta} \in \ell^2$ and noise variance $\boldsymbol{w}_t \sim \mathcal{N}(\boldsymbol{0}, G^2 \gamma_\infty^2(\boldsymbol{\beta})/(2\rho)\boldsymbol{I})$. If the learning rate satisfies $\eta < 1/R^2$, we have*

$$F_\infty(\boldsymbol{\beta}) \leq \left(1 + \left(1 - \sqrt{\eta R^2}\right)^{-2}\right)\eta R^2 \sigma_{\mathsf{sgd}}^2 + \left(1 + C_{\mathsf{kurt}}\left(1 - \sqrt{\eta R^2}\right)^{-2}\right)\frac{\eta G^2 \gamma_\infty^2(\boldsymbol{\beta})}{2\rho}\langle \boldsymbol{\beta}, \boldsymbol{T}\boldsymbol{\beta}\rangle.$$

*Proof.* We use shorthand $\sigma^2 = \frac{G^2 \gamma_\infty^2(\boldsymbol{\beta})}{2\rho}$. First, note that $\eta < 1/R^2$ also implies that $\eta\lambda_j < 1$ for each eigenvalue $\lambda_j$ of $\boldsymbol{H}$. The right side is well-defined since Lemma F.17 gives

$$|\langle \boldsymbol{\beta}, \boldsymbol{T}\boldsymbol{\beta}\rangle| \leq \sum_{j=1}^{d}\left|\sum_{t=0}^{\infty}\sum_{\tau=0}^{\infty}\beta_t\beta_\tau(1 - \eta\lambda_j)^{|t-\tau|}\right| \leq \|\boldsymbol{\beta}\|_2^2 \sum_{j=1}^{d}\frac{2}{\eta\lambda_j} < \infty \tag{42}$$

for $\beta \in \ell^2$. Next, using Proposition C.10, $\mathsf{Tr}\,[\boldsymbol{H}] \leq R^2$, and $\mathsf{Tr}\,[\boldsymbol{P}_\beta] = \langle \boldsymbol{\beta}, \boldsymbol{T}\boldsymbol{\beta}\rangle$, we get

$$\mathbb{E}\left\|\boldsymbol{\theta}_0^{(0)}\right\|_{\boldsymbol{H}}^2 = \mathsf{Tr}\left[\boldsymbol{H}\mathbb{E}\left[\boldsymbol{\theta}_0^{(0)} \otimes \boldsymbol{\theta}_0^{(0)}\right]\right] \leq \eta R^2 \sigma_{\mathsf{sgd}}^2 + \eta\sigma^2\langle \boldsymbol{\beta}, \boldsymbol{T}\boldsymbol{\beta}\rangle. \tag{43}$$

Similarly, using Proposition C.11, we get for $r \geq 1$ that

$$\mathbb{E}\left\|\boldsymbol{\theta}_0^{(r)}\right\|_{\boldsymbol{H}}^2 \leq (\eta R^2)^{r+1}\left(\sigma_{\mathsf{sgd}}^2 + \frac{C_{\mathsf{kurt}}\sigma^2}{R^2}\langle \boldsymbol{\beta}, \boldsymbol{T}\boldsymbol{\beta}\rangle\right).$$

We can ignore the remainder term since $\mathbb{E}\left\|\boldsymbol{\delta}_t^{(m)}\right\|_{\boldsymbol{H}}^2 \to 0$ as $m \to \infty$, from Proposition C.13. Thus, we get using Property C.7 and the triangle inequality on the norm $\boldsymbol{u} \mapsto \sqrt{\mathbb{E}\langle \boldsymbol{u}, \boldsymbol{H}\boldsymbol{u}\rangle}$ of a random vector $\boldsymbol{u}$ to get

$$\sqrt{\mathbb{E}\|\boldsymbol{\theta}_0'\|_{\boldsymbol{H}}^2} \leq \sum_{r=0}^{\infty}\sqrt{\mathbb{E}\left\|\boldsymbol{\theta}_0^{(r)}\right\|_{\boldsymbol{H}}^2}.$$

To complete the proof, we plug in Equations (42) and (43) and sum up the infinite series. We simplify the result using $\|\boldsymbol{x} + \boldsymbol{y}\|_{\boldsymbol{H}}^2 \leq 2\|\boldsymbol{x}\|_{\boldsymbol{H}}^2 + 2\|\boldsymbol{y}\|_{\boldsymbol{H}}^2$ and use $F(\boldsymbol{\theta}) - F(\boldsymbol{\theta}_\star) = (1/2)\|\boldsymbol{\theta} - \boldsymbol{\theta}_\star\|_{\boldsymbol{H}}^2$. □

**Frequency-domain description:** We now state and prove the frequency domain description of the upper bound (20).

**Theorem C.15.** *Consider the setting of Theorem C.14. If $B \in L^2$, i.e., $\int_{-\pi}^{\pi}|B(\omega)|^2\,\mathrm{d}\omega < \infty$, we have*

$$\begin{aligned} F_\infty(B) \leq\ & \left(1 + \left(1 - \sqrt{\eta R^2}\right)^{-2}\right)\eta R^2 \sigma_{\mathsf{sgd}}^2 \\ & + \left(1 + C_{\mathsf{kurt}}\left(1 - \sqrt{\eta R^2}\right)^{-2}\right)\frac{\eta^2 G^2 \gamma_\infty^2(B)}{2\pi\rho}\int_{-\pi}^{\pi}|B(\omega)|^2\,h(\omega)\,\mathrm{d}\omega. \end{aligned}$$

*Proof.* We again use the shorthand $\sigma^2 = \frac{G^2 \gamma_\infty^2(\boldsymbol{\beta})}{2\rho}$. First note that

$$h(\omega) \leq \sum_{j=1}^{d}\frac{\lambda_j}{1 + (1 - \eta\lambda_j)^2 - 2(1 - \eta\lambda_j)} = \sum_{j=1}^{d}\frac{1}{\eta^2\lambda_j} = \frac{\mathsf{Tr}\,[\boldsymbol{H}^{-1}]}{\eta^2}.$$

Thus, the right side is well-defined since

$$\int_{-\pi}^{\pi} |B(\omega)|^2 \, h(\omega) \mathrm{d}\omega \leq \frac{\mathsf{Tr}\left[\boldsymbol{H}^{-1}\right]}{\eta^2} \int_{-\pi}^{\pi} |B(\omega)|^2 \, \mathrm{d}\omega < \infty$$

by assumption. We use Lemma C.4 to get

$$\langle \boldsymbol{\beta}, \boldsymbol{T}\boldsymbol{\beta} \rangle = \sum_{j=1}^{d} \langle \boldsymbol{\beta}, \boldsymbol{T}_j\boldsymbol{\beta} \rangle \leq \sum_{j=1}^{d} \frac{\eta\lambda_j}{\pi} \int_{-\pi}^{\pi} \frac{|B(\omega)|^2 \mathrm{d}\omega}{|1 - \exp(i\omega) - \eta\lambda_j|^2} = \frac{\eta}{\pi} \int_{-\pi}^{\pi} |B(\omega)|^2 \, h(\omega) \, \mathrm{d}\omega \,.$$

$\square$

**Remark C.16** (Contribution per eigendirection). *The expression of Theorem C.15 contains a sum over the eigenvalues $\lambda_1, \ldots, \lambda_d$ of the Hessian matrix $\boldsymbol{H}$ through the function $h(\omega)$, defined in Eq. (21). Thus, the contribution of eigenvalue $\lambda_j$ to the error is proportional to (ignoring problem-dependent constants)*

$$Err_j := \int_{-\pi}^{\pi} \frac{\lambda_j \, |B(\omega)|^2 \, \mathrm{d}\omega}{|1 - \exp(i\omega) - \eta\lambda_j|^2} \,. \tag{44}$$

*For Noisy-SGD, we have that $B(\omega) = 1$, and the error $Err_j = \Theta(1)$ evaluates to an absolute constant (details in Corollary C.5). In other words, each eigendirection contributes a constant amount to the error, leading to a $O(d)$ dimension dependence in the asymptotic error.*

*On the other hand, as we discuss further in Remark C.23 (Appendix C.4), we have $Err_j \leq \widetilde{O}(\lambda_j)$ for $\nu$-Noisy-FTRL. Thus, the contribution of an eigendirection reduces proportional to the eigenvalues, leading to an effective dimension dependence for $\nu$-Noisy-FTRL.*

*These quantitative results can be connected intuitively to the signal in the gradients. Let $\lambda_1, \ldots, \lambda_d$ be the eigenvalues of $\boldsymbol{H}$ with $\lambda_1 = 1$. The negative gradient at each step pushes the iterates back towards the minimizer, thus mitigating the effect of the past noise. However, the signal in the gradient along tail eigendirections is small, making it ineffective in such directions. This leads to $Err_j = \Theta(1)$ for Noisy-SGD, which can be much larger than $\lambda_j$. On the other hand, the anti-correlations of $\nu$-DP-FTRL "subtract out" the previous noise, leading to $Err_j \propto \lambda_j$ for $\nu$-Noisy-FTRL, i.e., an improved effective dimension dependence.*

## C.3 PROOFS OF LOWER BOUNDS ON THE ASYMPTOTIC SUBOPTIMALITY

We now state and prove the lower bound part of (20) on the asymptotic suboptimality.

**Assumption C.17.** *In addition to Assumption C.2, the data distribution $\mathbb{P}_{\mathsf{data}}$ satisfies the following:*

*(A2') Worst-Case Residuals: For $(\boldsymbol{x}, y) \sim \mathbb{P}_{\mathsf{data}}$, the residual $\xi := y - \langle \boldsymbol{\theta}_\star, \boldsymbol{x} \rangle$ has variance $\mathbb{E}[\xi^2] = \sigma_{\mathsf{sgd}}^2$.*

Note that the variance of $\xi^2$ holds with equality under Assumption C.17.

**Theorem C.18.** *Suppose Assumption C.17 holds. Consider the sequence $(\boldsymbol{\theta}_t)_{t=-\infty}^{\infty}$ produced by the Noisy-FTRL update in Equation (25) with some given weights $\boldsymbol{\beta} \in \ell^1$. If the learning rate satisfies $\eta < 1/R^2$, we have*

$$F_\infty(\boldsymbol{\beta}) \geq \frac{\eta\sigma_{\mathsf{sgd}}^2}{2} \mathsf{Tr}\left[\boldsymbol{H}\right] + \frac{\eta^2 G^2 \gamma_\infty^2(B)}{4\pi\rho} \int_{-\pi}^{\pi} |B(\omega)|^2 \, h(\omega) \, \mathrm{d}\omega \geq \frac{\eta\sigma_{\mathsf{sgd}}^2}{2} \mathsf{Tr}\left[\boldsymbol{H}\right] + \frac{\eta G^2 \gamma_\infty^2(\boldsymbol{\beta})}{4\rho} \langle \boldsymbol{\beta}, \boldsymbol{T}\boldsymbol{\beta} \rangle \,,$$

*where $h(\omega)$ is defined in (21) and $\boldsymbol{T}$ is defined in (31). Furthermore, the minimal stationary error over all choices of $\boldsymbol{\beta}$ is bounded as*

$$\inf_{\boldsymbol{\beta}} \, F_\infty(\boldsymbol{\beta}) \geq \frac{1}{4} \left( 2\eta\sigma_{\mathsf{sgd}}^2 + \frac{\eta^2 G^2}{2\rho} \right) \mathsf{Tr}\left[\boldsymbol{H}\right]$$

*where the infimum is attained by $\boldsymbol{\beta}_\star$ whose DTFT $B_\star$ verifies $|B_\star(\omega)|^2 = 1/\sqrt{h(\omega)}$.*

Note that we assume $\boldsymbol{\beta} \in \ell^1$, i.e., $\|\boldsymbol{\beta}\|_1 = \sum_{\tau=0}^{\infty} |\beta_\tau| < \infty$ for technical reasons. This implies that $\boldsymbol{\beta} \in \ell^2$, which we assumed for the upper bounds.

The key idea behind the proof is that the variance of $\boldsymbol{\theta}'_t$ is no smaller than that of an LTI system with $\boldsymbol{x}_t \otimes \boldsymbol{x}_t$ replaced by its expectation $\boldsymbol{H}$. We can quantify this latter covariance with equality under Assumption C.17. We set up some notation and develop some preliminary results before proving this theorem.

Formally, consider the sequences $(\boldsymbol{\theta}_t^{(0)})_{t=-\infty}^{\infty}$ and $(\boldsymbol{\delta}_t^{(0)})_{t=-\infty}^{\infty}$ as defined in (36) (cf. Appendix C.2.1). They start at $t = -\infty$ from $\boldsymbol{\theta}_t^{(0)} = \boldsymbol{\theta}'_t$ and $\boldsymbol{\delta}_t^{(0)} = \boldsymbol{0}$. By Property C.7, we these satisfy $\boldsymbol{\theta}'_t = \boldsymbol{\theta}_t^{(0)} + \boldsymbol{\delta}_t^{(0)}$.

We use a technical result that $\boldsymbol{\theta}_t^{(0)}$ and $\boldsymbol{\delta}_t$ are uncorrelated. It is proved at the end of this section.

**Proposition C.19.** *Consider the setting of Theorem C.18. We have for all $t$ that*

$$\mathbb{E}\left[\boldsymbol{\theta}_t^{(0)} \otimes \boldsymbol{\delta}_t^{(0)}\right] = \boldsymbol{0}\,.$$

We now give the proof of Theorem C.18.

*Proof of Theorem C.18.* We use shorthand $\sigma^2 = \frac{G^2 \gamma_\infty^2(\boldsymbol{\beta})}{2\rho}$. Since $\boldsymbol{\theta}'_t = \boldsymbol{\theta}_t^{(0)} + \boldsymbol{\delta}_t^{(0)}$, we have

$$\mathbb{E}\left[\boldsymbol{\theta}'_t \otimes \boldsymbol{\theta}'_t\right] = \mathbb{E}\left[\boldsymbol{\theta}_t^{(0)} \otimes \boldsymbol{\theta}_t^{(0)}\right] + \mathbb{E}\left[\boldsymbol{\delta}_t^{(0)} \otimes \boldsymbol{\delta}_t^{(0)}\right] \succeq \mathbb{E}\left[\boldsymbol{\theta}_t^{(0)} \otimes \boldsymbol{\theta}_t^{(0)}\right] \tag{45}$$

where the cross terms disappear from Proposition C.19 for the first equality. We can get an expression for this term by following the proof of Proposition C.10: under Assumption C.17, we have that Equation (39) holds with equality. Thus, we get for all $t > -\infty$ that

$$F_\infty(B) = \mathsf{Tr}\left[\boldsymbol{H}\,\mathbb{E}\left[\boldsymbol{\theta}'_t \otimes \boldsymbol{\theta}'_t\right]\right] \succeq \mathsf{Tr}\left[\boldsymbol{H}\,\mathbb{E}\left[\boldsymbol{\theta}_t^{(0)} \otimes \boldsymbol{\theta}_t^{(0)}\right]\right]$$

$$= \frac{1}{2\pi}\int_{-\pi}^{\pi}\left(\eta^2 \sigma_{\mathsf{sgd}}^2 \mathsf{Tr}\left[\boldsymbol{H}^{1/2}\boldsymbol{M}_\omega \boldsymbol{H}\boldsymbol{M}_\omega^* \boldsymbol{H}^{1/2}\right] + \eta^2 \sigma^2 |B(\omega)|^2 \mathsf{Tr}\left[\boldsymbol{H}^{1/2}\boldsymbol{M}_\omega \boldsymbol{M}_\omega^* \boldsymbol{H}^{1/2}\right]\right)\mathrm{d}\omega\,. \tag{46}$$

We invoke Corollary C.5 to obtain

$$\int_{-\pi}^{\pi}\mathsf{Tr}\left[\boldsymbol{H}^{1/2}\boldsymbol{M}_\omega \boldsymbol{H}\boldsymbol{M}_\omega^* \boldsymbol{H}^{1/2}\right]\mathrm{d}\omega = \sum_{j=1}^{d}\lambda_j^2 \int_{-\pi}^{\pi}\frac{\mathrm{d}\omega}{|1 - \exp(i\omega) - \eta\lambda_j|^2}$$

$$\geq \sum_{j=1}^{d}\frac{\pi\lambda_j}{\eta} = \frac{\pi}{\eta}\mathsf{Tr}\left[\boldsymbol{H}\right]\,.$$

Similarly, we invoke Lemma C.4 to compute

$$\int_{-\pi}^{\pi}|B(\omega)|^2 \mathsf{Tr}\left[\boldsymbol{H}^{1/2}\boldsymbol{M}_\omega \boldsymbol{M}_\omega^* \boldsymbol{H}^{1/2}\right]\mathrm{d}\omega = \int_{-\pi}^{\pi}\left(\sum_{j=1}^{d}|B(\omega)|^2 \frac{\lambda_j}{|1 - \exp(i\omega) - \eta\lambda_j|^2}\right)\mathrm{d}\omega$$

$$= \int_{-\pi}^{\pi}|B(\omega)|^2\,h(\omega)\,\mathrm{d}\omega \geq \frac{\pi}{\eta}\langle\boldsymbol{\beta}, \boldsymbol{T}\boldsymbol{\beta}\rangle\,.$$

This establishes the lower bound for specific choices of $\boldsymbol{\beta}$.

Now, we turn to the universal lower bound. Using the expression for $\gamma_\infty(B)$ from Property C.1, we get that the lower bound from the theorem statement is

$$F_\infty(B) \geq \frac{\eta\sigma_{\mathsf{sgd}}^2}{2}\mathsf{Tr}\,[\boldsymbol{H}] + \frac{\eta^2 G^2}{8\pi^2\rho}\left(\int_{-\pi}^{\pi}\frac{\mathrm{d}\omega}{|B(\omega)|^2}\right)\left(\int_{-\pi}^{\pi}|B(\omega)|^2 h(\omega)\right). \tag{47}$$

The Cauchy-Schwarz inequality gives us that

$$\left(\int_{-\pi}^{\pi}\frac{\mathrm{d}\omega}{|B(\omega)|^2}\right)\left(\int_{-\pi}^{\pi}|B(\omega)|^2 h(\omega)\right) \geq \left(\int_{-\pi}^{\pi}\sqrt{h(\omega)}\,\mathrm{d}\omega\right)^2,$$

with equality attained for $|B(\omega)|^2 = 1/\sqrt{h(\omega)}$. This gives the universal lower bound on (47) over all possible choices of $B$ (or equivalently, all possible choices of $\boldsymbol{\beta}$). To further lower bound this, we use $\cos(\omega) \geq -1$ to get

$$h(\omega) = \sum_{j=1}^{d}\frac{\lambda_j}{1+(1-\eta\lambda_j)^2 - 2(1-\eta\lambda_j)\cos(\omega)} \geq \sum_{j=1}^{d}\frac{\lambda_j}{(2-\eta\lambda_j)^2} \geq \frac{1}{4}\sum_{j=1}^{d}\lambda_j = \frac{\mathsf{Tr}\,[\boldsymbol{H}]}{4}.$$

Thus, we get that (47) can be further lower bounded as

$$F_\infty(B) \geq \frac{\eta\sigma_{\mathsf{sgd}}^2}{2}\mathsf{Tr}\,[\boldsymbol{H}] + \frac{\eta^2 G^2}{8\pi^2\rho}\left(\int_{-\pi}^{\pi}\frac{\sqrt{\mathsf{Tr}\,[\boldsymbol{H}]}}{2}\,\mathrm{d}\omega\right)^2 = \frac{\eta\sigma_{\mathsf{sgd}}^2}{2}\mathsf{Tr}\,[\boldsymbol{H}] + \frac{\eta^2 G^2}{8\rho}\mathsf{Tr}\,[\boldsymbol{H}].$$

$\square$

**Missing technical proofs in the lower bound:** We now give the proof of Proposition C.19, which first relies on the following intermediate result.

**Proposition C.20.** *Consider the setting of Theorem C.18. We have for all $t, \tau$ that*

$$\mathbb{E}\left[\boldsymbol{w}_\tau \otimes \boldsymbol{\delta}_t^{(0)}\right] = \boldsymbol{0}.$$

*Proof.* For this proof, we start the sequences at $t = 0$ rather than $t = -\infty$. We drop the superscript to write $\boldsymbol{\delta}_t^{(0)}$ as $\boldsymbol{\delta}_t$. Define shorthand $\boldsymbol{Q}_t := \boldsymbol{I} - \eta\boldsymbol{x}_t \otimes \boldsymbol{x}_t$ and $\boldsymbol{R}_t := \boldsymbol{H} - \boldsymbol{x}_t \otimes \boldsymbol{x}_t$. We expand out the recursion to get

$$\begin{aligned}
\boldsymbol{\delta}_t &= \boldsymbol{Q}_{t-1}\boldsymbol{\delta}_{t-1} + \eta\boldsymbol{R}_{t-1}\boldsymbol{\theta}_{t-1}^{(0)} \\
&= \boldsymbol{Q}_{t-1}(\boldsymbol{Q}_{t-2}\boldsymbol{\delta}_{t-2} + \eta\boldsymbol{R}_{t-2}\boldsymbol{\theta}_{t-2}^{(0)}) + \eta\boldsymbol{R}_{t-1}\boldsymbol{\theta}_{t-1}^{(0)} \\
&= \boldsymbol{Q}_{t-1}\boldsymbol{Q}_{t-2}\cdots\boldsymbol{Q}_0\boldsymbol{\delta}_0 + \eta\left(\boldsymbol{R}_{t-1}\boldsymbol{\theta}_{t-1}^{(0)} + \boldsymbol{Q}_{t-1}\boldsymbol{R}_{t-2}\boldsymbol{\theta}_{t-2}^{(0)} + \cdots + \boldsymbol{Q}_{t-1}\cdots\boldsymbol{Q}_1\boldsymbol{R}_0\boldsymbol{\theta}_0^{(0)}\right).
\end{aligned}$$

The first term is zero because $\boldsymbol{\delta}_0 = \boldsymbol{0}$ at initialization. Since $\boldsymbol{R}_\tau$ is mean zero and independent of $\boldsymbol{\theta}_\tau^{(0)}$ and $\boldsymbol{R}_t$ for $t > \tau$, we have

$$\begin{aligned}
\frac{1}{\eta}\mathbb{E}[\boldsymbol{\delta}_t \otimes \boldsymbol{w}_\tau] &= \mathbb{E}[\boldsymbol{R}_{t-1}]\mathbb{E}\left[\boldsymbol{\theta}_{t-1}^{(0)} \otimes \boldsymbol{w}_\tau\right] \\
&\quad + \mathbb{E}[\boldsymbol{Q}_{t-1}]\,\mathbb{E}[\boldsymbol{R}_{t-2}]\mathbb{E}\left[\boldsymbol{\theta}_{t-2}^{(0)} \otimes \boldsymbol{w}_\tau\right] + \cdots + \mathbb{E}[\boldsymbol{Q}_{t-1}\cdots\boldsymbol{Q}_1]\,\mathbb{E}[\boldsymbol{R}_0]\mathbb{E}\left[\boldsymbol{\theta}_0^{(0)} \otimes \boldsymbol{w}_\tau\right] \\
&= \boldsymbol{0},
\end{aligned}$$

giving us the desired result. $\square$

*Proof of Proposition C.19.* We drop the superscript to write $\boldsymbol{\delta}_t^{(0)}$ as $\boldsymbol{\delta}_t$. We prove the claim by induction. At initialization, we have $\boldsymbol{\delta}_{-\infty} = \boldsymbol{0}$ so the hypothesis holds. Now assume that it holds at time $t$, i.e., $\mathbb{E}\left[\boldsymbol{\theta}_t^{(0)} \otimes \boldsymbol{\delta}_t\right] = \boldsymbol{0}$.

Next, we expand out $\mathbb{E}\left[\boldsymbol{\theta}_{t+1}^{(0)} \otimes \boldsymbol{\delta}_{t+1}\right]$ using their respective recursions. Note that $\boldsymbol{w}_t$, $\boldsymbol{H} - \boldsymbol{x}_t \otimes \boldsymbol{x}_t$ and $\xi_t$ are each zero mean and independent of all quantities appearing up to iteration $t$ (formally, they are independent of the $\sigma$-algebra generated by $(\boldsymbol{\theta}_t^{(0)}$ and $\boldsymbol{\delta}_t)$. This gives

$$\mathbb{E}\left[\boldsymbol{\theta}_{t+1}^{(0)} \otimes \boldsymbol{\delta}_{t+1}\right] = (\boldsymbol{I} - \eta\boldsymbol{H})\mathbb{E}\left[\boldsymbol{\theta}_t^{(0)} \otimes \boldsymbol{\delta}_t\right](\boldsymbol{I} - \eta\boldsymbol{H}) - \eta\mathbb{E}\left[\sum_{\tau=0}^{\infty}\beta_\tau\left(\boldsymbol{w}_{t-\tau} \otimes \delta_t^{(0)}\right)\right](\boldsymbol{I} - \eta\boldsymbol{H}). \tag{48}$$

The first term is zero by the induction hypothesis. For the second term, we can interchange the expectation and the infinite sum by the Fubini-Tonelli theorem since

$$\sum_{\tau=0}^{\infty}|\beta_\tau|\,\mathbb{E}\left|\left\langle\boldsymbol{w}_{t-\tau}, \boldsymbol{\delta}_t^{(0)}\right\rangle\right| \leq \|\boldsymbol{\beta}\|_1 \max_{\tau=0,\ldots,\infty}\mathbb{E}\left|\left\langle\boldsymbol{w}_{t-\tau}, \boldsymbol{\delta}_t^{(0)}\right\rangle\right| < \infty$$

since $\boldsymbol{\beta}_1 \in \ell^1$ and $\mathbb{E}\left|\left\langle\boldsymbol{w}_{t-\tau}, \boldsymbol{\delta}_t^{(0)}\right\rangle\right| < \infty$ because

$$\mathbb{E}\left\langle\boldsymbol{w}_{t-\tau}, \boldsymbol{\delta}_t^{(0)}\right\rangle = \mathsf{Tr}\left[\mathbb{E}\left[\boldsymbol{w}_{t-\tau} \otimes \boldsymbol{\delta}_t^{(0)}\right]\right] = 0$$

by Proposition C.20. By Proposition C.20 again, we thus get

$$\mathbb{E}\left[\sum_{\tau=0}^{\infty}\beta_\tau\left(\boldsymbol{w}_{t-\tau} \otimes \delta_t^{(0)}\right)\right] = \sum_{\tau=0}^{\infty}\beta_\tau\,\mathbb{E}\left[\left(\boldsymbol{w}_{t-\tau} \otimes \delta_t^{(0)}\right)\right] = \boldsymbol{0}.$$

$\square$

## C.4 ASYMPTOTICS OF $\nu$-NOISY-FTRL

We now state and prove the upper bound for $\nu$-Noisy-FTRL. Note that $\nu$-Noisy-FTRL can be described in the frequency domain as $|\hat{B}^\nu(\omega)|^2 = |1 - \nu - \exp(i\omega)|$.

For the proof, we define $\mathcal{I} : (0, 1)^2 \to \mathbb{R}_+$ as the integral

$$\mathcal{I}(a, b) := \int_{-\pi}^{\pi}\frac{|1 - a - \exp(i\omega)|}{|1 - b - \exp(i\omega)|^2}\,\mathrm{d}\omega. \tag{49}$$

The crux of the proof relies on a precise characterization of this integral, as we will shortly see below.

**Lemma C.21.** *Consider the integral $\mathcal{I}$ from* (49)*. It satisfies the following properties:*

*(i) For all $a \in (0, 1)$, we have*

$$\mathcal{I}(a, a) \leq 5\log(8/a).$$

*(ii) For all $a \leq b \leq 1/4$, we have*

$$\mathcal{I}(a, b) \leq \frac{128}{49}\log(8/a)\big(1 + O(a)\big).$$

*Proof.* The strategy is to reduce this integral to the standard elliptic integrals and leverage their properties to get the result. We start with the first part $\mathcal{I}(a,a)$. We use Lemma F.15 to rewrite in terms of the elliptic integral of the first kind $K(k) = \int_0^{\pi/2} d\omega / \sqrt{1 - k^2 \sin^2(\omega)}$ (denoted as (a)). Then, we use Property F.10 which says that $K(k) = O(-\log \sqrt{1-k^2})$ (denoted as (b)). This gives,

$$\mathcal{I}(a,a) \overset{(a)}{=} \frac{4}{2-a} K\left(\frac{\sqrt{1-a}}{1-a/2}\right) \overset{(b)}{\leq} \frac{5}{2-a} \log\left(\frac{4}{a}(2-a)\right) \leq 5 \log\left(\frac{8}{a}\right). \tag{50}$$

Similarly, we can express $\mathcal{I}(a,b)$ for $a \neq b$ in terms of the elliptic integral of the third kind $\Pi(\alpha^2, k)$, whose definition is given in (96). From Lemma F.16, we have for $a, b \in (0,1)$ that

$$\mathcal{I}(a,b) = \frac{2a^2}{b^2(1-a/2)} \Pi(\alpha^2, k) \quad \text{where} \quad \alpha^2 = \frac{b^2(1-a) - a^2(1-b)}{b^2(1-a/2)^2}$$

and $k = \sqrt{1-a}/(1-a/2)$. We invoke Property F.11 to bound the behavior of $\Pi(\alpha^2, k)$ as $k \to 1^-$ (i.e. $a \to 0^+$) to get

$$\mathcal{I}(a,b) \leq \frac{2a^2}{b^2(1-a/2)} \frac{1}{\sqrt{1-\alpha^2}} \log \frac{4}{\sqrt{1-k^2}} (1 + O(a))$$

$$= \frac{2(1-a/2)}{(1-b/2)^2} \log\left(\frac{4}{a}(2-a)\right)(1 + O(a)) \leq \frac{128}{49} \log(8/a)(1 + O(a)),$$

where the last inequality holds for $a \leq b \leq 1/4$. $\qquad \square$

We are now ready to prove the bounds for $\nu$-Noisy-FTRL.

**Proposition C.22.** *Consider the setting of Theorem C.15 with $\sigma_{\mathsf{sgd}}^2 = 0$. Then, $\nu$-Noisy-FTRL with $\nu \leq \eta\mu$ satisfies*

$$F_\infty(\hat{\boldsymbol{\beta}}^\nu) \leq C \max\{1, C_{\mathsf{kurt}}\} \eta^2 G^2 \rho^{-1} \mathsf{Tr}[\boldsymbol{H}] \log^2\left(\frac{8}{\nu}\right) + \widetilde{O}(\eta^3 R^2 \mu G^2 \rho^{-1}),$$

*for a universal constant $C > 0$, and $\widetilde{O}(\cdot)$ suppresses polylogarithmic terms in the problem parameters.*

*Proof.* We use $C$ to denote a universal constant that can change from line to line. We can express the bound of Theorem C.15 with our specific choice of $B(\omega)$ as

$$F_\infty(\hat{B}^\nu) \leq C \max\{1, C_{\mathsf{kurt}}\} \mathcal{I}(\nu, \nu) \sum_{j=1}^d \lambda_j \mathcal{I}(\nu, \eta\lambda_j). \tag{51}$$

For the $\mathcal{I}(\nu, \nu)$ term, we plug in Lemma C.21(i). We plug $a = \nu$ and $b = \eta\lambda_j$ into Lemma C.21(ii) to get (note that its conditions are satisfied)

$$\mathcal{I}(\nu, \eta\lambda_j) \leq C \log\left(\frac{8}{\nu}\right)(1 + O(\nu)). \tag{52}$$

The last term is $O(\nu) \leq O(\eta\mu)$. Plugging in (50) and (52) into (51) and using $\mathsf{Tr}[\boldsymbol{H}] = \sum_{j=1}^n \lambda_j \leq R^2$ completes the proof. $\qquad \square$

**Remark C.23** (Contribution per eigendirection)**.** *We continue the discussion of Remark C.16. The proof of Proposition C.22 shows that the contribution of the $j^{th}$ eigendirection to the asymptotic suboptimality is proportional to*

$$Err_j = \lambda_j \mathcal{I}(\nu, \eta\lambda_j).$$

*As long as $\nu \leq \eta\mu$, we get from Lemma C.21 that $Err_j \leq O(\lambda_j \log(1/\nu))$. Thus, the error contributed drops proportional to $\lambda_j$, leading to an effective dimension dependence for $\nu$-Noisy-FTRL.*

### C.5 ASYMPTOTICS OF ANTI-PGD

As we discussed in Table 2, anti-PGD (Orvieto et al., 2022) is a special case of Noisy-FTRL with $\beta = (1, -1, 0, \ldots)$. Then, we have that $(\mathrm{Toeplitz}(\beta))^{-1}$ is the lower triangular matrix of all ones, so we have $\gamma_T(\beta) = T$, or that its limiting sensitivity is infinite.

We can circumvent the infinity by damping $\beta = (1, -(1 - \nu), 0, \ldots)$ for some $0 < \nu < 1$ to be decided later. In this case, we have $B(\omega) = 1 - (1 - \nu)\exp(-i\omega)$, so that $|B(\omega)|^2 = |1 - \nu - \exp(i\omega)|^2$, which is the analogue of $\nu$-Noisy-FTRL with a square.

**Proposition C.24.** *Consider the setting of Theorem C.15 with $\sigma_{\mathsf{sgd}}^2 = 0$ and $\beta = (1, -(1 - \eta\lambda), 0, \ldots)$ for some $\lambda \in (0, 1/\eta]$. Then, we have,*

$$F_\infty(\beta) = \Theta\left(\eta G^2 \rho^{-1}\left(\nu d + \frac{\eta \mathsf{Tr}\,[\boldsymbol{H}]}{\nu}\right)\right).$$

*Further, if the learning rate satisfies $\eta = c/\mathsf{Tr}\,[\boldsymbol{H}]$ and we take $\beta = (1, -(1 - \sqrt{1/d}), \ldots)$, we get*

$$F_\infty(\beta) = \Theta\left((c^{1/2} + c^{-1/2})\eta^{3/2}\sigma^2\sqrt{d\,\mathsf{Tr}\,[\boldsymbol{H}]}\right).$$

*Proof.* Let $\sigma^2 = G^2/(2\rho)$. From Theorems C.15 and C.18, we get that

$$F_\infty(\beta) = \Theta\left(\eta^2\sigma^2\left(\int_{-\pi}^{\pi}\frac{\mathrm{d}\omega}{|1 - \nu - \exp(i\omega)|^2}\right)\left(\sum_{j=1}^{d}\lambda_j\int_{-\pi}^{\pi}\frac{|1 - \nu - \exp(i\omega)|^2}{|1 - \eta\lambda_j - \exp(i\omega)|^2}\,\mathrm{d}\omega\right)\right). \quad (53)$$

Using Lemma F.12, we have

$$\int_{-\pi}^{\pi}\frac{\mathrm{d}\omega}{|1 - \nu - \exp(i\omega)|^2} = \frac{2\pi}{\nu(2 - \nu)} = \Theta\left(\frac{1}{\nu}\right).$$

For the second integral, we expand out the numerator and invoke Lemma F.12 again to get

$$\frac{1}{2\pi}\int_{-\pi}^{\pi}\frac{|1 - \nu - \exp(i\omega)|^2}{|1 - \eta\lambda_j - \exp(i\omega)|^2}\,\mathrm{d}\omega = \frac{1 + (1 - \nu)^2}{\eta\lambda_j(2 - \eta\lambda_j)} - 2(1 - \nu)\frac{1 - \eta\lambda_j}{\eta\lambda_j(2 - \eta\lambda_j)}$$

$$= \Theta\left(\frac{\nu^2}{\eta\lambda_j} + 1\right),$$

where we use $1 \leq 2 - \nu \leq 2$ and the same for $\lambda_j$ instead of $\lambda$. Plugging the two integrals back into (53) completes the proof. $\square$

### C.6 EFFECTIVE DIMENSION AND THE STABLE RANK

The stable/numerical rank $\mathsf{srank}(\boldsymbol{A})$ of a matrix $\boldsymbol{A}$ is defined as

$$\mathsf{srank}(\boldsymbol{A}) = \frac{\|\boldsymbol{A}\|_F^2}{\sigma_{\max}(\boldsymbol{A})^2},$$

i.e., the squared ratio of the Frobenius norm of a matrix to its largest singular value (Rudelson & Vershynin, 2007). By comparing this to our definition of the effective dimension, we find that $d_{\mathsf{eff}}(\boldsymbol{H}) = \mathsf{srank}(\boldsymbol{H}^{1/2})$. Note that the effective dimension is also called the "intrinsic dimension" by Martinsson & Tropp (2020).

The stable rank of a matrix is a continuous function while the true rank is discontinuous. Thus, it is highly desirable for the error of a numerical algorithm to scale with the stable rank of its matrix input rather than the

true rank (Rudelson & Vershynin, 2007; Martinsson & Tropp, 2020). The stable rank is thus a fundamental quantity appearing in various fields such as randomized linear algebra (Cohen et al., 2016; Martinsson & Tropp, 2020) and matrix concentration (Hsu et al., 2011; Minsker, 2017).

Our results show that $\nu$-DP-FTRL's error has the desirable property of scaling with the stable rank (i.e. effective dimension) of the Hessian $\boldsymbol{H}$ rather than its true rank (i.e. the problem's dimension).

### C.7   Proofs of Technical Lemmas

We now prove Lemma C.4.

*Proof of Lemma C.4.* Denote
$$I = \int_{-\pi}^{\pi} \frac{|B(\omega)|^2 \, \mathrm{d}\omega}{|1 - \eta\lambda_j - \exp(i\omega)|^2} \,.$$

The denominator is simply
$$|1 - \exp(i\omega) - \eta\lambda_j|^2 = 1 + (1 - \eta\lambda_j)^2 - 2(1 - \eta\lambda_j)\cos\omega \,. \tag{54}$$

We expand the numerator as
$$\begin{aligned}
|B(\omega)|^2 &= \sum_{t=0}^{\infty} \beta_t^2 + \sum_{t=0}^{\infty} \sum_{\tau=0}^{t-1} \beta_t \beta_\tau \big( \exp(i\omega(t-\tau)) + \exp(-i\omega(\tau - t)) \big) \\
&= \sum_{t=0}^{\infty} \beta_t^2 + 2 \sum_{t=0}^{\infty} \sum_{\tau=0}^{t-1} \beta_t \beta_\tau \cos(\omega(t-\tau)) \\
&= \sum_{t=0}^{\infty} \sum_{\tau=0}^{\infty} \beta_t \beta_\tau \cos(\omega(t-\tau)) \,.
\end{aligned} \tag{55}$$

This is bounded since the Cauchy-Schwarz inequality gives
$$|B(\omega)|^2 \le \|\boldsymbol{\beta}\|_2^2 < \infty \,.$$

Thus, we can apply Fubini's theorem to exchange the sum and integral to give
$$\begin{aligned}
I &= \sum_{t=0}^{\infty} \sum_{\tau=0}^{\infty} \beta_t \beta_\tau \int_{-\pi}^{\pi} \frac{\cos(\omega(t-\tau))\mathrm{d}\omega}{1 + (1 - \eta\lambda_j)^2 - 2(1 - \eta\lambda_j)\cos(\omega)} \\
&= \sum_{t=0}^{\infty} \sum_{\tau=0}^{\infty} \frac{2\pi}{1 - (1 - \eta\lambda_j)^2} (1 - \eta\lambda_j)^{|t-\tau|} = \frac{2\pi\langle\boldsymbol{\beta}, \boldsymbol{T}_j\boldsymbol{\beta}\rangle}{\eta\lambda_j(2 - \eta\lambda_j)} \,,
\end{aligned}$$

where we evaluated the integral using Lemma F.12. We use $1 \le 2 - \eta\lambda_j \le 2$ to complete the proof.   $\square$

## D   Finite-Time Privacy-Utility Tradeoffs for Linear Regression

The goal of this section is to establish the finite time convergence of DP-FTRL. The key idea of the proof is to establish high probability bounds on the $\ell_2$ norm of the iterates of Noisy-FTRL and use that to deduce a clip norm that does not clip any gradients with high probability.

The outline of this section is as follows:

- **Appendix D.1**: Preliminaries, including setup, notation and assumptions.

- **Appendix D.2**: High probability bounds the iterates of Noisy-FTRL.
- **Appendix D.3**: Expected bounds on the iterates of Noisy-FTRL.
- **Appendix D.4**: Connecting DP-FTRL to Noisy-FTRL for the final bound privacy-utility bounds (Corollary D.14 for DP-SGD and Corollary D.15 for DP-FTRL).

## D.1   SETUP, ASSUMPTIONS, AND NOTATION

In this section, we fix the precise notation and assumptions. We also give some preliminary results.

### D.1.1   ASSUMPTIONS

We make the following assumptions throughout this section.

**Assumption D.1.** *The data distribution $\mathbb{P}_{\mathsf{data}}$ satisfies the following:*

- **(B1)** *Input Distribution: The inputs have mean $\mathbb{E}[\boldsymbol{x}] = \boldsymbol{0}$ and covariance $\mathbb{E}[\boldsymbol{x} \otimes \boldsymbol{x}] =: \boldsymbol{H}$. We have $\mu \boldsymbol{I} \preceq \boldsymbol{H} \preceq L\boldsymbol{I}$ for $\mu, L > 0$. Further, $\boldsymbol{H}^{-1/2}\boldsymbol{x}$ is element-wise independent and sub-Gaussian with variance proxy 1, e.g. $\boldsymbol{H}^{-1/2}\boldsymbol{x} \sim \mathcal{N}(0, \boldsymbol{I})$.*
- **(B2)** *Noise Distribution: There exists a $\boldsymbol{\theta}_\star \in \mathbb{R}^d$ such that $y = \langle \boldsymbol{\theta}_\star, \boldsymbol{x} \rangle + \xi$, where $\xi$ is independent of $\boldsymbol{x}$ and is zero-mean sub-Gaussian with variance proxy $\sigma_{\mathsf{sgd}}^2$, e.g. $\xi \sim \mathcal{N}(0, \sigma_{\mathsf{sgd}}^2)$.*

These assumptions are a strengthening of Assumption C.2 which are necessitated by concentration arguments to follow below.

### D.1.2   NOTATION

- As in Assumption C.2, we denote $R^2$ as the smallest number such that the fourth moment of $\boldsymbol{x}$ is bounded as

$$\mathbb{E}\left[\|\boldsymbol{x}\|_2^2 \, \boldsymbol{x} \otimes \boldsymbol{x}\right] \preceq R^2 \boldsymbol{H} \,. \tag{56}$$

Under Assumption (**B1**), we have $R^2 = \Theta(\mathsf{Tr}\,[\boldsymbol{H}])$ always. While $\mathsf{Tr}\,[\boldsymbol{H}] \leq R^2$ directly follows from (56) using Jensen's inequality, we show that $R^2 \leq 3\mathsf{Tr}\,[\boldsymbol{H}]$ in Property C.3 in Appendix C.1.
- It is convenient to rewrite the Noisy-FTRL recursion (23) in terms of the difference $\boldsymbol{\theta}_t' := \boldsymbol{\theta}_t - \boldsymbol{\theta}_\star$ as

$$\boldsymbol{\theta}_{t+1}' = \big(\boldsymbol{I} - \eta(\boldsymbol{x}_t \otimes \boldsymbol{x}_t)\big)\boldsymbol{\theta}_t' + \eta\,\xi_t\boldsymbol{x}_t - \eta\sum_{\tau=0}^{t} \beta_\tau \boldsymbol{w}_{t-\tau} \,. \tag{57}$$

We will show in the upcoming Property D.2 that $\boldsymbol{\theta}_t' = \hat{\boldsymbol{\theta}}_t + \widetilde{\boldsymbol{\theta}}^{\,\mathsf{sgd}} + \widetilde{\boldsymbol{\theta}}^{\,\mathsf{dp}}$, where $\hat{\boldsymbol{\theta}}_t$ captures the effect of the initial iterate, $\widetilde{\boldsymbol{\theta}}^{\,\mathsf{sgd}}$ captures the effect of the SGD noise, and $\widetilde{\boldsymbol{\theta}}^{\,\mathsf{dp}}$ captures the effect of the additive DP noise. We will define these quantities now and state and prove Property D.2 later. Note that these recursions are defined for the same sequences of input realizations $(\boldsymbol{x}_0, \boldsymbol{x}_1, \ldots)$ drawn from $\mathbb{P}_{\mathsf{data}}$, linear model noise realizations $(\xi_0, \xi_1, \ldots)$, and DP noise realizations $(\boldsymbol{w}_0, \boldsymbol{w}_1, \ldots)$.
- We define the noise-free version of the DP-FTRL recursion as $\hat{\boldsymbol{\theta}}_0 = \boldsymbol{\theta}_0'$ and

$$\hat{\boldsymbol{\theta}}_{t+1} = \big(\boldsymbol{I} - \eta(\boldsymbol{x}_t \otimes \boldsymbol{x}_t)\big)\hat{\boldsymbol{\theta}}_t \,. \tag{58}$$

- The effect of the SGD noise in the Noisy-FTRL process can be quantified by creating a process starting from $\widetilde{\boldsymbol{\theta}}_0^{\,\mathsf{sgd}} = \boldsymbol{0}$ with no DP noise (i.e. $\boldsymbol{w}_\tau \equiv \boldsymbol{0}$):

$$\widetilde{\boldsymbol{\theta}}_{t+1}^{\,\mathsf{sgd}} = \big(\boldsymbol{I} - \eta(\boldsymbol{x}_t \otimes \boldsymbol{x}_t)\big)\widetilde{\boldsymbol{\theta}}_t^{\,\mathsf{sgd}} + \eta\,\xi_t\boldsymbol{x}_t \,. \tag{59}$$

- The effect of the DP noise in the Noisy-FTRL process can be quantified by creating a process starting from $\widetilde{\boldsymbol{\theta}}_0^{\mathsf{dp}} = \mathbf{0}$ with no SGD noise (i.e., $\xi_t \equiv 0$):

$$\widetilde{\boldsymbol{\theta}}_{t+1}^{\mathsf{dp}} = \left(\boldsymbol{I} - \eta(\boldsymbol{x}_t \otimes \boldsymbol{x}_t)\right)\widetilde{\boldsymbol{\theta}}_t^{\mathsf{dp}} - \eta \sum_{\tau=0}^{t} \beta_\tau \boldsymbol{w}_{t-\tau} \,. \tag{60}$$

- For an input $\boldsymbol{x}_t$ drawn from $\mathbb{P}_{\mathsf{data}}$ We define the matrix

$$\boldsymbol{Q}_t := \boldsymbol{I} - \eta \boldsymbol{x}_t \otimes \boldsymbol{x}_t \,. \tag{61}$$

Note that $\mathbb{E}[\boldsymbol{Q}_t] = \boldsymbol{I} - \eta \boldsymbol{H}$.
- Define the linear operator $\mathcal{P} : \mathbb{S}_+^d \to \mathbb{S}_+^d$ that operates on the cone of PSD matrices given by

$$\mathcal{P}\boldsymbol{M} = \mathbb{E}[(\boldsymbol{I} - \eta\boldsymbol{x} \otimes \boldsymbol{x})\boldsymbol{M}(\boldsymbol{I} - \eta\boldsymbol{x} \otimes \boldsymbol{x})] \,, \tag{62}$$

where $\boldsymbol{x}$ is an input drawn from $\mathbb{P}_{\mathsf{data}}$. By definition, we have $\mathbb{E}[\boldsymbol{Q}_t\boldsymbol{M}\boldsymbol{Q}_t] = \mathcal{P}\boldsymbol{M}$ and by independence,

$$\mathbb{E}[\boldsymbol{Q}_t\boldsymbol{Q}_{t-1}\boldsymbol{M}\boldsymbol{Q}_{t-1}\boldsymbol{Q}_t] = \mathcal{P}(\mathcal{P}\boldsymbol{M}) = \mathcal{P}^2\boldsymbol{M} \,. \tag{63}$$

This extends to higher powers of $\mathcal{P}$ as well. Finally, we will heavily use the fact that $\mathsf{Tr}\,[\mathcal{P}\boldsymbol{M}] \le (1 - \eta\mu)\mathsf{Tr}\,[\boldsymbol{M}]$ for PSD matrices $\boldsymbol{M}$ (see Lemma F.18 for a proof).
- For each iteration $t$, we define the PSD matrix $\boldsymbol{\Sigma}_t^{\mathsf{sgd}}$ as

$$\boldsymbol{\Sigma}_t^{\mathsf{sgd}} = \boldsymbol{x}_{t-1} \otimes \boldsymbol{x}_{t-1} + \boldsymbol{Q}_{t-1}(\boldsymbol{x}_{t-2} \otimes \boldsymbol{x}_{t-2})\boldsymbol{Q}_{t-1} + \cdots + \boldsymbol{Q}_{t-1}\cdots\boldsymbol{Q}_1(\boldsymbol{x}_0 \otimes \boldsymbol{x}_0)\boldsymbol{Q}_1\cdots\boldsymbol{Q}_{t-1} \,, \tag{64}$$

- For each iteration $t$, we define the PSD matrix $\boldsymbol{\Sigma}_t^{\mathsf{dp}}$ as

$$\begin{aligned} \boldsymbol{\Sigma}_t^{\mathsf{dp}} &= \sum_{\tau=0}^{t-1} \boldsymbol{V}_{t,\tau}\boldsymbol{V}_{t,\tau}^\top \quad \text{where} \\ \boldsymbol{V}_{t,\tau} &= \begin{cases} \beta_\tau\boldsymbol{I} + \beta_{\tau-1}\boldsymbol{Q}_{t-1} + \cdots + \beta_0\boldsymbol{Q}_{t-1}\cdots\boldsymbol{Q}_{t-\tau} \,, & \text{if } 1 \le \tau \le t-1 \,, \\ \beta_0\boldsymbol{I} \,, & \text{if } \tau = 0 \,. \end{cases} \end{aligned} \tag{65}$$

### D.1.3 PRELIMINARY RESULTS

The first result is a decomposition of the Noisy-FTRL process into three processes: (a) gradient descent without additive noise, (b) a noise process with only noise from the linear model, and (c) a noise process with only the DP noise.

**Property D.2.** *For the sequences $\boldsymbol{\theta}_t', \hat{\boldsymbol{\theta}}_t, \widetilde{\boldsymbol{\theta}}_t^{\mathsf{sgd}}, \widetilde{\boldsymbol{\theta}}_t^{\mathsf{dp}}$ defined in Equations* (57) *to* (60)*, we have the following:*

$$\boldsymbol{\theta}_t' = \hat{\boldsymbol{\theta}}_t + \widetilde{\boldsymbol{\theta}}_t^{\mathsf{sgd}} + \widetilde{\boldsymbol{\theta}}_t^{\mathsf{dp}} \tag{66}$$

$$\hat{\boldsymbol{\theta}}_t = \boldsymbol{Q}_t\cdots\boldsymbol{Q}_0\boldsymbol{\theta}_0' \tag{67}$$

$$\widetilde{\boldsymbol{\theta}}_t^{\mathsf{sgd}} = \eta\left(\boldsymbol{x}_t\xi_t + \boldsymbol{Q}_t\boldsymbol{x}_{t-1}\xi_{t-1} + \cdots + \boldsymbol{Q}_t\cdots\boldsymbol{Q}_1\boldsymbol{x}_0\xi_0\right) \tag{68}$$

$$\begin{aligned} \widetilde{\boldsymbol{\theta}}_t^{\mathsf{dp}} &= -\eta\left(\sum_{\tau=0}^{t}\beta_\tau\boldsymbol{w}_{t-\tau} + \boldsymbol{Q}_t\sum_{\tau=0}^{t-1}\beta_\tau\boldsymbol{w}_{t-1-\tau} + \cdots + \boldsymbol{Q}_t\cdots\boldsymbol{Q}_1(\beta_0\boldsymbol{w}_0)\right) \\ &= -\eta\Big(\beta_0\boldsymbol{w}_{t-1} + (\beta_1\boldsymbol{I} + \beta_0\boldsymbol{Q}_{t-1})\boldsymbol{w}_{t-2} + \cdots + (\beta_{t-1}\boldsymbol{I} + \beta_{t-2}\boldsymbol{Q}_{t-1} + \cdots + \beta_0\boldsymbol{Q}_{t-1}\cdots\boldsymbol{Q}_1)\boldsymbol{w}_0\Big) \,. \end{aligned} \tag{69}$$

*Proof.* The expressions follow from unrolling their respective updates. By unrolling the DP-FTRL update (57), we get,

$$
\begin{aligned}
\boldsymbol{\theta}'_{t+1} &= \boldsymbol{Q}_t \boldsymbol{\theta}'_t + \eta \boldsymbol{x}_t \xi_t - \eta \sum_{\tau=0}^{t} \beta_\tau \boldsymbol{w}_{t-\tau} \\
&= \boldsymbol{Q}_t \boldsymbol{Q}_{t-1} \boldsymbol{\theta}'_{t-1} + \eta \left( \boldsymbol{x}_t \xi_t + \boldsymbol{Q}_t \boldsymbol{x}_{t-1} \xi_{t-1} \right) - \eta \left( \sum_{\tau=0}^{t} \beta_\tau \boldsymbol{w}_{t-\tau} + \boldsymbol{Q}_t \sum_{\tau=0}^{t-1} \beta_\tau \boldsymbol{w}_{t-1-\tau} \right) \\
&= \boldsymbol{Q}_t \cdots \boldsymbol{Q}_0 \boldsymbol{\theta}'_0 + \eta \left( \boldsymbol{x}_t \xi_t + \boldsymbol{Q}_t \boldsymbol{x}_{t-1} \xi_{t-1} + \cdots + \boldsymbol{Q}_t \cdots \boldsymbol{Q}_1 \boldsymbol{x}_0 \xi_0 \right) \\
&\quad - \eta \left( \sum_{\tau=0}^{t} \beta_\tau \boldsymbol{w}_{t-\tau} + \boldsymbol{Q}_t \sum_{\tau=0}^{t-1} \beta_\tau \boldsymbol{w}_{t-1-\tau} + \cdots + \boldsymbol{Q}_t \cdots \boldsymbol{Q}_1 (\beta_0 \boldsymbol{w}_0) \right) .
\end{aligned}
$$

Unrolling Equations (58) to (60) respectively gives Equations (67) to (69), and comparing them with the expression above gives Equation (66). □

## D.2 High-Probability Bounds on Noisy-FTRL

The goal of this subsection is to prove a high probability bound on norms of the iterates of Noisy-FTRL. We require a technical convergence condition on the weights $\boldsymbol{\beta}$.

**Definition D.3.** *A sequence $\boldsymbol{\beta} = (\beta_0, \beta_1, \ldots)$ is said to satisfy Half-Expo Decay with parameter $\nu \in (0, 1)$ if for all nonnegative integers $\tau$, we have*

$$
|\beta_0|(1-\nu)^{\tau/2} + |\beta_1|(1-\nu)^{(\tau-1)/2} + \cdots + |\beta_\tau| \le C(1-\nu)^{\tau/2} \tag{70}
$$

*for a universal constant $C > 0$.*

**Theorem D.4.** *Fix a constant $0 < p < 1$ and suppose the Assumption D.1 holds. Consider the sequence $(\boldsymbol{\theta}_t)_{t=0}^{T-1}$ of iterates and the sequence $(\boldsymbol{g}_t)_{t=0}^{T-1}$ of gradients when running Noisy-FTRL for $T$ iterations with noise coefficients $\boldsymbol{\beta} = (\beta_0, \ldots, \beta_{T-1})$, DP noise $\boldsymbol{w}_t \sim \mathcal{N}(\boldsymbol{0}, \sigma^2 \boldsymbol{I})$ of a given variance[6] $\sigma^2$, a learning rate $\eta \le \left( cR^2 \log(T/p) \right)$ for a universal constant $c \ge 1$. Further, suppose that $\boldsymbol{\beta}$ satisfies Half-Expo Decay with parameter $\nu$ for some $\nu \le \eta\mu$. Then, with probability at least $1 - p$, we have*

$$
\|\boldsymbol{\theta}'_t\|_2^2 \le C \left( \|\boldsymbol{\theta}'_0\|_2^2 + \frac{\eta R^2 \sigma_{\mathsf{sgd}}^2}{\mu} + \frac{\eta^2 \sigma^2 d \|\boldsymbol{\beta}\|_1^2}{\nu} \right) \log^3 \left( \frac{T}{p} \right) \quad and
$$

$$
\|\boldsymbol{g}_t\|_2^2 \le C R^4 \left( \|\boldsymbol{\theta}'_0\|_2^2 + \frac{\eta R^2 \sigma_{\mathsf{sgd}}^2}{\mu} + \frac{\sigma_{\mathsf{sgd}}^2}{R^2} + \frac{\eta^2 \sigma^2 d \|\boldsymbol{\beta}\|_1^2}{\nu} \right) \log^5 \left( \frac{T}{p} \right) .
$$

*for a universal constant $C$.*

We prove this theorem over a sequence of intermediate results.

### D.2.1 Proof Setup: Definition of Events

The proof strategy relies on defining some events (that hold with high probability from concentration of measure) and proving the required boundedness under those events. Consider $0 < p < 1$ and a universal constant $C$ from statement of Theorem D.4. We define the following events.

---

[6]In the context of this paper, we have $\sigma^2 = G^2 \gamma(\boldsymbol{\beta})^2 / (2\rho)$.

- Define the event where the inputs are bounded in norm as:

$$\mathcal{E}_1 := \bigcap_{t=0}^{T-1} \left\{ \|\boldsymbol{x}_t\|_2^2 \leq CR^2 \log\left(\frac{T}{p}\right) \right\}. \tag{71}$$

- Define an event where the noise in the linear model is bounded as:

$$\mathcal{E}_2 := \bigcap_{t=0}^{T-1} \left\{ |\xi_t|^2 \leq 2\sigma_{\mathsf{sgd}}^2 \log\left(\frac{2T}{p}\right) \right\}. \tag{72}$$

- Define the event where the norm of $\widetilde{\boldsymbol{\theta}}^{\,\mathsf{sgd}}$ defined in (59) is bounded

$$\mathcal{E}_1^{\mathsf{sgd}} := \bigcap_{t=0}^{T-1} \left\{ \left\|\widetilde{\boldsymbol{\theta}}^{\,\mathsf{sgd}}\right\|_2^2 \leq C\eta^2 \sigma_{\mathsf{sgd}}^2 \,\mathsf{Tr}\left[\boldsymbol{\Sigma}_t^{\mathsf{sgd}}\right] \log\left(\frac{T}{p}\right) \right\}, \tag{73}$$

where we define the random matrix $\boldsymbol{\Sigma}_t^{\mathsf{sgd}} = \boldsymbol{x}_{t-1} \otimes \boldsymbol{x}_{t-1} + \boldsymbol{Q}_{t-1}(\boldsymbol{x}_{t-2} \otimes \boldsymbol{x}_{t-2})\boldsymbol{Q}_{t-1} + \cdots + \boldsymbol{Q}_{t-1}\cdots\boldsymbol{Q}_1(\boldsymbol{x}_0 \otimes \boldsymbol{x}_0)\boldsymbol{Q}_1\cdots\boldsymbol{Q}_{t-1}$ (see also (64)). When this event holds, we have that $\boldsymbol{0} \preceq \boldsymbol{Q}_t \preceq \boldsymbol{I}$ for $t = 0, \ldots, T-1$ as long as $\eta \leq 1/\left(CR^2 \log(T/p)\right)$. Indeed, in this case, we have

$$\boldsymbol{I} - \eta \boldsymbol{x}_t \otimes \boldsymbol{x}_t \succeq \left(1 - \eta\|\boldsymbol{x}_t\|_2^2\right)\boldsymbol{I} \succeq \boldsymbol{0}. \tag{74}$$

- The components of the sum defining $\boldsymbol{\Sigma}_t^{\mathsf{sgd}}$ are the PSD matrices $\boldsymbol{W}_{t,\tau}$, defined for $\tau \leq t-1$ as

$$\boldsymbol{W}_{t,\tau} = \begin{cases} \boldsymbol{Q}_{t-1}\cdots\boldsymbol{Q}_{\tau+1}(\boldsymbol{x}_\tau \otimes \boldsymbol{x}_\tau)\boldsymbol{Q}_{\tau+1}\cdots\boldsymbol{Q}_{t-1}, & \text{if } \tau < t-1, \\ \boldsymbol{x}_{t-1} \otimes \boldsymbol{x}_{t-1}, & \text{if } \tau = t-1. \end{cases} \tag{75}$$

Define the event where these are bounded in trace as

$$\mathcal{E}_2^{\mathsf{sgd}} := \bigcap_{t=0}^{T-1}\bigcap_{\tau=0}^{t-1} \left\{ \mathsf{Tr}\left[\boldsymbol{W}_{t,\tau}\right] \leq \frac{T^2 R^2}{p}(1-\eta\mu)^{t-1-\tau} \right\}. \tag{76}$$

- Define the event where the norm of $\widetilde{\boldsymbol{\theta}}^{\,\mathsf{dp}}$ defined in (60) is bounded as

$$\mathcal{E}_1^{\mathsf{dp}} := \bigcap_{t=0}^{T-1} \left\{ \left\|\widetilde{\boldsymbol{\theta}}_t^{\mathsf{dp}}\right\|_2^2 \leq C\eta^2 \sigma^2 \,\mathsf{Tr}\left[\boldsymbol{\Sigma}_t^{\mathsf{dp}}\right] \log\left(\frac{T}{p}\right) \right\}, \tag{77}$$

where $\boldsymbol{\Sigma}_t^{\mathsf{dp}}$ is defined in (65).
- Define the event where the matrix $\boldsymbol{V}_{t,\tau}$ defined in (65) is bounded in trace:

$$\mathcal{E}_2^{\mathsf{dp}} := \bigcap_{t=0}^{T-1}\bigcap_{\tau=0}^{t-1} \left\{ \mathsf{Tr}\left[\boldsymbol{V}_{t,\tau}\boldsymbol{V}_{t,\tau}^\top\right] \leq \frac{T^2 d}{p}\left(\sum_{k=0}^{\tau} |\beta_k|(1-\eta\mu)^{(\tau-k)/2}\right) \right\}. \tag{78}$$

We show that all these events hold with high probability.

**Proposition D.5.** *Consider the setting of Theorem D.4. We have,*

$$\mathbb{P}\left(\mathcal{E}_1 \cap \mathcal{E}_2 \cap \mathcal{E}_1^{\mathsf{sgd}} \cap \mathcal{E}_2^{\mathsf{sgd}} \cap \mathcal{E}_1^{\mathsf{dp}} \cap \mathcal{E}_2^{\mathsf{dp}}\right) \geq 1 - 6p.$$

*Proof.* We will show that each of the events holds with probability at least $1 - p$ and a union bound gives the desired result.

**Event $\mathcal{E}_1$:** Since $\boldsymbol{z}_t = \boldsymbol{H}^{-1/2}\boldsymbol{x}_t$ is element-wise independent and 1-sub-Gaussian, we have from the Hanson-Wright inequality (Lemma F.6) that

$$\mathbb{P}(\|\boldsymbol{x}_t\|_2^2 > C\mathsf{Tr}\,[\boldsymbol{H}]\log(1/p)) = \mathbb{P}(\langle \boldsymbol{z}_t, \boldsymbol{H}\boldsymbol{z}_t\rangle > C\mathsf{Tr}\,[\boldsymbol{H}]\log(1/p)) \le p\,.$$

Taking a union bound over $t = 0, 1, \ldots, T - 1$ gives that $\mathbb{P}(\mathcal{E}_1) \ge 1 - p$.

**Event $\mathcal{E}_2$:** Since $\xi_t$ is sub-Gaussian with mean zero and variance proxy $\sigma_{\mathsf{sgd}}^2$, we have,

$$\mathbb{P}(|\xi_t| > s) \le 2\exp\left(-\frac{s^2}{2\sigma_{\mathsf{sgd}}^2}\right)\,.$$

Setting the right side equal to $p/T$ and taking a union bound over $t = 0, 1, \ldots, T - 1$ gives $\mathbb{P}(\mathcal{E}_2) \ge 1 - p$.

**Event $\mathcal{E}_1^{\mathsf{sgd}}$:** From the expression for $\widetilde{\boldsymbol{\theta}}_t^{\mathsf{sgd}}$ from (68), we can say that $\widetilde{\boldsymbol{\theta}}_t^{\mathsf{sgd}}$ conditioned on $\boldsymbol{x}_0, \ldots, \boldsymbol{x}_{t-1}$ is mean zero and satisfies

$$\widetilde{\boldsymbol{\theta}}_t^{\mathsf{sgd}} = \eta \underbrace{\begin{bmatrix}\boldsymbol{x}_{t-1} & \boldsymbol{Q}_{t-1}\boldsymbol{x}_{t-1} & \cdots & (\boldsymbol{Q}_{t-1}\cdots\boldsymbol{Q}_1\boldsymbol{x}_0)\end{bmatrix}}_{=:\boldsymbol{M}_t}\begin{bmatrix}\xi_{t-1}\\ \vdots \\ \xi_0\end{bmatrix}\,.$$

Using the assumption that each $\xi_\tau$ is independent and sub-Gaussian with variance proxy $\sigma_{\mathsf{sgd}}^2$, we get from the Hanson-Wright inequality (Lemma F.6) again that

$$\mathbb{P}\left(\left\|\widetilde{\boldsymbol{\theta}}_t^{\mathsf{sgd}}\right\|_2^2 > C\eta^2\sigma_{\mathsf{sgd}}^2\,\mathsf{Tr}\,[\boldsymbol{M}_t\boldsymbol{M}_t^\top]\log(1/p)\right) = \mathbb{P}\left(\langle \boldsymbol{\xi}_{:t}, \boldsymbol{M}_t\boldsymbol{M}_t^\top\boldsymbol{\xi}_{:t}\rangle > C\eta^2\sigma_{\mathsf{sgd}}^2\,\mathsf{Tr}\,[\boldsymbol{M}_t\boldsymbol{M}_t^\top]\log(1/p)\right) \le p\,.$$

Next, we confirm that

$$\mathsf{Tr}\,[\boldsymbol{M}_t\boldsymbol{M}_t^\top] = \|\boldsymbol{x}_{t-1}\|_2^2 + \|\boldsymbol{Q}_{t-1}\boldsymbol{x}_{t-1}\|_2^2 + \cdots + \|\boldsymbol{Q}_{t-1}\cdots\boldsymbol{Q}_1\boldsymbol{x}_0\|_2^2 = \mathsf{Tr}\,\left[\boldsymbol{\Sigma}_t^{\mathsf{sgd}}\right]\,.$$

Finally, a union bound over $t = 0, 1, \ldots, T - 1$ gives that $\mathbb{P}(\mathcal{E}_1^{\mathsf{sgd}}) \ge 1 - p$.

**Event $\mathcal{E}_2^{\mathsf{sgd}}$:** Markov's inequality gives

$$\mathbb{P}\left(\mathsf{Tr}\,[\boldsymbol{W}_{t,\tau}] > s\right) \le \frac{1}{s}\mathbb{E}\,[\boldsymbol{W}_{t,\tau}] \le (1 - \eta\mu)^{t-1-\tau}\frac{R^2}{s}$$

where the calculations for the expected bound are deferred to Lemma D.9. Taking a union bound over all $T(T + 1)/2 \le T^2$ choices of $(t, \tau)$ gives $\mathbb{P}(\mathcal{E}_2^{\mathsf{sgd}}) \ge 1 - p$.

**Event $\mathcal{E}_1^{\mathsf{dp}}$:** From the expression for $\widetilde{\boldsymbol{\theta}}_t^{\mathsf{dp}}$ from (69), we deduce that

$$\widetilde{\boldsymbol{\theta}}_t^{\mathsf{dp}}\,|\,\boldsymbol{x}_0, \ldots, \boldsymbol{x}_{t-1} \sim \mathcal{N}(\boldsymbol{0}, \eta^2\sigma^2\boldsymbol{\Sigma}_t^{\mathsf{dp}})\,.$$

Invoking the Hanson-Wright inequality (Lemma F.6) and union bounding over $t = 0, \ldots, T - 1$ gives $\mathbb{P}(\mathcal{E}_1^{\mathsf{dp}}) \ge 1 - p$.

**Event $\mathcal{E}_2^{\mathsf{dp}}$:** Markov's inequality gives

$$\mathbb{P}\left(\mathsf{Tr}\,[\boldsymbol{V}_{t,\tau}\boldsymbol{V}_{t,\tau}^\top] > s\right) \le \frac{1}{s}\mathbb{E}\,[\boldsymbol{V}_{t,\tau}\boldsymbol{V}_{t,\tau}^\top] \le \left(\sum_{k=0}^\tau |\beta_k|(1 - \eta\mu)^{(\tau-k)/2}\right)\frac{d}{s}$$

where we defer the technical calculations involved in bounding the expectation above to Lemma D.10. Taking a union bound over all $T(T + 1)/2 \le T^2$ choices of $(t, \tau)$ gives $\mathbb{P}(\mathcal{E}_2^{\mathsf{dp}}) \ge 1 - p$. $\qquad\square$

### D.2.2 HIGH PROBABILITY BOUNDS ON COMPONENT RECURSIONS

**Bound on the noise-less iterates:** We start with $\hat{\boldsymbol{\theta}}_t$ from (58).

**Proposition D.6.** *Under event $\mathcal{E}_1$ and if $\eta \leq (CR^2 \log(T/p))^{-1}$, we have that $\left\|\hat{\boldsymbol{\theta}}_t\right\|_2 \leq \|\boldsymbol{\theta}_0'\|_2$.*

*Proof.* Using the fact that $\mathbf{0} \preceq \boldsymbol{Q}_t \preceq \boldsymbol{I}$ under $\mathcal{E}_1$ (cf. Equation (74)), we get

$$\left\|\hat{\boldsymbol{\theta}}_t\right\|_2 = \|\boldsymbol{Q}_{t-1} \cdots \boldsymbol{Q}_0 \boldsymbol{\theta}_0'\|_2 \leq \|\boldsymbol{Q}_{t-1}\|_2 \cdots \|\boldsymbol{Q}_0\|_2 \|\boldsymbol{\theta}_0'\|_2 \leq \|\boldsymbol{\theta}_0'\|_2 .$$

$\square$

**Bound on $\widetilde{\boldsymbol{\theta}}_t^{\mathsf{sgd}}$:** We turn to $\widetilde{\boldsymbol{\theta}}_t^{\mathsf{sgd}}$ from (59).

**Proposition D.7.** *Under events $\mathcal{E}_1, \mathcal{E}_1^{\mathsf{sgd}}, \mathcal{E}_2^{\mathsf{sgd}}$, and $\eta \leq (CR^2 \log(T/p))^{-1}$, we have*

$$\left\|\widetilde{\boldsymbol{\theta}}_t^{\mathsf{sgd}}\right\|_2^2 \leq C \left(\frac{\eta R^2}{\mu}\right) \log^3 \left(\frac{T}{p}\right) .$$

*Proof.* Under $\mathcal{E}_1^{\mathsf{sgd}}$, we have

$$\left\|\widetilde{\boldsymbol{\theta}}^{\mathsf{sgd}}\right\|_2^2 \leq C\eta^2 \sigma_{\mathsf{sgd}}^2 \, \mathsf{Tr}\left[\boldsymbol{\Sigma}_t^{\mathsf{sgd}}\right] \log\left(\frac{T}{p}\right) . \tag{79}$$

We bound $\mathsf{Tr}\left[\boldsymbol{\Sigma}_t\right] = \sum_{\tau=0}^{t-1} \mathsf{Tr}\left[\boldsymbol{W}_{t,\tau}\right]$ for $\boldsymbol{W}_{t,\tau}$ defined in (75). We have two bounds for $\mathsf{Tr}\left[\boldsymbol{W}_{t,\tau}\right]$:

(a) Using $\mathbf{0} \preceq \boldsymbol{Q}_t \preceq \boldsymbol{I}$ under $\mathcal{E}_1$ (cf. Equation (74)), we bound

$$\mathsf{Tr}\left[\boldsymbol{W}_{t,\tau}\right] = \|\boldsymbol{Q}_{t-1} \cdots \boldsymbol{Q}_{\tau+1} \boldsymbol{x}_\tau\|_2^2 \leq \|\boldsymbol{Q}_{t-1}\|_2^2 \cdots \|\boldsymbol{Q}_{\tau+1}\|_2^2 \|\boldsymbol{x}_\tau\|_2^2 \leq CR^2 \log(T/p) .$$

(b) Under event $\mathcal{E}_2^{\mathsf{sgd}}$, we have the bound

$$\mathsf{Tr}\left[\boldsymbol{W}_{t,\tau}\right] \leq \frac{T^2 R^2}{p} (1 - \eta\mu)^{t-1-\tau} .$$

Using the first bound for the last $\tau \leq t - 1$ iterations and the second bound for the rest, we get

$$\mathsf{Tr}\left[\boldsymbol{\Sigma}_t^{\mathsf{sgd}}\right] \leq \sum_{k=0}^{t-\tau-1} \frac{T^2 R^2}{p} (1 - \eta\mu)^{t-1-\tau} \mathbb{1}\left(\tau < t - 1\right) + \tau \left(CR^2 \log(T/p)\right)$$

$$\leq \frac{T^2 R^2}{p} (1 - \eta\mu)^\tau \sum_{k=0}^{t-\tau-1} (1 - \eta\mu)^k \mathbb{1}\left(\tau < t - 1\right) + \tau \left(CR^2 \log(T/p)\right)$$

$$\leq \frac{T^2 R^2}{p} \frac{\exp(-\eta\mu\tau)}{\eta\mu} \mathbb{1}\left(\tau < t - 1\right) + \tau \left(CR^2 \log(T/p)\right) .$$

Choosing $\tau = \min\left\{t - 1, \frac{1}{\eta\mu} \log\left(\frac{T^2}{Cp \log(T/p)}\right)\right\}$ as per Lemma F.20 gives

$$\mathsf{Tr}\left[\boldsymbol{\Sigma}_t^{\mathsf{sgd}}\right] \leq \frac{CR^2 \log(T/p)}{\eta\mu} \left(1 + \log\left(\frac{T^2}{p \log(T/p)}\right)\right) \leq \frac{C'R^2}{\eta\mu} \log^2(T/p)$$

for some absolute constants $C, C'$. Plugging this back into (79) completes the proof. $\square$

**Bound on $\widetilde{\boldsymbol{\theta}}_t^{\mathsf{dp}}$:** We turn to $\widetilde{\boldsymbol{\theta}}_t^{\mathsf{dp}}$ from (60).

**Proposition D.8.** *Consider the setting of Theorem D.4. Under events $\mathcal{E}_1, \mathcal{E}_1^{\mathsf{dp}}, \mathcal{E}_2^{\mathsf{dp}}$, and $\eta \leq (CR^2 \log(T/p))^{-1}$, we have*

$$\left\| \widetilde{\boldsymbol{\theta}}_t^{\mathsf{sgd}} \right\|_2^2 \leq C \left( \frac{\eta R^2}{\mu} \right) \log^3 \left( \frac{T}{p} \right) .$$

*Proof.* Based on the bound on $\left\| \widetilde{\boldsymbol{\theta}}_t^{\mathsf{dp}} \right\|_2$ from $\mathcal{E}_1^{\mathsf{dp}}$, we bound $\mathsf{Tr}\left[ \boldsymbol{\Sigma}_t^{\mathsf{dp}} \right] = \sum_{\tau=0}^{t-1} \mathsf{Tr}\left[ \boldsymbol{V}_{t,\tau} \boldsymbol{V}_{t,\tau}^\top \right]$. We bound each trace on the right side in two ways:

(a) We have $\mathsf{Tr}\left[ \boldsymbol{V}_{t,\tau} \boldsymbol{V}_{t,\tau}^\top \right] \leq \|\boldsymbol{\beta}\|_1^2 d$ from Lemma D.10.

(b) Under $\mathcal{E}_2^{\mathsf{dp}}$ and the assumption $(*)$ of Half-Expo Decay of $\beta$ with parameter $\nu \leq \eta\mu$, we also have

$$\mathsf{Tr}\left[ \boldsymbol{V}_{t,\tau} \boldsymbol{V}_{t,\tau}^\top \right] \leq \frac{T^2 d}{p} \left( \sum_{\tau=0}^{\tau} |\beta_k|(1 - \eta\mu)^{(\tau-k)/2} \right)^2$$

$$\leq \frac{T^2 d}{p} \left( \sum_{\tau=0}^{\tau} |\beta_k|(1 - \nu)^{(\tau-k)/2} \right)^2$$

$$\overset{(*)}{\leq} \frac{CT^2 d}{p} (1 - \nu)^\tau .$$

Using the first bound for the first $\tau$ iterations and the second bound for the rest, we get

$$\mathsf{Tr}\left[ \boldsymbol{\Sigma}_t^{\mathsf{dp}} \right] \leq \tau \left( \|\boldsymbol{\beta}\|_1^2 d \right) + \sum_{k=\tau}^{t-1} \frac{CT^2 d}{p} (1 - \nu)^k \mathbb{1}\left( \tau > t - 1 \right)$$

$$\leq \tau \left( \|\boldsymbol{\beta}\|_1^2 d \right) + \frac{CT^2 d}{p} (1 - \nu)^\tau \sum_{k=0}^{\infty} (1 - \nu)^k \mathbb{1}\left( \tau > t - 1 \right)$$

$$\leq \tau \left( \|\boldsymbol{\beta}\|_1^2 d \right) + \frac{CT^2 d \exp(-\nu\tau)}{p\nu} \mathbb{1}\left( \tau > t - 1 \right) .$$

Choosing $\tau \leq \left\{ t - 1, \frac{1}{\nu} \log(CT^2/p\|\boldsymbol{\beta}\|_1^2) \right\}$ as per Lemma F.20, we get,

$$\mathsf{Tr}\left[ \boldsymbol{\Sigma}_t^{\mathsf{dp}} \right] \leq \frac{\|\boldsymbol{\beta}\|_1^2 d}{\nu} \left( 1 + \log\left( \frac{CT^2}{p\|\boldsymbol{\beta}\|_1^2} \right) \right) \leq C' \frac{\|\boldsymbol{\beta}\|_1^2 d}{\nu} \log\left( \frac{T}{p} \right) ,$$

where we used $\|\boldsymbol{\beta}\|_1 \geq |\beta_0| = 1$ and $C, C'$ are some universal constants. Combining this with the bound on $\left\| \widetilde{\boldsymbol{\theta}}_t^{\mathsf{dp}} \right\|_2$ asserted by $\mathcal{E}_1^{\mathsf{dp}}$ completes the proof. $\qquad \square$

### D.2.3 COMPLETING THE PROOF OF THE HIGH PROBABILITY BOUNDS

We are now ready to prove Theorem D.4.

*Proof of Theorem D.4.* Under events $\mathcal{E}_1, \mathcal{E}_1^{\mathsf{sgd}}, \mathcal{E}_2^{\mathsf{sgd}}, \mathcal{E}_1^{\mathsf{dp}}, \mathcal{E}_2^{\mathsf{dp}}$, we have bounds on the norms of $\hat{\boldsymbol{\theta}}_t, \widetilde{\boldsymbol{\theta}}_t^{\mathsf{sgd}}, \widetilde{\boldsymbol{\theta}}_t^{\mathsf{dp}}$ respectively from Propositions D.6 to D.8. We combine them with the triangle inequality and Equation (66) of Property D.2 to the claimed bound on $\|\boldsymbol{\theta}_t'\|_2$.

Next, for the gradients, we use the triangle and Cauchy-Schwarz inequalities on the definition $g_t = x_t\langle x_t, \theta'_t\rangle - x_t\xi_t$ to get

$$\|g_t\|_2^2 \leq 2\|x_t\|_2^4\|\theta'_t\|_2^2 + 2\|x_t\|_2^2|\xi_t|_2^2 .$$

Plugging in the bounds on $\|x_t\|_2$ and $|\xi|_t$ from $\mathcal{E}_1$ and $\mathcal{E}_2$ respectively gives the claimed bound on $\|g_t\|_2^2$.

Finally, all the events above hold with probability at least $1 - 6p$ from Proposition D.5. Substituting $p/6$ for $p$ and adjusting the constants completes the proof. $\qquad\square$

### D.2.4 HELPER LEMMAS

**Lemma D.9.** *Consider the setting of Theorem D.4 and consider the PSD matrices $W_{t,\tau}$, defined for $\tau \leq t-1$ as*

$$W_{t,\tau} = \begin{cases} Q_{t-1}\cdots Q_{\tau+1}(x_\tau \otimes x_\tau)Q_{\tau+1}\cdots Q_{t-1}, & \text{if } \tau < t-1, \\ x_{t-1} \otimes x_{t-1}, & \text{if } \tau = t-1. \end{cases}$$

*We have that $\mathbb{E}[\mathsf{Tr}\,[W_{t,\tau}]] \leq R^2(1-\eta\mu)^{t-1-\tau}$.*

*Proof.* For $\tau = t-1$, we have $\mathbb{E}[W_{t,t-1}] = \mathsf{Tr}\,[H] \leq R^2$. For $\tau < t-1$, we have by independence of each $x_t$ that

$$\mathsf{Tr}\,[\mathbb{E}[W_{t,\tau}]] = \mathsf{Tr}\,[\mathbb{E}[Q_{t-1}\cdots Q_{\tau+1}HQ_{\tau+1}\cdots Q_{t-1}]] = \mathsf{Tr}\,[\mathbb{E}[Q_{t-1}\cdots Q_\tau(\mathcal{P}H)Q_\tau\cdots Q_{t-1}]] = \cdots$$
$$= \mathsf{Tr}\,[\mathcal{P}^{t-1-\tau}H] .$$

Recursively bounding $\mathsf{Tr}\,[\mathcal{P}^\tau H] = \mathsf{Tr}\,[\mathcal{P}(\mathcal{P}^{\tau-1}H)] \leq (1-\eta\mu)\mathsf{Tr}\,[\mathcal{P}^{\tau-1}H]$ from Lemma F.18 completes the proof. $\qquad\square$

**Lemma D.10.** *Consider $V_{t,\tau}$ as defined in (65). We have that*

$$\mathbb{E}\left[\mathsf{Tr}\,[V_{t,\tau}V_{t,\tau}^\top]\right] \leq d\left(\sum_{k=0}^\tau |\beta_k|(1-\eta\mu)^{(\tau-k)/2}\right) .$$

*Further, if the event $\mathcal{E} = \cap_{\tau=1}^t\{Q_t \succeq 0\}$ holds, then we also have*

$$\mathsf{Tr}\,[V_{t,\tau}V_{t,\tau}^\top] \leq d\left(\sum_{k=0}^\tau |\beta_k|\right)^2 .$$

*Proof.* Since $t$ is fixed throughout, we simply write $V_{t,\tau}$ as $V_\tau$. We define a sequence of matrices $A_0, \ldots, A_\tau$ as $A_0 = \beta_0 I$ and

$$A_{k+1} = \beta_{k+1}I + Q_{t-\tau+k}A_k$$

for $k = 0, \ldots, \tau - 1$. We first prove the expected bound followed by the absolute bound.

**Expected bound:** Then, we successively deduce the following.

(a) We have $A_k = \beta_k I + \beta_{k-1}Q_{t-\tau+k-1} + \cdots + \beta_0 Q_{t-\tau+k-1}\ldots Q_{t-\tau}$ by simply unrolling the recursions.

(b) We immediately recognize that $V_\tau = A_\tau$.

(c) By independence of each $Q_t$, taking an expectation of the expression in (a) gives

$$\mathbb{E}[A_k] = \sum_{l=0}^k \beta_l(I - \eta H)^{k-l} .$$

(d) We establish a recursion

$$\mathbb{E}\mathsf{Tr}\left[\boldsymbol{A}_{k+1}\boldsymbol{A}_{k+1}^\top\right] \le d\beta_{k+1}^2 + 2d|\beta_{k+1}|\sum_{l=0}^{k}|\beta_l|(1-\eta\mu)^{k-l+1} + (1-\eta\mu)\mathbb{E}\mathsf{Tr}\left[\boldsymbol{A}_k\boldsymbol{A}_k^\top\right].$$

Indeed, by expanding out the square of the recursion and using the independence of the $\boldsymbol{x}_t$'s, we get

$$\mathbb{E}\mathsf{Tr}\left[\boldsymbol{A}_{k+1}\boldsymbol{A}_{k+1}^\top\right] = \beta_{k+1}^2\mathsf{Tr}\left[\boldsymbol{I}\right] + 2\beta_{k+1}\mathsf{Tr}\left[(\boldsymbol{I}-\eta\boldsymbol{H})\mathbb{E}[\boldsymbol{A}_k]\right] + \mathsf{Tr}\left[\mathcal{P}(\mathbb{E}[\boldsymbol{A}_k\boldsymbol{A}_k^\top])\right]$$

$$\le d\beta_{k+1}^2 + 2|\beta_{k+1}|\sum_{l=0}^{k}|\beta_l|\,\mathsf{Tr}\left[(\boldsymbol{I}-\eta\boldsymbol{H})^{k-l+1}\right] + (1-\eta\mu)\mathbb{E}\mathsf{Tr}\left[\boldsymbol{A}_k\boldsymbol{A}_k^\top\right],$$

where we plugged in the expression for $\mathbb{E}[\boldsymbol{A}_k]$ from item (c) and used Lemma F.18 to bound the last term. Using $\boldsymbol{0} \preceq \boldsymbol{I} - \eta\boldsymbol{H} \preceq (1-\eta\mu)\boldsymbol{I}$ gives the claimed expression.

(e) Using induction and the recursion from part (d), we prove that

$$\mathbb{E}\mathsf{Tr}\left[\boldsymbol{A}_k\boldsymbol{A}_k^\top\right] \le d\left(\sum_{l=0}^{k}|\beta_l|(1-\eta\mu)^{(k-l)/2}\right)^2.$$

Together with part (b), this gives the desired result.
Indeed, the base case holds because $\mathbb{E}\mathsf{Tr}\left[\boldsymbol{A}_0\boldsymbol{A}_0^\top\right] = \beta_0^2 d$. Supposing the induction hypothesis holds for some $k < \tau - 1$, we use the recursion of item (d) to get

$$\frac{1}{d}\,\mathbb{E}\mathsf{Tr}\left[\boldsymbol{A}_{k+1}\boldsymbol{A}_{k+1}^\top\right] \le \beta_{k+1}^2 + 2|\beta_{k+1}|\sum_{l=0}^{k}|\beta_l|(1-\eta\mu)^{k-l+1} + \left(\sum_{l=0}^{k}|\beta_l|(1-\eta\mu)^{\frac{k-l+1}{2}}\right)^2$$

$$\le \beta_{k+1}^2 + 2|\beta_{k+1}|\sum_{l=0}^{k}|\beta_l|(1-\eta\mu)^{\frac{k-l+1}{2}} + \left(\sum_{l=0}^{k}|\beta_l|(1-\eta\mu)^{\frac{k-l+1}{2}}\right)^2$$

$$= \left(\sum_{l=0}^{k+1}|\beta_l|(1-\eta\mu)^{\frac{k-l+1}{2}}\right)^2,$$

where the second inequality used $1 - \eta\mu \le 1$.

**Absolute bound:** Next, we prove the absolute bound, assuming that $\mathcal{E}$ holds. Again, we successively deduce:

(a) We starting with $\boldsymbol{A}_k = \beta_k\boldsymbol{I} + \beta_{k-1}\boldsymbol{Q}_{t-\tau+k-1} + \cdots + \beta_0\boldsymbol{Q}_{t-\tau+k-1}\ldots\boldsymbol{Q}_{t-\tau}$.

(b) Then, we get

$$|\mathsf{Tr}\left[\boldsymbol{A}_k\right]| \le |\beta_k|d + |\beta_{k-1}|\,|\mathsf{Tr}\left[\boldsymbol{Q}_{t-\tau+k-1}\right]| + \cdots + |\beta_0|\,|\mathsf{Tr}\left[\boldsymbol{Q}_{t-\tau+k-1}\cdots\boldsymbol{Q}_{t-\tau}\right]| \le d\sum_{l=0}^{k}|\beta_l|,$$

where we bound each of the traces by $d$ using Lemma F.19 (since we have $\boldsymbol{Q}_k \preceq \boldsymbol{I}$ under $\mathcal{E}$).

(c) By a similar logic, we get

$$\left|\mathsf{Tr}\left[\boldsymbol{Q}_{t-\tau+k}\boldsymbol{A}_k + \boldsymbol{A}_k^\top\boldsymbol{Q}_{t-\tau+k}\right]\right|$$
$$\le 2|\beta_k|\mathsf{Tr}\left[\boldsymbol{Q}_{t-\tau+k}\right] + 2|\beta_1|\,|\mathsf{Tr}\left[\boldsymbol{Q}_{t-\tau+k}\boldsymbol{Q}_{t-\tau+k-1}\right]| + \cdots + 2|\beta_0|\,|\mathsf{Tr}\left[\boldsymbol{Q}_{t-\tau+k}\cdots\boldsymbol{Q}_{t-\tau}\right]|$$
$$\le 2d\sum_{l=0}^{k}|\beta_l|.$$

(d) We prove by induction that $\mathrm{Tr}\left[\boldsymbol{A}_k \boldsymbol{A}_k^\top\right] \leq d\left(\sum_{l=0}^k |\beta_l|\right)^2$.

The base case holds since $\mathrm{Tr}\left[\boldsymbol{A}_0 \boldsymbol{A}_0^\top\right] = d\beta_0^2$. Supposing the induction hypothesis holds for some integer $1 \leq k < t - 1$, we use the recursion of $\boldsymbol{A}_{k+1}$ to calculate

$$\mathrm{Tr}\left[\boldsymbol{A}_{k+1} \boldsymbol{A}_{k+1}^\top\right] = d\beta_{k+1}^2 + \beta_{k+1}\mathrm{Tr}\left[\boldsymbol{Q}_{t-\tau+k}\boldsymbol{A}_k + \boldsymbol{A}_k^\top Q_{t-\tau+k}\right] + \mathrm{Tr}\left[\boldsymbol{Q}_{t-\tau+k}\boldsymbol{A}_k \boldsymbol{A}_k^\top \boldsymbol{Q}_{t-\tau+k}\right]$$

$$\leq d\beta_{k+1}^2 + 2d|\beta_{k+1}|\sum_{l=0}^k |\beta_l| + \mathrm{Tr}\left[\boldsymbol{A}_k \boldsymbol{A}_k^\top\right] \leq d\left(\sum_{l=0}^{k+1} |\beta_l|\right)^2 .$$

Finally, item (d) together with $\boldsymbol{A}_\tau = \boldsymbol{V}_{t,\tau}$ completes the proof. □

### D.3 EXPECTED BOUNDS ON NOISY-FTRL

Our goal in this section is to prove the following finite-time convergence guarantee of Noisy-FTRL in terms of the asymptotic suboptimality.

**Theorem D.11.** *Consider problem* (22) *and suppose Assumption C.2 holds. For a given a starting iterate* $\boldsymbol{\theta}_0 \in \mathbb{R}^d$, *weights* $\boldsymbol{\beta} \in \ell^2$, *learning rate* $\eta < 1/R^2$, *consider the sequence* $(\boldsymbol{\theta}_t)_{t=0}^\infty$ *produced by the iteration* (23) *where* $\boldsymbol{w}_t \sim \mathcal{N}(\boldsymbol{0}, \sigma^2 \boldsymbol{I})$ *with* $\sigma^2 = G^2 \gamma_\infty^2(\boldsymbol{\beta})/(2\rho)$. *Then, for any* $t \geq 0$, *we have,*

$$\mathbb{E}\left[F(\boldsymbol{\theta}_t) - F(\boldsymbol{\theta}_\star)\right] \leq \left(\sqrt{\tfrac{L}{\mu}\exp(-\eta\mu t)\left(F(\boldsymbol{\theta}_0) - F(\boldsymbol{\theta}_\star)\right)} + \sqrt{F_\infty(\boldsymbol{\beta})}\right)^2 .$$

We start with some preliminary lemmas. The first lemma is about the covariance of the noise process and is a generalization of (Jain et al., 2017a, Lemma 3) to linearly correlated additive noise.

**Lemma D.12.** *Consider the sequence* $(\widetilde{\boldsymbol{\theta}}_t)_{t=0}^\infty$ *generated by Noisy-FTRL starting from* $\widetilde{\boldsymbol{\theta}}_t = \boldsymbol{\theta}_\star$ *with noise correlations* $\boldsymbol{\beta} \in \ell^2$ *and learning rate* $\eta \leq 1/R^2$. *Under Assumption C.2, we have that its covariance*

$$\boldsymbol{S}_t := \mathbb{E}\left[\left(\widetilde{\boldsymbol{\theta}}_t - \boldsymbol{\theta}_\star\right) \otimes \left(\widetilde{\boldsymbol{\theta}}_t - \boldsymbol{\theta}_\star\right)\right]$$

*satisfies: (a)* $\boldsymbol{S}_t \preceq \boldsymbol{S}_{t+1}$ *for all* $t \geq 0$, *and (b) the sequence* $(\boldsymbol{S}_t)_{t=0}^\infty$ *converges element-wise as* $t \to \infty$.

*Proof.* Recall the notation $\boldsymbol{Q}_t = \boldsymbol{I} - \eta\boldsymbol{x} \otimes \boldsymbol{x}_t$ and $\mathcal{P}\boldsymbol{M} = \mathbb{E}[\boldsymbol{Q}_t \boldsymbol{M} \boldsymbol{Q}_t]$. We use the shorthand $\widetilde{\boldsymbol{\theta}}_t' := \widetilde{\boldsymbol{\theta}}_t - \boldsymbol{\theta}_\star$. We first prove that the covariance is increasing in a PSD sense and argue that its limit exists.

**Part 1: Non-decreasing noise:** By unrolling the update equation and using $\widetilde{\boldsymbol{\theta}}_t' = \boldsymbol{0}$, we get

$$\widetilde{\boldsymbol{\theta}}_t' = \eta\left(\boldsymbol{x}_{t-1}\xi_{t-1} + \boldsymbol{Q}_{t-1}\boldsymbol{x}_{t-2}\xi_{t-2} + \cdots + \boldsymbol{Q}_{t-1}\cdots\boldsymbol{Q}_1\boldsymbol{x}_0\xi_0\right)$$
$$- \eta\left(\beta_0\boldsymbol{w}_{t-1} + (\beta_1\boldsymbol{I} + \beta_0\boldsymbol{Q}_{t-1})\boldsymbol{w}_{t-2} + \cdots + (\beta_{t-1}\boldsymbol{I} + \beta_{t-2}\boldsymbol{Q}_{t-1} + \cdots + \beta_0\boldsymbol{Q}_{t-1}\cdots\boldsymbol{Q}_1)\boldsymbol{w}_0\right).$$
$$(80)$$

Next, we calculate $\mathbb{E}\left[\widetilde{\boldsymbol{\theta}}_t' \otimes \widetilde{\boldsymbol{\theta}}_t'\right]$. By independence, all the cross terms cancel out, so it suffices to write out the second moment of each of the terms above. For the SGD noise terms that contain $\boldsymbol{x}_\tau\xi_\tau$, we get for $\tau = 0, \ldots, t-1$ that

$$\mathbb{E}\left[(\boldsymbol{Q}_{t-1}\cdots\boldsymbol{Q}_{t-\tau+1}\boldsymbol{x}_{t-\tau}\xi_{t-\tau}) \otimes (\boldsymbol{Q}_{t-1}\cdots\boldsymbol{Q}_{t-\tau+1}\boldsymbol{x}_{t-\tau}\xi_{t-\tau})\right] = \mathcal{P}^\tau\left(\mathbb{E}[\xi^2\boldsymbol{x} \otimes \boldsymbol{x}]\right) =: \mathcal{T}_\tau . \quad (81)$$

Since it is a second-moment term, we have $\mathcal{T}_\tau \succeq \mathbf{0}$. For the DP noise terms, denote $\boldsymbol{x}^{\otimes 2} = \boldsymbol{x} \otimes \boldsymbol{x} = \boldsymbol{x}\boldsymbol{x}^\top$. Then, we have for $\tau = 0$ to $t-1$ that

$$\frac{1}{\sigma^2}\mathbb{E}\left((\beta_\tau \boldsymbol{I} + \beta_{\tau-1}\boldsymbol{Q}_{t-1} + \beta_{\tau-2}\boldsymbol{Q}_{t-1}\boldsymbol{Q}_{t-2} + \cdots + \beta_0\boldsymbol{Q}_{t-1}\cdots\boldsymbol{Q}_{t-\tau})\boldsymbol{w}_{t-\tau-1}\right)^{\otimes 2}$$

$$= \mathbb{E}\left(\beta_\tau \boldsymbol{I} + \beta_{\tau-1}\boldsymbol{Q}_{t-1} + \beta_{\tau-2}\boldsymbol{Q}_{t-1}\boldsymbol{Q}_{t-2} + \cdots + \beta_0\boldsymbol{Q}_{t-1}\cdots\boldsymbol{Q}_{t-\tau}\right)^{\otimes 2}$$

$$= \beta_\tau^2 \boldsymbol{I} + 2\beta_\tau \sum_{k=0}^{\tau-1}\beta_k(\boldsymbol{I} - \eta\boldsymbol{H})^{\tau-k} + \sum_{k=0}^{\tau-1}\sum_{l=0}^{\tau-1}\beta_k\beta_l\,\mathbb{E}\left[\boldsymbol{Q}_{t-1}\cdots\boldsymbol{Q}_{t-\tau+k}\boldsymbol{Q}_{t-\tau+l}\cdots\boldsymbol{Q}_{t-1}\right]$$

$$= \beta_\tau^2 \boldsymbol{I} + 2\beta_\tau \sum_{k=0}^{\tau-1}\beta_k(\boldsymbol{I} - \eta\boldsymbol{H})^{\tau-k} + 2\sum_{k=0}^{\tau-1}\sum_{l=0}^{k}\beta_k\beta_l\,\mathbb{E}\left[\boldsymbol{Q}_{t-1}\cdots\boldsymbol{Q}_{t-\tau+l}(\boldsymbol{I} - \eta\boldsymbol{H})^{k-l}\boldsymbol{Q}_{t-\tau+l}\cdots\boldsymbol{Q}_{t-1}\right]$$

$$= \beta_\tau^2 \boldsymbol{I} + 2\beta_\tau \sum_{k=0}^{\tau-1}\beta_k(\boldsymbol{I} - \eta\boldsymbol{H})^{\tau-k} + 2\sum_{k=0}^{\tau-1}\sum_{l=0}^{k}\beta_k\beta_l\,\mathcal{P}^{\tau-k}\left((\boldsymbol{I} - \eta\boldsymbol{H})^{k-l}\right) =: \mathcal{T}_\tau'. \tag{82}$$

By this being a second moment, we have that $\mathcal{T}_\tau' \succeq \mathbf{0}$. Plugging in (81) and (82) into the second moment of (80), we get,

$$\mathbb{E}\left[\widetilde{\boldsymbol{\theta}}_{t+1}' \otimes \widetilde{\boldsymbol{\theta}}_{t+1}'\right] = \eta^2 \sum_{\tau=0}^{t}(\mathcal{T}_\tau + \sigma^2\mathcal{T}_\tau')$$

$$= \mathbb{E}\left[\widetilde{\boldsymbol{\theta}}_t' \otimes \widetilde{\boldsymbol{\theta}}_t'\right] + \eta^2(\mathcal{T}_t + \sigma^2\mathcal{T}_t') \succeq \mathbb{E}\left[\widetilde{\boldsymbol{\theta}}_t' \otimes \widetilde{\boldsymbol{\theta}}_t'\right].$$

This shows that the noise is non-decreasing in a PSD sense.

**Part 2: Convergence of the covariance:** Next, we show that the noise sequence converges. From the update equation $\widetilde{\boldsymbol{\theta}}_{t+1}' = \boldsymbol{Q}_t\widetilde{\boldsymbol{\theta}}_t' + \eta\boldsymbol{x}_t\xi_t - \eta\sum_{\tau=0}^{t}\beta_\tau\boldsymbol{w}_{t-\tau}$, we get

$$\boldsymbol{S}_{t+1} = \mathcal{P}\boldsymbol{S}_t + \eta^2\mathbb{E}[\xi^2\boldsymbol{x}\otimes\boldsymbol{x}] + \eta^2\sigma^2\sum_{\tau=0}^{t}\beta_\tau^2\boldsymbol{I}$$

$$- \eta(\boldsymbol{I} - \eta\boldsymbol{H})\sum_{\tau=0}^{t}\beta_\tau\mathbb{E}\left[\widetilde{\boldsymbol{\theta}}_t' \otimes \boldsymbol{w}_{t-\tau}\right] - \eta\sum_{\tau=0}^{t}\beta_\tau\mathbb{E}\left[\boldsymbol{w}_{t-\tau} \otimes \widetilde{\boldsymbol{\theta}}_t'\right](\boldsymbol{I} - \eta\boldsymbol{H}).$$

For $\tau = 0$, the term $\mathbb{E}[\widetilde{\boldsymbol{\theta}}_t' \otimes \boldsymbol{w}_{t-\tau}]$ and its transpose are both $\mathbf{0}$. For $\tau > 0$, we have from (80) that

$$-\mathbb{E}\left[\widetilde{\boldsymbol{\theta}}_t' \otimes \boldsymbol{w}_{t-\tau}\right] = \eta\mathbb{E}\left[\beta_{\tau-1}\boldsymbol{I} + \beta_{\tau-2}\boldsymbol{Q}_{t-1} + \cdots + \beta_0\boldsymbol{Q}_{t-1}\cdots\boldsymbol{Q}_{t-\tau+1}\right]\mathbb{E}[\boldsymbol{w}_{t-\tau} \otimes \boldsymbol{w}_{t-\tau}]$$

$$= \eta\sigma^2\left(\beta_{\tau-1}\boldsymbol{I} + \beta_{\tau-2}(\boldsymbol{I} - \eta\boldsymbol{H}) + \cdots + \beta_0(\boldsymbol{I} - \eta\boldsymbol{H})^{\tau-1}\right).$$

Plugging this back in gives

$$\boldsymbol{S}_{t+1} = \mathcal{P}\boldsymbol{S}_t + \eta^2\mathbb{E}[\xi^2\boldsymbol{x}\otimes\boldsymbol{x}] + \eta^2\sigma^2\sum_{\tau=0}^{t}\beta_\tau^2\boldsymbol{I} + 2\eta^2\sigma^2\sum_{\tau=1}^{t}\sum_{k=0}^{\tau-1}\beta_\tau\beta_k(\boldsymbol{I} - \eta\boldsymbol{H})^{\tau-k}$$

$$= \mathcal{P}\boldsymbol{S}_t + \eta^2\mathbb{E}[\xi^2\boldsymbol{x}\otimes\boldsymbol{x}] + \eta^2\sigma^2\sum_{\tau=0}^{t}\sum_{k=0}^{t}\beta_\tau\beta_k(\boldsymbol{I} - \eta\boldsymbol{H})^{|\tau-k|}. \tag{83}$$

Next, we take a trace of (83). For the first term, we get

$$\mathsf{Tr}\left[\mathcal{P}\boldsymbol{S}_t\right] = \mathsf{Tr}\left[\boldsymbol{S}_t\right] - 2\eta\mathsf{Tr}\left[\boldsymbol{H}\boldsymbol{S}_t\right] + \eta^2\mathsf{Tr}\left[\boldsymbol{S}_t\mathbb{E}[\|\boldsymbol{x}_t\|_2^2\boldsymbol{x}_t\otimes\boldsymbol{x}_t]\right]$$

$$\leq \mathsf{Tr}\left[\boldsymbol{S}_t\right] - \eta\mathsf{Tr}\left[\boldsymbol{H}\boldsymbol{S}_t\right](2 - \eta R^2)$$

$$\leq (1 - \eta\mu)\mathsf{Tr}\left[\boldsymbol{S}_t\right],$$

where we use (a) $\mathbb{E}[\|\boldsymbol{x}_t\|_2^2\boldsymbol{x}_t\otimes\boldsymbol{x}_t] \preceq R^2\boldsymbol{H}$, (b) $\eta \leq 1/R^2$, and (c) $\boldsymbol{H} \succeq \mu\boldsymbol{I}$. By assumption, we also get that $\mathsf{Tr}\left[\mathbb{E}[\xi^2\boldsymbol{x}\otimes\boldsymbol{x}]\right] \leq \sigma_{\mathsf{sgd}}^2\mathsf{Tr}\left[\boldsymbol{H}\right] \leq \sigma_{\mathsf{sgd}}^2 R^2$. Finally, we have using Lemma F.17 that

$$\sum_{\tau=0}^{t}\sum_{k=0}^{t}\beta_\tau\beta_k\sum_{j=1}^{d}(1 - \eta\lambda_j)^{|\tau-k|} \leq \|\boldsymbol{\beta}\|_2^2\sum_{j=1}^{d}\left(\frac{2 - \eta\lambda_j}{\eta\lambda_j}\right) \leq \frac{2\|\boldsymbol{\beta}\|_2^2\mathsf{Tr}\left[\boldsymbol{H}^{-1}\right]}{\eta}.$$

Thus, we get

$$\mathsf{Tr}\left[\boldsymbol{S}_{t+1}\right] \leq (1 - \eta\mu)\mathsf{Tr}\left[\boldsymbol{S}_t\right] + 2\eta\sigma^2\|\boldsymbol{\beta}\|_2^2\,\mathsf{Tr}\left[\boldsymbol{H}^{-1}\right] + \eta^2 R^2\sigma_{\mathsf{sgd}}^2.$$

By unrolling this out, we get a uniform bound for all $t$:

$$\mathsf{Tr}\left[\boldsymbol{S}_t\right] \leq \frac{1}{\mu}\left(2\sigma^2\|\boldsymbol{\beta}\|_2^2\,\mathsf{Tr}\left[\boldsymbol{H}^{-1}\right] + \eta R^2\sigma_{\mathsf{sgd}}^2\right) < \infty$$

since $\boldsymbol{\beta} \in \ell^2$. For any fixed vector $\boldsymbol{v}$, $\langle\boldsymbol{v}, \boldsymbol{S}_t\boldsymbol{v}\rangle$ thus has a limit from the monotone convergence theorem. From this, it follows that every diagonal entry of $\boldsymbol{S}_t$ converges (take $\boldsymbol{v}$ as a standard basis vector) and then every off-diagonal entry of $\boldsymbol{S}_t$ also converges (take $\boldsymbol{v}$ as the sum of two standard basis vectors). This shows that $\boldsymbol{S}_t$ converges element-wise. $\qquad\square$

We are now ready to prove Theorem D.11.

*Proof of Theorem D.11.* Define $F_\infty^\star(\boldsymbol{\beta})$ as the asymptotic suboptimality of a process that starts from $\boldsymbol{\theta}_0 = \boldsymbol{\theta}_\star$. We will prove the desired result with $F_\infty^\star(\boldsymbol{\beta})$ in the place of $F_\infty(\boldsymbol{\beta})$. Finally, we will show that $F_\infty(\boldsymbol{\beta})$ is independent of its starting iterate so $F_\infty(\boldsymbol{\beta}) = F_\infty^\star(\boldsymbol{\beta})$.

We first separate the effects of the noise and the initial iterate using Property D.2. We invoke Lemma D.12 for the former and directly bound the latter. Lastly, we combine them both with a triangle inequality. Recall that use the shorthand $\boldsymbol{\theta}_t' := \boldsymbol{\theta}_t - \boldsymbol{\theta}_\star$ and $\boldsymbol{Q}_t := \boldsymbol{I} - \eta\boldsymbol{x}_t\otimes\boldsymbol{x}_t$.

**Effect of the initialization:** We first calculate

$$\mathbb{E}[\boldsymbol{Q}_t^2] = \boldsymbol{I} - 2\eta\boldsymbol{H} + \eta^2\mathbb{E}\left[\|\boldsymbol{x}_t\|_2^2\boldsymbol{x}_t\otimes\boldsymbol{x}_t\right] \preceq \boldsymbol{I} - 2\eta\boldsymbol{H} + \eta^2 R^2\boldsymbol{H} \preceq \boldsymbol{I} - \eta\boldsymbol{H} \preceq (1 - \eta\mu)\boldsymbol{I},$$

where the first inequality follows from (56), the second since $\eta \leq 1/R^2$, and the third since $\boldsymbol{H} \succeq \mu\boldsymbol{I}$. Letting $\mathcal{F}_t$ denote the sigma algebra generated by $\boldsymbol{x}_0, \ldots, \boldsymbol{x}_{t-1}$, we get

$$\mathbb{E}\left[\left\|\hat{\boldsymbol{\theta}}_{t+1}\right\|_2^2\Big|\mathcal{F}_t\right] = \left\langle\hat{\boldsymbol{\theta}}_t, \mathbb{E}[\boldsymbol{Q}_t^2]\hat{\boldsymbol{\theta}}_t\right\rangle \leq (1 - \eta\mu)\left\|\hat{\boldsymbol{\theta}}_t\right\|_2^2 \leq \exp(-\eta\mu)\left\|\hat{\boldsymbol{\theta}}_t\right\|_2^2.$$

Taking an unconditional expectation and unrolling this and using $\mu\boldsymbol{I} \preceq \boldsymbol{H} \preceq L\boldsymbol{I}$ (Assumption (B1)) gives

$$\mathbb{E}\left\|\hat{\boldsymbol{\theta}}_t\right\|_{\boldsymbol{H}}^2 \leq L\,\mathbb{E}\left\|\hat{\boldsymbol{\theta}}_t\right\|_2^2 \leq L\,\exp(-\eta\mu t)\,\|\boldsymbol{\theta}_0'\|_2^2 \leq \frac{L}{\mu}\,\exp(-\eta\mu t)\,\|\boldsymbol{\theta}_0'\|_{\boldsymbol{H}}^2. \tag{84}$$

**Effect of the noise:** Define $\widetilde{\boldsymbol{\theta}}_t' := \widetilde{\boldsymbol{\theta}}_t^{\mathsf{sgd}} + \widetilde{\boldsymbol{\theta}}_t^{\mathsf{dp}}$. We get from Lemma D.12 that there exists a PSD matrix $\boldsymbol{S}_\infty$ such that

$$\boldsymbol{0} = \mathbb{E}\left[\widetilde{\boldsymbol{\theta}}_0'\otimes\widetilde{\boldsymbol{\theta}}_0'\right] \preceq \mathbb{E}\left[\widetilde{\boldsymbol{\theta}}_1'\otimes\widetilde{\boldsymbol{\theta}}_1'\right] \preceq \cdots \preceq \lim_{t\to\infty}\mathbb{E}\left[\widetilde{\boldsymbol{\theta}}_t'\otimes\widetilde{\boldsymbol{\theta}}_t'\right] =: \boldsymbol{S}_\infty.$$

Multiplying by $\boldsymbol{H}$ and taking a trace, we get,

$$0 \le \mathbb{E}\left\|\widetilde{\boldsymbol{\theta}}_0'\right\|_{\boldsymbol{H}}^2 \le \mathbb{E}\left\|\widetilde{\boldsymbol{\theta}}_1'\right\|_{\boldsymbol{H}}^2 \le \cdots \le \lim_{t\to\infty} \mathbb{E}\left\|\widetilde{\boldsymbol{\theta}}_t'\right\|_{\boldsymbol{H}}^2 = \mathsf{Tr}\left[\boldsymbol{H}\boldsymbol{S}_\infty\right]. \tag{85}$$

Thus, $\widetilde{\boldsymbol{\theta}}_t = \widetilde{\boldsymbol{\theta}}_t' + \boldsymbol{\theta}_\star$ is a process that starts from $\widetilde{\boldsymbol{\theta}}_0 = \boldsymbol{\theta}_\star$ and satisfies the conditions of Lemma D.12. This in turn gives

$$0 \le \mathbb{E}\left[F(\widetilde{\boldsymbol{\theta}}_0) - F(\boldsymbol{\theta}_\star)\right] \le \mathbb{E}\left[F(\widetilde{\boldsymbol{\theta}}_1) - F(\boldsymbol{\theta}_\star)\right] \le \cdots \le \lim_{t\to\infty} \mathbb{E}\left[F(\widetilde{\boldsymbol{\theta}}_t) - F(\boldsymbol{\theta}_\star)\right] = \frac{1}{2}\mathsf{Tr}\left[\boldsymbol{H}\boldsymbol{S}_\infty\right], \tag{86}$$

which equals $F_\infty^\star(\boldsymbol{\beta})$ by definition.

**Combining both processes:** From the triangle inequality of the norm $\boldsymbol{u} \mapsto \sqrt{\mathbb{E}\|\boldsymbol{u}\|_{\boldsymbol{H}}^2}$, we get

$$\sqrt{\mathbb{E}\|\boldsymbol{\theta}_t'\|_{\boldsymbol{H}}^2} \le \sqrt{\mathbb{E}\left\|\hat{\boldsymbol{\theta}}_t\right\|_{\boldsymbol{H}}^2} + \sqrt{\mathbb{E}\left\|\widetilde{\boldsymbol{\theta}}_t'\right\|_{\boldsymbol{H}}^2}.$$

Plugging in (84) and (85) gives

$$\sqrt{\mathbb{E}\left[F(\boldsymbol{\theta}_t) - F(\boldsymbol{\theta}_\star)\right]} \le \sqrt{\frac{L}{2\mu}\exp(-\eta\mu t)\left\|\hat{\boldsymbol{\theta}}_0'\right\|_{\boldsymbol{H}}^2} + \sqrt{\frac{1}{2}\mathsf{Tr}\left[\boldsymbol{H}\boldsymbol{S}_\infty\right]}$$

$$= \sqrt{\frac{L}{\mu}\exp(-\eta\mu t)\left(F(\boldsymbol{\theta}_0) - F(\boldsymbol{\theta}_\star)\right)} + \sqrt{F_\infty^\star(\boldsymbol{\beta})},$$

where the last equality followed from (86). This establishes the required statement with $F_\infty^\star$ in place of $F^\infty$. Taking $t \to \infty$, we see that

$$\sqrt{F_\infty(\boldsymbol{\beta})} = \lim_{t\to\infty}\sqrt{\mathbb{E}\left[F(\boldsymbol{\theta}_t) - F(\boldsymbol{\theta}_\star)\right]} = \sqrt{F_\infty^\star(\boldsymbol{\beta})},$$

for any fixed $\eta$ or that $F_\infty = F_\infty^\star$ irrespective of $\boldsymbol{\theta}_0$. $\qquad\square$

## D.4 PRIVACY-UTILITY GUARANTEES OF DP-FTRL

We now state a general privacy-utility bound for DP-FTRL in terms of the asymptotics of Noisy-FTRL run with the same parameters.

**Theorem D.13.** *Fix a constant* $0 < p < 1$ *and suppose the Assumption D.1 holds. Fix some noise coefficients* $\boldsymbol{\beta} = (\beta_0, \ldots, \beta_{T-1})$ *that satisfy Half-Expo Decay with parameter* $\eta\widetilde{\nu}$ *for some* $\widetilde{\nu} \le \mu$. *Consider the sequence* $(\boldsymbol{\theta}_t)_{t=0}^{T-1}$ *of iterates and the sequence* $(\boldsymbol{g}_t)_{t=0}^{T-1}$ *of gradients when running DP-FTRL for* $T$ *iterations with noise coefficients* $\boldsymbol{\beta}$, *gradient clip norm* $G = cR^2 \max\left\{\|\boldsymbol{\theta}_0 - \boldsymbol{\theta}_\star\|_2, \sqrt{\eta R^2\sigma_{\mathsf{sgd}}^2/\mu}, \sigma_{\mathsf{sgd}}/R\right\}\log^{5/2}\left(\frac{T}{p}\right)$, *and a learning rate*

$$\eta \le \min\left\{\frac{1}{CR^2\log(T/p)}, \frac{\widetilde{\nu}\rho}{8C^2R^4 d\gamma_\infty^2(\boldsymbol{\beta})\|\boldsymbol{\beta}\|_1^2\log^5(T/p)}\right\},$$

*and DP noise* $\boldsymbol{w}_t \sim \mathcal{N}(\boldsymbol{0}, \sigma_{\mathsf{dp}}^2 G^2 \boldsymbol{I})$ *with squared noise multiplier* $\sigma_{\mathsf{dp}}^2 = \gamma(\boldsymbol{\beta})^2/(2\rho)$. *Then, we have the following:*

(a) $(\boldsymbol{\theta}_t)_{t=0}^T$ *is* $\rho$-*zCDP.*

(b) *Let $\mathcal{E}$ denote the event where no gradients are clipped, i.e, $\mathcal{E} = \cap_{t=0}^{T-1}\{\|\boldsymbol{g}_t\|_2 \le G\}$. We have, $\mathbb{P}(\mathcal{E}) \ge 1 - p$.*

(c) *We have,*

$$\mathbb{E}\left[(F(\boldsymbol{\theta}_t) - F(\boldsymbol{\theta}_\star)) \cdot \mathbb{1}(\mathcal{E})\right] \le \frac{2L}{\mu} \exp(-\eta\mu t)\left(F(\boldsymbol{\theta}_0) - F(\boldsymbol{\theta}_\star)\right) + 2\,\hat{F}_\infty(\boldsymbol{\beta})\,,$$

*where $\hat{F}_\infty(\boldsymbol{\beta})$ is the asymptotic suboptimality of Noisy-FTRL run with the same parameters.*

*Proof.* Part (a) follows from Theorem 1.1. For part (b), we bound the gradient norms from Theorem D.4 as

$$\|\boldsymbol{g}_t\|_2 \le CR^2\left(\|\boldsymbol{\theta}_0'\|_2 + \sqrt{\frac{\eta R^2 \sigma_{\mathsf{sgd}}^2}{\mu}} + \frac{\sigma_{\mathsf{sgd}}}{R} + G\sqrt{\frac{\eta \sigma^2 d\|\boldsymbol{\beta}\|_1^2}{\widetilde{\nu}}}\right)\log^{5/2}\left(\frac{T}{p}\right)$$

$$\le CR^2\left(\|\boldsymbol{\theta}_0'\|_2 + \sqrt{\frac{\eta R^2 \sigma_{\mathsf{sgd}}^2}{\mu}} + \frac{\sigma_{\mathsf{sgd}}}{R}\right)\log^{5/2}\left(\frac{T}{p}\right) + \frac{G}{4}$$

$$\le 4\max\left\{CR^2 \max\left\{\|\boldsymbol{\theta}_0'\|_2, \sqrt{\frac{\eta R^2 \sigma_{\mathsf{sgd}}^2}{\mu}}, \frac{\sigma_{\mathsf{sgd}}}{R}\right\}\log^{5/2}\left(\frac{T}{p}\right), \frac{G}{4}\right\} \le G$$

where the second inequality follows from the condition on the learning rate and we take $c = 4C$ in the definition of $G$ for the last inequality. Thus, $\mathcal{E}$ holds whenever the bound of Theorem D.4 holds, so we have $\mathbb{P}(\mathcal{E}) \ge 1 - p$.

For part (c), consider the sequence $(\boldsymbol{\phi}_t)_{t=0}^T$ produced by running Noisy-FTRL with $\boldsymbol{\phi}_0 = \boldsymbol{\theta}_0$ and the same realizations $(\boldsymbol{x}_t, \xi_t, \boldsymbol{w}_t)$ of random inputs, linear model noise, and DP noise. On $\mathcal{E}$, we have that $\boldsymbol{\phi}_t = \boldsymbol{\theta}_t$ for all $t$. Thus, we have,

$$\mathbb{E}\left[(F(\boldsymbol{\theta}_t) - F(\boldsymbol{\theta}_\star)) \cdot \mathbb{1}(\mathcal{E})\right] = \mathbb{E}\left[(F(\boldsymbol{\phi}_t) - F(\boldsymbol{\theta}_\star)) \cdot \mathbb{1}(\mathcal{E})\right] \le \mathbb{E}\left[F(\boldsymbol{\phi}_t) - F(\boldsymbol{\theta}_\star)\right]\,,$$

since $\mathbb{1}(\mathcal{E}) \le 1$. This can now be bounded using Theorem D.11 to complete the proof. $\qquad\square$

We can instantiate these rates for DP-SGD and DP-FTRL. Recall that we have $\kappa = L/\mu$, $d_{\mathsf{eff}} = \mathsf{Tr}\,[\boldsymbol{H}]/L$, and $R^2 = \Theta(\mathsf{Tr}\,[\boldsymbol{H}])$.

**Corollary D.14.** *Consider the setting of Theorem D.13 with $T$ large enough that $T/\log^5(T/p) \ge c\kappa^2 d_{\mathsf{eff}}^2 d/\rho$. The final suboptimality of DP-SGD at an appropriate choice of the learning rate is (ignoring absolute constants),*

$$\mathbb{E}\left[(F(\boldsymbol{\theta}_T) - F(\boldsymbol{\theta}_\star)) \cdot \mathbb{1}(\mathcal{E})\right] \le \frac{L}{\mu}\exp\left(-\frac{\rho T}{c\kappa^2 d_{\mathsf{eff}}^2 d\log^5(T/p)}\right)$$

$$+ \kappa\,d_{\mathsf{eff}}\left(\frac{d\mathsf{Tr}\,[\boldsymbol{H}]\|\boldsymbol{\theta}_0 - \boldsymbol{\theta}_\star\|_2^2}{\rho T} + \frac{d\sigma_{\mathsf{sgd}}^2}{\rho T} + \frac{\sigma_{\mathsf{sgd}}^2}{T}\right)\mathrm{polylog}\,(T)\,.$$

*Proof.* We plug in the asymptotic suboptimality bound of Noisy-SGD into the bound of Theorem D.13. We get two terms depending on the learning rate $\eta$: the first $\exp(-\eta\mu T)$ term and the second $O(\eta)$ term coming from the asymptotic suboptimality. We balance both the terms subject to the maximum bound on $\eta$ using

Lemma F.21 to get

$$\mathbb{E}\left[(F(\boldsymbol{\theta}_T) - F(\boldsymbol{\theta}_\star)) \cdot \mathbb{1}(\mathcal{E})\right] \leq \frac{L}{\mu} \exp\left(-\frac{\rho\mu^2 T}{cR^4 d \log^5(T/p)}\right)$$
$$+ \frac{\text{polylog}(T)}{\mu T} \left(\frac{dR^4 \|\boldsymbol{\theta}_0 - \boldsymbol{\theta}_\star\|_2^2}{\rho} + \frac{d\sigma_{\text{sgd}}^2 R^2}{\rho} + \sigma_{\text{sgd}}^2 R^2\right) .$$

Rearranging the constants completes the proof. $\qquad\square$

**Corollary D.15.** *Consider the setting of Theorem D.13 with $T$ large enough that $T/\log^7(T/p) \geq \frac{c\kappa^2 d_{\text{eff}}^2 d}{\rho} \log\left(\frac{c\kappa^2 d_{\text{eff}}^2 d}{\rho}\right)$. For $\nu$-DP-FTRL with an appropriate choice of the parameter $\nu$ and learning rate $\eta$, we have (ignoring absolute constants),*

$$\mathbb{E}\left[(F(\boldsymbol{\theta}_T) - F(\boldsymbol{\theta}_\star)) \cdot \mathbb{1}(\mathcal{E})\right] \leq \frac{L}{\mu} \exp\left(-\frac{\rho T}{c\kappa^2 d_{\text{eff}}^2 d \log^7(T/p) \log(\kappa^2 d_{\text{eff}}^2 d/\rho)}\right)$$
$$+ \kappa d_{\text{eff}} \left(\frac{\kappa d_{\text{eff}} \text{Tr}[\boldsymbol{H}] \|\boldsymbol{\theta}_0 - \boldsymbol{\theta}_\star\|_2^2}{\rho T^2} + \frac{\kappa d_{\text{eff}} \sigma_{\text{sgd}}^2}{\rho T^2} + \frac{\sigma_{\text{sgd}}^2}{T}\right) \text{polylog}(T) .$$

*Proof.* We plug in the asymptotic error for $\nu$-Noisy-FTRL from Proposition C.22 into Theorem D.13 to get that

$$\mathbb{E}\left[(F(\boldsymbol{\theta}_T) - F(\boldsymbol{\theta}_\star)) \cdot \mathbb{1}(\mathcal{E})\right] \leq \frac{L}{\mu} \exp(-\mu\eta T) + \eta\sigma_{\text{sgd}}^2 R^2 + \eta^2 \frac{R^2 G^2}{\rho} \log^2 \frac{1}{\eta\mu} , \tag{87}$$

where $G^2$ is as given in the statement of Theorem D.13. For our choice of $\boldsymbol{\beta}$, we have $\|\boldsymbol{\beta}\|_1^2 \leq 4$ always and $\gamma(\boldsymbol{\beta})^2 \leq 5\log(1/\eta\mu)$ from Equation (50) (from the proof of Proposition C.22). Thus, the largest learning rate permitted must satisfy

$$\eta \log^2 \frac{1}{\eta\mu} \leq \frac{\eta\rho}{cR^2 d \log^5(T/p)} .$$

From Lemma F.22, we can ensure with a more stringent condition

$$\eta \leq \frac{\mu\rho}{cR^4 d \log^5(T/p) \log^2(cR^4 d \log(T/p)/(\mu^2\rho))} .$$

Finally, this is implied by imposing the requirement

$$\eta \leq \frac{\mu\rho}{cR^4 d \log^7(T/p) \log\left(\frac{R^4 d}{\mu^2\rho}\right)} =: \eta_{\max} .$$

We now tune $\eta$ to minimize the bound (87) subject to $\eta \leq \eta_{\max}$ using Lemma F.21. Thus gives,

$$\mathbb{E}\left[(F(\boldsymbol{\theta}_T) - F(\boldsymbol{\theta}_\star)) \cdot \mathbb{1}(\mathcal{E})\right] \leq \frac{L}{\mu} \exp\left(-\frac{\rho\mu^2 T}{cR^4 d \log^7(T/p) \log \frac{R^4 d}{\rho\mu^2}}\right)$$
$$+ \frac{\text{polylog}(T)}{\mu T} \left(\frac{R^6 \|\boldsymbol{\theta}_0 - \boldsymbol{\theta}_\star\|_2^2}{\rho\mu T} + \frac{R^4 \sigma_{\text{sgd}}^2}{\rho\mu^2 T^2} + \sigma_{\text{sgd}}^2 R^2\right) .$$

Rewriting the constants completes the proof. $\qquad\square$

## E  PROOFS FOR GENERAL STRONGLY CONVEX FUNCTIONS

We prove the results from Theorem 3.1. Under the assumptions of the theorem, clipping does not occur in DP-FTRL so the updates can be written as

$$\boldsymbol{\theta}_{t+1} = \boldsymbol{\theta}_t - \eta \left( (\boldsymbol{B}\boldsymbol{w})_t + (\boldsymbol{g}_t + \hat{\boldsymbol{w}}_t) \right) \tag{88}$$

where

$$\boldsymbol{g}_t = \nabla F(\boldsymbol{\theta}_t), \quad \hat{\boldsymbol{w}}_t = \nabla f(\boldsymbol{\theta}_t; \boldsymbol{z}_t) - \mathbb{E}_{\boldsymbol{z} \sim \mathbb{P}_{\mathsf{data}}} \left[ \nabla f(\boldsymbol{\theta}_t; \boldsymbol{z}) \right]$$

and $\hat{\boldsymbol{w}}_t$ is a random variable that, conditioned on $\boldsymbol{\theta}_t$, is bounded by $\sigma_{\mathsf{sgd}}$ with probability 1. Below, $\boldsymbol{I}_d$ denotes the $d \times d$ identity matrix.

**Theorem E.1.** $\boldsymbol{\lambda} = \{\lambda_t\}_{t=-\infty}^{\infty}$ *be such that* $\lambda_t \geq 0 \quad \forall t \in \mathbb{Z}$,

$$\sum_{t=-\infty}^{\infty} \lambda_t \leq 2\lambda_0$$

*and let* $\Lambda$ *denote the Discrete-time Fourier transform (DTFT) of* $\boldsymbol{\lambda}$. *Let*

$$M_\lambda(\omega) = A(\omega)^{*\top} \widetilde{M}_\lambda(\omega) A(\omega) \tag{89a}$$

$$A(\omega) = \begin{pmatrix} \eta \boldsymbol{I}_d & 0 \\ (1 - \exp(i\omega)) \boldsymbol{I}_d & -\eta \boldsymbol{I}_d \end{pmatrix} \tag{89b}$$

$$\widetilde{M}_\lambda(\omega) = \begin{pmatrix} -\mu L \left( \Lambda(\omega) + \Lambda(\omega)^* \right) \boldsymbol{I}_d & \mu \Lambda(\omega) \boldsymbol{I}_d + L \Lambda(\omega)^* \boldsymbol{I}_d \\ \mu \Lambda^*(\omega) \boldsymbol{I}_d + L \Lambda(\omega) \boldsymbol{I}_d & - \left( \Lambda(\omega) + \Lambda(\omega)^* \right) \boldsymbol{I}_d \end{pmatrix} \tag{89c}$$

*Then, for any non-negative valued function* $\psi : [-\pi, \pi] \mapsto \mathbb{R}_+$ *such that*

$$M_\lambda(\omega) \preceq \begin{pmatrix} -\eta^2 \boldsymbol{I}_d & 0 \\ 0 & \psi(\omega) \boldsymbol{I}_d \end{pmatrix} \quad \forall \omega \in [-\pi, \pi] \tag{90}$$

*We have that*

$$\lim_{t \to \infty} \mathbb{E} \left[ \frac{\sum_{t=-T}^{T} \|\boldsymbol{\theta}_t - \boldsymbol{\theta}^\star\|_2^2}{2T+1} \right] \leq \frac{2d}{2\pi\eta^2} \int_{-\pi}^{\pi} \left( |B(\omega)|^2 G^2 \rho^{-1} \gamma_\infty^2(B) + \sigma_{\mathsf{sgd}}^2 \right) \psi(\omega) \, \mathrm{d}\omega$$

*where* $S_{sgd}$ *is the power spectral density of* $\widetilde{\boldsymbol{w}}$. *In particular, if the density of* $\boldsymbol{\theta}_t$ *converges to a stationary distribution, the expected value of*

$$\lim_{t \to \infty} \mathbb{E} \left[ \|\boldsymbol{\theta}_t - \boldsymbol{\theta}^\star\|_2^2 \right]$$

*under the stationary distribution is bounded as above.*

*Proof.* We assume without loss of generality that $\nabla F(0) = 0$ so that the origin is the global optimum of $F$ (else we can translate the origin to achieve this). Since $\boldsymbol{g} = \nabla F(\boldsymbol{\theta})$ satisfies

$$\langle \boldsymbol{g} - L\boldsymbol{\theta}, \mu\boldsymbol{\theta} - \boldsymbol{g} \rangle \geq 0 \quad \forall \boldsymbol{\theta}, \boldsymbol{g} \,.$$

Then, we can write down the following family of integral quadratic constraints relating $\boldsymbol{g} = (\dots, \boldsymbol{g}_0, \boldsymbol{g}_1, \boldsymbol{g}_2, \dots)$ and $\boldsymbol{\theta} = (\dots, \boldsymbol{\theta}_0, \boldsymbol{\theta}_1, \boldsymbol{\theta}_2, \dots)$ in terms of their Fourier transforms $\Theta(\omega), G(\omega)$ (Heath & Wills (2005) Eq. 27-29):

$$\int_{-\pi}^{\pi} \begin{pmatrix} \Theta(\omega) \\ G(\omega) \end{pmatrix}^* \begin{pmatrix} -\mu L \left( \Lambda(\omega) + \Lambda(\omega)^* \right) \boldsymbol{I}_d & \mu \left( \Lambda(\omega) \right) \boldsymbol{I}_d + L \left( \Lambda(\omega)^* \right) \boldsymbol{I}_d \\ \mu \left( \Lambda^*(\omega) \right) \boldsymbol{I}_d + L \left( \Lambda(\omega) \right) \boldsymbol{I}_d & - \left( \Lambda(\omega) + \Lambda(\omega)^* \right) \boldsymbol{I}_d \end{pmatrix} \begin{pmatrix} \Theta(\omega) \\ G(\omega) \end{pmatrix} \mathrm{d}\omega \geq 0 \,. \tag{91}$$

Noting that from (88), we have that

$$\Theta\left(\omega\right)\left(\exp\left(i\omega\right)-1\right)=-\eta\left(G\left(\omega\right)+Z\left(\omega\right)\right)\implies G\left(\omega\right)=\left(\frac{1-\exp\left(i\omega\right)}{\eta}\right)\Theta\left(\omega\right)-Z\left(\omega\right)$$

where $Z$ denotes the DTFT of $\boldsymbol{\zeta}=\boldsymbol{Bw}+\hat{\boldsymbol{w}}$. Plugging this into the above quadratic constraint and multiplying by $\eta^2$, we obtain

$$\int_{-\pi}^{\pi}\begin{pmatrix}\Theta\left(\omega\right)\\Z\left(\omega\right)\end{pmatrix}^*M_\lambda\left(\omega\right)\begin{pmatrix}\Theta\left(\omega\right)\\Z\left(\omega\right)\end{pmatrix}\mathrm{d}\omega\geq 0\,. \tag{92}$$

Since $M_\lambda\left(\omega\right)\preceq\begin{pmatrix}-\eta^2\boldsymbol{I}_d & 0\\0 & \psi\left(\omega\right)\boldsymbol{I}_d\end{pmatrix}$ we obtain that

$$\int_{-\pi}^{\pi}\begin{pmatrix}\Theta\left(\omega\right)\\Z\left(\omega\right)\end{pmatrix}^*\begin{pmatrix}-\eta^2\boldsymbol{I}_d & 0\\0 & \psi\left(\omega\right)\end{pmatrix}\begin{pmatrix}\Theta\left(\omega\right)\\Z\left(\omega\right)\end{pmatrix}\mathrm{d}\omega\geq 0\implies\frac{\mathbb{E}\left[\int_{-\pi}^{\pi}\|\Theta\left(\omega\right)\|^2\right]}{\mathbb{E}\left[\int_{-\pi}^{\pi}\left\|\sqrt{\psi\left(\omega\right)}Z\left(\omega\right)\right\|^2\right]}\leq 1$$

$$\implies\quad\frac{\lim_{T\to\infty}\mathbb{E}\left[\frac{\sum_{t=-T}^{T}\|\boldsymbol{\theta}_t\|^2}{2T+1}\right]}{\lim_{T\to\infty}\mathbb{E}\left[\frac{\sum_{t=-T}^{T}\left\|\sqrt{\psi}[\boldsymbol{\zeta}](t)\right\|^2}{2T+1}\right]}\leq\frac{1}{\eta^2}$$

where $\sqrt{\zeta}[z]$ denotes the LTI operator with transfer function $\sqrt{\zeta\left(\omega\right)}$ applied to the signal $\boldsymbol{\zeta}$.

The denominator of the final line above is the power spectral density of $\sqrt{\kappa}[\boldsymbol{\zeta}]$ (since $\sqrt{\kappa}[\boldsymbol{\zeta}]$ is a wide-sense stationary stochastic process). By the Cauchy-Schwarz inequality for random variables, this is bounded above by

$$2d\left(|B\left(\omega\right)|^2\rho^{-1}\gamma_\infty^2\left(B\right)+\sigma_{\mathsf{sgd}}^2\right)\psi\left(\omega\right)$$

where the first term in brackets is the power spectral density of the Gaussian random process $\boldsymbol{Bw}$ and the second term is an upper bound on the power spectral density of $\hat{\boldsymbol{w}}$. Hence, by Theorem F.2, we have the desired result. $\qquad\square$

### E.1 Proof of Theorem 3.1

Given the above theorem and smooth convexity parameter $L$, we know that the asymptotic suboptimality $F_\infty$ is bounded above by

$$\frac{2Ld}{2\pi\eta^2}\int_{-\pi}^{\pi}\left(|B\left(\omega\right)|^2\rho^{-1}\gamma_\infty^2\left(B\right)G^2+\sigma_{\mathsf{sgd}}^2\right)\psi\left(\omega\right)\mathrm{d}\omega\,.$$

Now, the constraint (90) can be rewritten as

$$\begin{pmatrix}-\eta^2 & 0\\0 & \psi\left(\omega\right)\end{pmatrix}-$$

$$\begin{pmatrix}\eta & 0\\1-\exp\left(i\omega\right) & -\eta\end{pmatrix}^{*\top}\begin{pmatrix}-\mu L\left(\Lambda\left(\omega\right)+\Lambda\left(\omega\right)^*\right) & \mu\Lambda\left(\omega\right)+L\Lambda\left(\omega\right)^*\\\mu\Lambda^*\left(\omega\right)+L\Lambda\left(\omega\right) & -\left(\Lambda\left(\omega\right)+\Lambda\left(\omega\right)^*\right)\end{pmatrix}\begin{pmatrix}\eta & 0\\1-\exp\left(i\omega\right) & -\eta\end{pmatrix}\succeq 0 \tag{93}$$

since all the matrices involved are Hadamard products of the $2\times 2$ matrices above and the identity matrix.

Thus, for each $\omega$, $\psi(\omega)$ must satisfy a $2 \times 2$ PSD constraint which can be rewritten as a Second Order Cone Program (SOCP) constraint. Furthermore, the constraint on $\lambda$ from theorem E.1 is a linear constraint. Since the projection of a convex set in $\psi, \lambda$ to $\psi$ is convex, $\psi$ belongs to a convex set. Furthermore, if we take $\lambda$ such that $\lambda_\tau = 0$ for $|\tau| > T_{\max}$ for some $T_{\max} > 0$, the constraint on $\lambda$ can be written as

$$2\lambda_0 \geq \sum_{\tau=-T_{\max}}^{T_{\max}} \lambda_t \,.$$

Further, if we discretize $\omega$ to a uniform grid on $[-\pi, \pi]$, the constraints (93) can be written as a finite collection of SOCP constraints linking $\psi(\omega)$ and $\lambda$.

## F    TECHNICAL DEFINITIONS AND LEMMAS

We review several relevant technical definitions and lemmas here:

- **Appendix F.1**: Fourier Analysis of Linear Time-Invariant Systems.
- **Appendix F.2**: Stationary covariance of SGD.
- **Appendix F.3**: Concentration of Measure.
- **Appendix F.4**: Review of definitions and useful properties of elliptic integrals.

### F.1    LINEAR TIME-INVARIANT (LTI) SYSTEMS

We first review the definition and some useful properties of discrete-time Linear Time-Invariant (LTI) systems. We refer to the textbook (Oppenheim et al., 1997) for a more detailed description.

**Definition F.1.** *An input-output system $\boldsymbol{y}_t = \mathcal{A}_t(\boldsymbol{x})$ with an input sequence $\boldsymbol{x} = (\boldsymbol{x}_t)_{t=-\infty}^{\infty}$ in some input space $\mathcal{X}$ and an output sequence $(\boldsymbol{y}_t)_{t=-\infty}^{\infty}$ in an output space $\mathcal{Y}$ is said to be LTI if it satisfies two properties:*

- **Linearity***: For any $\mathcal{X}$-valued sequences $\boldsymbol{x}^{(1)}, \boldsymbol{x}^{(2)}, \ldots$ and scalars $\alpha_1, \alpha_2, \ldots$, we have*

$$\mathcal{A}_t\left(\sum_{j=1}^{\infty} \alpha_j \boldsymbol{x}^{(j)}\right) = \sum_{j=1}^{\infty} \alpha_j \mathcal{A}_t(\boldsymbol{x}^{(j)}) \,.$$

- **Time-Invariance***: For any $t_0 \in \mathbb{Z}$, the sequence $\boldsymbol{x}'$ defined as $\boldsymbol{x}_t' := \boldsymbol{x}_{t-t_0}$ satisfies $\mathcal{A}_t(\boldsymbol{x}') = \mathcal{A}_{t-t_0}(\boldsymbol{x})$.*

Throughout this paper, we consider LTI systems in the Euclidean space $\mathcal{X} = \mathbb{R}^d$.

LTI systems can be viewed as linear operators defined on the Hilbert space of *signals* in $\mathbb{R}^d$:

$$\ell_{2e}^d = \left\{(\boldsymbol{x}_t)_{t=-\infty}^{\infty} : \quad \boldsymbol{x}_t \in \mathbb{R}^d \quad \text{and} \quad \sum_{\tau=-t}^{t} \|\boldsymbol{x}_\tau\|_2^2 < \infty \quad \forall t \in \mathbb{Z}\right\} \,.$$

We use the notation $\overrightarrow{\boldsymbol{x}} = (\boldsymbol{x}_t)_{t=-\infty}^{\infty} \in \ell_{2e}^d$ to denote an entire sequence. The Hilbert space $\ell_{2e}^d$ is endowed with the inner product $\langle \overrightarrow{\boldsymbol{x}}, \overrightarrow{\boldsymbol{y}} \rangle = \sum_{t=-\infty}^{\infty} \boldsymbol{x}_t^\top \boldsymbol{y}_t$.

**Asymptotic stability:** An LTI system is said to be asymptotically stable if its output decays to zero for any input sequence that is bounded, i.e., for which there exists $T > -\infty$ such that $\boldsymbol{x}_t = 0 \quad \forall t > T$.

**LTI systems in 1D:** We highlight some key properties of LTI systems in $d = 1$ dimension, i.e. $\mathcal{X} = \mathbb{R}$. This conveys the key ideas before we describe the extension in higher dimensions. LTI systems can be described

in linear algebraic notation by the action of an infinite Toeplitz matrix $\boldsymbol{H} = \text{Toeplitz}(\boldsymbol{h})$ (i.e., the first column of $\boldsymbol{H}$ is $\boldsymbol{h}$) on an element $\overrightarrow{\boldsymbol{x}} \in \ell_{2e}$:

$$\overrightarrow{\boldsymbol{y}} = \boldsymbol{H} \overrightarrow{\boldsymbol{x}} \iff y_t = \sum_{\tau=-\infty}^{\infty} \boldsymbol{H}_{t,\tau} x_\tau = \left(\boldsymbol{h} \star \overrightarrow{\boldsymbol{x}}\right)_t \quad \forall t \in \mathbb{Z}$$

where $\star$ denotes the convolution operator. This property is represented more elegantly in the Fourier domain. Consider the discrete-time Fourier transform (DTFT) $X : [-\pi, \pi] \to \mathbb{C}$ of $\overrightarrow{\boldsymbol{x}}$, defined by

$$X(\omega) = \sum_{t=-\infty}^{\infty} x_t \exp(-i\omega t).$$

Similarly, let $Y(\omega)$ denote the DTFT of $\overrightarrow{\boldsymbol{y}}$ and $G(\omega)$[7] denote the DTFT of $\boldsymbol{h}$. Then, we have $Y(\omega) = G(\omega)X(\omega)$. Here, $\boldsymbol{h}$ is known as the **impulse response** and $G(\omega)$ is known as the **transfer function**.

**Multivariate LTI systems:** The previous concepts can be directly extended to higher dimensions and multivariate LTI systems admit a clean representation in the Fourier domain.

Let $\boldsymbol{x}_t \in \mathbb{R}^d$ be the input and $\boldsymbol{y}_t \in \mathbb{R}^p$ be the output of an LTI system. The DTFT $\boldsymbol{X}(\omega) = \sum_{t=-\infty}^{\infty} \boldsymbol{x}_t \exp(-i\omega t) \in \mathbb{C}^d$ outputs a $d$-dimensional complex vector, and $\boldsymbol{Y}(\omega) \in \mathbb{C}^p$ similarly.

The transfer function $\boldsymbol{G}(\omega)$ in this case can be represented as a complex matrix in $\mathbb{C}^{p \times d}$. Similar to the scalar case, the Fourier domain description of this LTI system is given as $\boldsymbol{Y}(\omega) = \boldsymbol{G}(\omega)\boldsymbol{X}(\omega)$, where the latter product is the standard matrix-vector product over complex numbers.

**Variance of LTI systems driven by white noise:** The Fourier-domain analysis of an LTI system (particularly its transfer function) helps us characterize the covariance of the output $\boldsymbol{y}_t$ as a function of the covariance of the input $\boldsymbol{x}_t$. The following theorem presents the result for multivariate LTI systems driven by white noise.

**Theorem F.2.** *Consider an asymptotically-stable LTI system with $\mathbb{R}^d$-valued inputs $(\boldsymbol{x}_t)_{t=-\infty}^{\infty}$ and $\mathbb{R}^p$-valued outputs $(\boldsymbol{y}_t)_{-\infty}^{\infty}$ and a transfer function $\boldsymbol{G}(\omega) \in \mathbb{C}^{p \times d}$. Suppose that $\boldsymbol{x}_t$ is a stationary white noise sequence with covariance matrix $\boldsymbol{\Sigma} \in \mathbb{R}^{d \times d}$, i.e., $\mathbb{E}[\boldsymbol{x}_t] = \boldsymbol{0}$ and $\mathbb{E}[\boldsymbol{x}_t \otimes \boldsymbol{x}_\tau] = \boldsymbol{\Sigma}$ if $t = \tau$ and $\boldsymbol{0}_{d \times d}$ otherwise for all $t, \tau$. Then, we have for all $t > -\infty$ that*

$$\mathbb{E}[\boldsymbol{y}_t \otimes \boldsymbol{y}_t] = \frac{1}{2\pi} \int_{-\pi}^{\pi} \boldsymbol{G}(\omega) \, \boldsymbol{\Sigma} \, \boldsymbol{G}(\omega)^* \, \mathrm{d}\omega.$$

## F.2 Stationary Covariance of Stochastic Gradient Descent for Linear Regression

We now give a result characterizing the stationary covariance of SGD for linear regression (Bach & Moulines, 2013; Défossez & Bach, 2015; Jain et al., 2017b;a).

**Theorem F.3** (Lemma 5 of (Jain et al., 2017a))**.** *Consider the recursion $\boldsymbol{\delta}_0 = \boldsymbol{0}$ and*

$$\boldsymbol{\delta}_{t+1} = (\boldsymbol{I} - \eta \boldsymbol{x}_t \otimes \boldsymbol{x}_t) \, \boldsymbol{\delta}_t + \eta \boldsymbol{\zeta}_t,$$

*for all $t \geq 0$ where*

- *$\boldsymbol{x}_t$ are i.i.d. with mean $\boldsymbol{0}$, covariance $\boldsymbol{H}$, and*
- *$\boldsymbol{\zeta}_t$ are i.i.d. with mean $\boldsymbol{0}$, covariance $\mathbb{E}[\boldsymbol{\zeta}_t \otimes \boldsymbol{\zeta}_t] \preceq \sigma^2 \boldsymbol{H}$.*

*Further, if $\mathbb{E}\left[\|\boldsymbol{x}_t\|_2^2 \, (\boldsymbol{x}_t \otimes \boldsymbol{x}_t)\right] \preceq R^2 \boldsymbol{H}$ and $\eta < 1/R^2$, then we have for all $t \geq 0$.*

$$\mathbb{E}[\boldsymbol{\delta}_t \otimes \boldsymbol{\delta}_t] \preceq \frac{\eta \sigma^2}{1 - \eta R^2} \, \boldsymbol{I}.$$

---

[7]The transfer function $G(\omega)$ here is not to be confused with the clip norm $G$ used in the rest of the manuscript; this section is a self-contained technical reference.

### F.3 CONCENTRATION OF MEASURE

We recall the definition of sub-Gaussian random variables and list some useful concentration inequalities.

**Definition F.4.** *A real-valued random variable $X$ is said to be sub-Gaussian with variance proxy $\sigma^2$ if for all $\lambda \in \mathbb{R}$, we have*

$$\mathbb{E}[\exp(\lambda(X - \mu))] \leq \exp(\lambda^2 \sigma^2 / 2) ,$$

*where $\mu = \mathbb{E}[X]$. If in addition, the variance of $X$ exactly equals $\sigma^2$, it is said to be* strictly sub-Gaussian.

The cumulants of strict sub-Gaussian random variables are closely related to those of a Gaussian (Arbel et al., 2020, Prop. 3.2).

**Property F.5.** *If $X$ is strictly sub-Gaussian with mean zero and variance $\sigma^2$, we have $\mathbb{E}[X^3] = 0$ and $\mathbb{E}[X^4] \leq 3\sigma^4 = \mathbb{E}[Y^4]$ for $Y \sim \mathcal{N}(0, \sigma^2)$.*

Next, we state the **Hanson-Wright inequality** for the concentration of quadratic forms; see e.g. (Rudelson & Vershynin, 2013).

**Lemma F.6.** *Let $\boldsymbol{\xi} = (\xi_1, \ldots, \xi_d)$ be such that each $\xi_j$ is independent and sub-Gaussian with mean zero and variance proxy $\sigma^2$. Then, we have for any matrix $\boldsymbol{A} \in \mathbb{R}^{d \times d}$,*

$$\mathbb{P}(\langle \boldsymbol{\xi}, \boldsymbol{A}\boldsymbol{\xi} \rangle - \mathbb{E}[\langle \boldsymbol{\xi}, \boldsymbol{A}\boldsymbol{\xi} \rangle] > t) \leq \exp\left(-c \min\left\{\frac{t^2}{\sigma^4 \|\boldsymbol{A}\|_F^2}, \frac{t}{\sigma^2 \|\boldsymbol{A}\|_2}\right\}\right) ,$$

*for a universal constant $c$. Consequently, for any $\rho < 1/3$ and symmetric PSD matrix $\boldsymbol{A}$, we have with probability $1 - \rho$ that*

$$\langle \boldsymbol{\xi}, \boldsymbol{A}\boldsymbol{\xi} \rangle \leq C\sigma^2 \left(\mathsf{Tr}\,[\boldsymbol{A}] \sqrt{\log\frac{1}{\rho}} + \|\boldsymbol{A}\|_2 \log\frac{1}{\rho}\right) \leq C'\sigma^2 \mathsf{Tr}\,[\boldsymbol{A}] \log\frac{1}{\rho} ,$$

*for universal constants $C, C'$.*

The second part follows from the first one under the simplifications $\|\boldsymbol{A}\|_2 \leq \|\boldsymbol{A}\|_F \leq \mathsf{Tr}\,[\boldsymbol{A}]$ and $\mathbb{E}[\langle \boldsymbol{\xi}, \boldsymbol{A}\boldsymbol{\xi} \rangle] \leq \sigma^2 \mathsf{Tr}\,[\boldsymbol{A}]$ for $\boldsymbol{A}$ PSD.

**Remark F.7.** *Explicit values for the constant $c$ in Lemma F.6 (and thus for $C, C'$) are known for the case when $\xi_1, \ldots, \xi_d \sim \mathcal{N}(0, \sigma^2)$: $c \approx 0.1457 \geq 1/8$, $C \leq 8$, $C' \leq 16$ (Moshksar, 2021).*

### F.4 REVIEW OF ELLIPTIC INTEGRALS

We recall some definitions and useful properties of elliptic integrals. We refer to (NIS, §19) and (Byrd & Friedman, 2013) for details.

The three canonical elliptic integral forms are:

(i) The complete elliptic integral of the first kind $K : (0, 1) \to [0, \infty)$ is

$$K(k) := \int_0^{\pi/2} \frac{\mathrm{d}\omega}{\sqrt{1 - k^2 \sin^2(\omega)}} . \tag{94}$$

(ii) The complete elliptic integral of the second kind $E : (0, 1) \to [0, \infty)$ is

$$E(k) := \int_0^{\pi/2} \sqrt{1 - k^2 \sin^2(\omega)} \, \mathrm{d}\omega . \tag{95}$$

(iii) The complete elliptic integral of the third kind $\Pi : (\mathbb{R} \setminus \{\pm 1\}) \times (0, 1) \to \mathbb{R}$ is denoted conventionally as $\Pi(\alpha^2, k)$ where $\alpha^2$ is allowed to take negative values. It is defined as

$$\Pi(\alpha^2, k) := \int_0^{\pi/2} \frac{\mathrm{d}\omega}{(1 - \alpha^2 \sin^2(\omega))\sqrt{1 - k^2 \sin^2(\omega)}} \ . \tag{96}$$

The corresponding integrals where $1 - k^2 \sin^2(\omega)$ is replaced with $1 + k^2 \sin^2(\omega)$ can also be expressed using the elliptic integrals (NIS, Eq. (19.7.2), (19.7.5)).

**Property F.8.** *For any $m \in (0, 1)$, we have*

$$\int_0^{\pi/2} \frac{\mathrm{d}\omega}{\sqrt{1 + m \sin^2(\omega)}} = \frac{1}{\sqrt{1 + m}} \ K\left(\sqrt{\frac{m}{1 + m}}\right) \ . \tag{97}$$

**Property F.9.** *For any $m \in (0, 1)$ and any $\alpha^2 \in \mathbb{R} \setminus \{\pm 1\}$ such that $\alpha^2 + m \neq 0$, we have*

$$\int_0^{\pi/2} \frac{\mathrm{d}\omega}{(1 - \alpha^2 \sin^2(\omega))\sqrt{1 + m \sin^2(\omega)}}$$
$$= \frac{m}{(m + \alpha^2)\sqrt{1 + m}} \ K\left(\sqrt{\frac{m}{1 + m}}\right) + \frac{\alpha^2}{(m + \alpha^2)\sqrt{1 + m}} \Pi\left(\frac{m + \alpha^2}{1 + m}, \sqrt{\frac{m}{1 + m}}\right) \ . \tag{98}$$

The next few properties are about the asymptotics of the elliptic integrals; see e.g. (NIS, Eq. (19.9.1)) for $K(\cdot)$ and (NIS, Eq. (19.12.4)) for $\Pi$.

**Property F.10.** *For all $k \in (0, 1)$, we have*

$$\log\left(\frac{4}{\sqrt{1 - k^2}}\right) \leq K(k) \leq \left(1 + \frac{1 - k^2}{4}\right) \log\left(\frac{4}{\sqrt{1 - k^2}}\right) \leq \frac{5}{4} \log\left(\frac{4}{\sqrt{1 - k^2}}\right) \ .$$

**Property F.11.** *For all $k, \alpha^2 \in (0, 1)$, we have*

$$\Pi(\alpha^2, k) \leq \frac{1}{1 - \alpha^2} \ \log\left(\frac{4}{\sqrt{1 - k^2}}\right) \left(1 + O\left(\sqrt{1 - k^2}\right)\right) \ .$$

### F.5 USEFUL INTEGRALS

We list several useful definite integrals in this section.

**Direct Evaluation:** The first one is a cosine integral divided by a quadratic form.[8]

**Lemma F.12.** *For reals $0 < |b| < a$ and an integer $l$, we have*

$$\int_{-\pi}^{\pi} \frac{\cos(l\omega)\mathrm{d}\omega}{a^2 + b^2 - 2ab \cos \omega} = \frac{2\pi}{a^2 - b^2} \left(\frac{b}{a}\right)^{|l|} \ .$$

The next lemma is also about rational cosine functions.[9]

---

[8]See https://math.stackexchange.com/a/816253.
[9]See https://math.stackexchange.com/a/1235309.

**Lemma F.13.** *For scalar a, we have*

$$\int_{-\pi}^{\pi} \frac{\mathrm{d}\omega}{1 + a\cos(\omega)} = \begin{cases} \frac{2\pi}{1-a^2}, & \text{if } |a| < 1, \\ +\infty, & \text{if } |a| = 1. \end{cases}$$

The next one is similar to the previous one.

**Lemma F.14.** *We have that*

$$\int_{-\pi}^{\pi} \frac{\mathrm{d}\omega}{\sqrt{1 - \cos(\omega)}} = +\infty.$$

*Proof.* We successively deduce

$$\int_{-\pi}^{\pi} \frac{\mathrm{d}\omega}{\sqrt{1 - \cos(\omega)}} = \frac{1}{\sqrt{2}} \int_{-\pi}^{\pi} \frac{\mathrm{d}\omega}{|\sin(\omega/2)|} = 2\sqrt{2} \int_{0}^{\pi/2} \frac{\mathrm{d}\omega}{\sin(\omega)} = +\infty,$$

where we used that $\int \mathrm{d}\omega / \sin(\omega) = -\log|\csc(\omega) + \cot(\omega)| + C$. $\qquad\square$

**Reductions to Elliptic Integrals:** We now list several cosine integrals that can be reduced to elliptic integrals (see Appendix F.4 for their definitions).

**Lemma F.15.** *For any $a \in (0, 1)$, we have*

$$\int_{-\pi}^{\pi} \frac{\mathrm{d}\omega}{|1 - a - \exp(i\omega)|} = \frac{4}{2-a} K\left(\frac{\sqrt{1-a}}{1 - a/2}\right), \tag{99}$$

*where $K(\cdot)$ is the complete elliptic integral of the first kind, cf. (94).*

*Proof.* Using $\cos(\omega) = 1 - 2\sin^2(\omega/2)$ and the substitution $\omega' = \omega/2$, we successively deduce

$$\int_{-\pi}^{\pi} \frac{\mathrm{d}\omega}{|1 - a - \exp(i\omega)|} = 2 \int_0^{\pi} \frac{\mathrm{d}\omega}{\sqrt{1 + (1-a)^2 - 2(1-a)\cos(\omega)}}$$

$$= 2 \int_0^{\pi} \frac{\mathrm{d}\omega}{\sqrt{a^2 + 4(1-a)\sin^2(\omega/2)}}$$

$$= \frac{4}{a} \int_0^{\pi/2} \frac{\mathrm{d}\omega'}{\sqrt{1 + 4\left(\frac{1-a}{a^2}\right)\sin^2(\omega')}}.$$

Applying Property F.8 to reduce this to the standard elliptic integral completes the proof. $\qquad\square$

The next lemma handles a more general case. Note that it recovers Lemma F.15 when $a = b$ since $\Pi(0, k) = K(k)$ by definition.

**Lemma F.16.** *For any $a, b \in (0, 1)$, we have*

$$\int_{-\pi}^{\pi} \frac{|1 - a - \exp(i\omega)|}{|1 - b - \exp(i\omega)|^2} \mathrm{d}\omega = \frac{2a^2}{b^2(1 - a/2)} \Pi\left(\frac{b^2(1-a) - a^2(1-b)}{b^2(1 - a/2)^2}, \frac{\sqrt{1-a}}{1 - a/2}\right), \tag{100}$$

*where $\Pi$ is the complete elliptic integral of the third kind, cf. (96).*

*Proof.* We assume that $a \neq b$ to begin and handle the case of $a = b$ by continuity. Denote $h(a, \omega) = \sqrt{1 + (1-a)^2 - 2(1-a)\cos(\omega)}$

$$\int_{-\pi}^{\pi} \frac{|1 - a - \exp(i\omega)|}{|1 - b - \exp(i\omega)|^2} \, d\omega = \int_{-\pi}^{\pi} \frac{|1 - a - \exp(i\omega)|^2}{|1 - a - \exp(i\omega)| \, |1 - b - \exp(i\omega)|^2} \, d\omega$$
$$= \frac{1 + (1-a)^2}{h(a, \omega) \, h(b, \omega)^2} - 2(1-a) \frac{\cos(\omega)}{h(a, \omega) \, h(b, \omega)^2} \, .$$

We next add and subtract terms to make the numerator of the second term read $h(b, \omega)^2$ to give

$$\int_{-\pi}^{\pi} \frac{|1 - a - \exp(i\omega)|}{|1 - b - \exp(i\omega)|^2} \, d\omega = \int_{-\pi}^{\pi} \frac{1 + (1-a)^2 - \frac{1-a}{1-b}\left(1 + (1-b)^2\right)}{h(a, \omega) \, h(b, \omega)^2} \, d\omega + \frac{1-a}{1-b} \int_{-\pi}^{\pi} \frac{d\omega}{h(a, \omega)} \, . \quad (101)$$

From Lemma F.15, the second term above can be written as

$$\frac{1-a}{1-b} \int_{-\pi}^{\pi} \frac{d\omega}{h(a, \omega)} = \frac{4(1-a)}{(1-b)(2-a)} \, K\left(\frac{\sqrt{1-a}}{1 - a/2}\right) \, . \quad (102)$$

The first term of (101) can similarly be reduced to the elliptic integral form with $\cos(\omega) = 1 - 2\sin^2(\omega/2)$ and the substitution $\omega' = \omega/2$ as

$$\int_{-\pi}^{\pi} \frac{d\omega}{h(a, \omega) \, h(b, \omega)^2} = \frac{2}{ab^2} \int_0^{\pi} \frac{d\omega}{\sqrt{1 + \frac{4(1-a)}{a^2} \sin^2(\omega/2)} \left(1 + \frac{4(1-b)}{b^2} \sin^2(\omega/2)\right)}$$
$$= \frac{4}{ab^2} \int_0^{\pi/2} \frac{d\omega'}{\sqrt{1 + \frac{4(1-a)}{a^2} \sin^2(\omega')} \left(1 + \frac{4(1-b)}{b^2} \sin^2(\omega')\right)} \, .$$

This can be written in terms of elliptic integrals using Property F.9 as

$$\int_0^{\pi/2} \frac{d\omega'}{\sqrt{1 + \frac{4(1-a)}{a^2} \sin^2(\omega')} \left(1 + \frac{4(1-b)}{b^2} \sin^2(\omega')\right)}$$
$$= \frac{a}{2-a} \left(\frac{b^2(1-a)}{b^2(1-a) - a^2(1-b)}\right) K(k) - \frac{a^3(1-b)}{(2-a)(b^2(1-a) - a^2(1-b))} \Pi(\alpha^2, k) \, , \quad (103)$$

with $k = \sqrt{1-a}/(1 - a/2)$ and

$$\alpha^2 = \frac{b^2(1-a) - a^2(1-b)}{b^2(1 - a/2)^2} \, .$$

Plugging in (102) and (103) into (101), we find that the $K(\cdot)$ term cancels out, completing the proof. $\square$

## F.6 OTHER HELPER RESULTS

We list several other miscellaneous useful results.

**Lemma F.17.** *For a sequence $\boldsymbol{\beta} = (\beta_0, \beta_1, \ldots) \in \ell^2$ and a constant $0 \leq c < 1$, we have*

$$\sum_{t=0}^{\infty} \sum_{\tau=0}^{\infty} \beta_t \beta_\tau c^{|t-\tau|} \leq \left(\frac{1+c}{1-c}\right) \|\boldsymbol{\beta}\|_2^2 \, .$$

*Proof.* We break the sum into powers of $c$ and use the Cauchy-Schwarz inequality $(*)$ to get

$$\sum_{t=0}^{\infty}\sum_{\tau=0}^{\infty}\beta_t\beta_\tau c^{|t-\tau|} = \|\boldsymbol{\beta}\|_2^2 + 2\sum_{k=1}^{\infty}c^k\left(\sum_{t=0}^{\infty}\beta_t\beta_{t+k}\right)$$

$$\stackrel{(*)}{\leq} \|\boldsymbol{\beta}\|_2^2 + 2\sum_{k=1}^{\infty}c^k\|\boldsymbol{\beta}\|_2^2 .$$

Summing up the geometric series with a multiplier $0 \leq c < 1$ completes the proof. $\qquad\square$

**Lemma F.18.** *Consider a random vector $\boldsymbol{x}$ that satisfies $\mathbb{E}[\boldsymbol{x}] = 0$, $\mathbb{E}[\boldsymbol{x}\otimes\boldsymbol{x}] = \boldsymbol{H} \succeq \mu I$ for some $\mu > 0$ and $\mathbb{E}\left[\|\boldsymbol{x}\|_2^2\boldsymbol{x}\otimes\boldsymbol{x}\right] \preceq R^2\boldsymbol{H}$. Then, we have for all $\eta \leq 1/R^2$ and all PSD matrices $\boldsymbol{M}$ that*

$$\mathsf{Tr}\left[(\boldsymbol{I} - \eta\boldsymbol{x}\otimes\boldsymbol{x})\boldsymbol{M}(\boldsymbol{I} - \eta\boldsymbol{x}\otimes\boldsymbol{x})\right] \leq (1 - \eta\mu)\mathsf{Tr}\left[\boldsymbol{M}\right] .$$

*Proof.* The left side above (call it "LHS") is bounded by

$$\begin{aligned}
\mathrm{LHS} &= \mathsf{Tr}\left[\boldsymbol{M}\right] - 2\eta\mathsf{Tr}\left[\boldsymbol{M}\boldsymbol{M}\right] + \eta^2\mathsf{Tr}\left[\mathbb{E}\left[\|\boldsymbol{x}\|_2^2\boldsymbol{x}\otimes\boldsymbol{x}\right]\boldsymbol{M}\right] \\
&\leq \mathsf{Tr}\left[\boldsymbol{M}\right] - 2\eta\mathsf{Tr}\left[\boldsymbol{H}\boldsymbol{M}\right] + \eta^2 R^2\mathsf{Tr}\left[\boldsymbol{H}\boldsymbol{M}\right] \\
&\leq \mathsf{Tr}\left[\boldsymbol{M}\right] - \eta\mathsf{Tr}\left[\boldsymbol{H}\boldsymbol{M}\right] \\
&\leq (1 - \eta\mu)\mathsf{Tr}\left[\boldsymbol{M}\right] ,
\end{aligned}$$

where we used (a) $\mathbb{E}\left[\|\boldsymbol{x}\|_2^2\boldsymbol{x}\otimes\boldsymbol{x}\right] \preceq R^2\boldsymbol{H}$, (b) $\eta \leq 1/R^2$, and (c) $\boldsymbol{H} \succeq \mu\boldsymbol{I}$. $\qquad\square$

**Lemma F.19.** *For PSD matrices $\boldsymbol{0} \preceq \boldsymbol{A}_1, \ldots, \boldsymbol{A}_k \preceq \boldsymbol{I}$ of shape $d \times d$, we have $|\mathsf{Tr}\left[\boldsymbol{A}_1\cdots\boldsymbol{A}_k\right]| \leq d$.*

*Proof.* Recall the inner product $\langle\boldsymbol{A}, \boldsymbol{B}\rangle = \mathsf{Tr}\left[\boldsymbol{A}\boldsymbol{B}^\top\right]$ on the space of real $d \times d$ matrices. Using Hölder's inequality on the Schatten $p$-norms, we get

$$|\mathsf{Tr}\left[\boldsymbol{A}_1\ldots\boldsymbol{A}_k\right]| = |\langle\boldsymbol{A}_1, \boldsymbol{A}_k\cdots\boldsymbol{A}_2\rangle| \leq \|\boldsymbol{A}_1\|_{S_1}\|\boldsymbol{A}_k\cdots, \boldsymbol{A}_2\|_{S_\infty} .$$

Here, the Schatten 1-norm $\|\cdot\|_{S_1}$ is the $\ell_1$ norm of the singular values (i.e. the nuclear norm); this is just the trace for a PSD matrix. Thus,

$$\|\boldsymbol{A}_1\|_{S_1} = \mathsf{Tr}\left[\boldsymbol{A}_1\right] \leq \mathsf{Tr}\left[\boldsymbol{I}\right] = 1 .$$

The $\|\cdot\|_{S_\infty}$ is the $\ell_\infty$ norm of the singular values, i.e. the operator norm $\|\cdot\|_2$. We get,

$$\|\boldsymbol{A}_k\cdots\boldsymbol{A}_2\|_2 \leq \|\boldsymbol{A}_k\|_2\cdots\|\boldsymbol{A}_2\|_2 \leq 1 .$$

$\qquad\square$

**Lemma F.20.** *For some fixed integer $t \geq 1$ and constants $a > 0$, $\rho \in (0, 1)$, define the function*

$$f(\tau) = \tau + \frac{1}{\rho a}\exp(-a\tau)\,\mathbb{1}\left(\tau < t - 1\right) .$$

*For $\hat{\tau} = \min\{t - 1, a^{-1}\log(1/\rho)\}$, we have,*

$$f(\hat{\tau}) = \min\left\{t - 1, \frac{1}{a}(1 + \log(1/\rho))\right\} \leq \frac{1}{a}(1 + \log(1/\rho)) .$$

*Proof.* The convex function $\tau \mapsto \tau + \frac{1}{\rho a} \exp(-a\tau)$ is minimized at $\tau_\star = a^{-1} \log(1/\rho) > 0$ with a minimum value of $a^{-1}(1 + \log(1/\rho))$. If $t - 1 \le \hat{\tau}_\star$, we take $\hat{\tau} = t - 1$ and $f(\hat{\tau}) = t - 1 \le \hat{\tau} \le a^{-1}(1 + \log(1/\rho))$. $\qquad \square$

The next lemma is from (Pillutla et al., 2023, Lemma 13).

**Lemma F.21.** *Consider a function* $\varphi : [0, \eta_{\max}] \to \mathbb{R}_+$ *given by*

$$\varphi(\eta) = A \exp(-\mu\eta T) + B\eta + C\eta^2 \log^2 \left( \frac{1}{\eta\mu} \right) ,$$

*given some constants* $\eta_{\max}, \mu, A, B, C > 0$. *If* $T \ge (\mu\eta_{\max})^{-1}$, *then we have*

$$\varphi(\eta_\star) \le A \exp(-\mu\eta_{\max}T) + \frac{3B}{\mu T} \left( 1 \vee \log \frac{A\mu T}{B} \right) + \frac{3C}{\mu^2 T^2} \left( 1 \vee \log \frac{A\mu^2 T^2}{C} \right)^2 \log^2(T) ,$$

*for some* $\eta_\star \le \eta_{\max}$ *depending on* $A, B, C, \mu, T$.

**Lemma F.22.** *For* $0 < c < 1/4$, *we have,*

$$0 < x \le \frac{c}{9 \log^2(9/c)} \quad \Longrightarrow \quad x \log^2(1/x) \le c .$$

## G    EMPIRICAL DETAILS

We train image-classification models using the CIFAR10 dataset and language models using the Stack Overflow Next Word Prediction (SONWP) dataset available on `tensorflow-datasets`.

### G.1    IMAGE CLASSIFICATION

Image classification has long been studied in DP ML. For example, the original DP-SGD work of Abadi et al. (2016) focused on this task. We use CIFAR10 which has 50,000 training and 10,000 test examples. We evaluate and compute test accuracies on the entire test set, following the open-sourced code of Kairouz et al. (2021a). We reuse the network architecture, dataset processing, and initialization strategies presented in Kairouz et al. (2021a); in particular, the architecture we use can be found in their Table 2 (b).

**Setup and Tuning:** We train all mechanisms for 2000 steps using a batch size of 500 and a clip norm of 1. This leads to ML training dynamics of 20 epochs and 100 steps per epoch. We performed some initial small grid searches which showed nearly ubiquitously that momentum of 0.95 (searched over the grid $0, 0.85, 0.9, 0.95$) and a linear learning rate cooldown $0.05\times$ the initial learning rate over the last 500 steps of training improved model utility for all privacy levels. Thus, we fix these settings for all mechanisms except DP-SGD, for which no momentum performed best. For each mechanism, we then run a tuning grid search for the learning rate on coefficients in $\{1, 2, 5\}$ on powers in [-2, 3], selecting the best mechanism for each privacy level from this interval. Final experiments are repeated 12 times in each setting and show 95% bootstrapped confidence intervals.

Some mechanisms include additional hyperparameters that specify the exact mechanism's structure. For example, ME is specified by both the number of steps $n$ and the max number of participations $k$. We include such parameters in the mechanism name. For all mechanisms, $n = 2000$.

### G.2    LANGUAGE MODELING

Language modeling has been prominently studied in user-level DP contexts, usually in conjunction with federated learning (e.g. McMahan et al., 2018). DP training is important for real-world applications of language

models trained on user data as these models can memorize their training data if appropriate mitigations are not applied (Carlini et al., 2019; 2021; 2022; Ippolito et al., 2022; Anil et al., 2023; Kudugunta et al., 2023). Indeed, DP already plays an important role in this application, as evidenced by Google's use of DP for training on-device language models (McMahan & Thakurta, 2022; Xu et al., 2023). StackOverflow Next Word Prediction contains over $10^8$ examples contributed non-identically from 342,477 users. The goal of this task is to predict the next word given a sequence of words. We use the same setup as Choquette-Choo et al. (2023b).

**Setup and Tuning:** We consider a version of DP-FTRL that works with "generalized gradients", i.e., the client update resulting from multiple local gradient steps on a client's data; this is a common strategy to "lift" learning algorithms to the federated learning setting (Kairouz et al., 2021b). We refer to (Reddi et al., 2020) for details. All mechanisms use an $\ell_2$ clip norm of 1, a server momentum of 0.95, and a client learning rate of 1.0. They also use a server learning rate cool-down over the last 25% rounds. Initial tuning showed that these were favorable parameter settings. We train all mechanisms for 2052 steps and report the final evaluation accuracy of the model as reported on a held-out set of $10,000$ examples. We zero out large updates whose $\ell_\infty$ norm exceeds 100. We use the tuned server learning rates from Choquette-Choo et al. (2023b) for all existing mechanisms. For the proposed $\nu$-DP-FTRL mechanisms, we do not perform extensive tuning due to computational costs and instead tune the parameter to minimize the $\ell_2$ error (3) of the total noise added due to $B$ (cf. Choquette-Choo et al., 2023a, Figure 11).

