# OpenReview forum: "Correlated Noise Provably Beats Independent Noise for Differentially Private Learning"
_ICLR.cc/2024/Conference — ICLR 2024 poster_

### Official Review · Reviewer_rSjm · 2023-10-30

**Soundness:** 2 fair
**Presentation:** 1 poor
**Contribution:** 2 fair
**Rating:** 1
**Confidence:** 4

**Summary:**

This paper studies optimization (e.g., SGD) under differential privacy where the noise added to gradients correlates.

**Strengths:**

It is plausible that correlated noise can help in some way.
Researching this seems a valuable contribution.

If the text (or at least the appendix) would be clear and self-contained, and if the text would make the claims fully explicit, then this may become an interesting contribution discussing several interesting issues.

**Weaknesses:**

Parts of the text are very hard to follow.  E.g.,
* The text introduces the limit with T going to \infty, without explaining how this can make sense for a bounded privacy budget (i.e., for a finite privacy budget and an infinite (large) number of iterations, an infinite (large) amount of noise should be added in every step to the gradient, making it hard to converge.
* While Eq (1) defines f(\theta, z) as objective function, Eq (6) seems to treat f as the derivative of the objective function.
* The proof of theorem 2.1 cites Fourrier analysis, but doesn't make explicit what is the derivation the authors have in mind.  There is even no proof that the asymptotic optimality defined by Eq (4) exists (i.e., the limit converges), which is non-trivial as the more iterations are performed, the higher the amount of noise per iteration needs to be.

The text also is insufficiently explicit leading the reader to incorrect assumptions on what is meant.  For example, while most of the text just uses "differential privacy", Appendix A.2 suddenly says that instead of "neighboring datasets" as most machine learning literature considers, the current paper considers zero-out neighborhood, where two sequences of gradients are adjacent if they only differ in one gradient.  This is clearly not the case if we are performing an optimization and are comparing the algorithm being run on neighboring datasets, in which case (almost) every gradient will be different.  As a result, Appendix B seems to be not really proving theorem 2.1 but a variant where "neighboring datasets" is replaced by "zero-out neighbors", which changes the meaning of the theorem.

The paper is not self-contained, and a lot of terms are not even explained in the appendix.

**Questions:**

Before Eq (3) the text says that B is a Toeplitz matrix.  How shall we read B(\omega) after Eq (3)?  B is not a function, and the righthandside of B(\omega)=... evaluates to a complex number rather than a real-valued matrix.

Does the series \beta_t need to satisfy any property to make the sums and limits converge?

How is \gamma_T defined?  The text says that one can infer Eq (4) from this definition.  In Eq 4, how shall we read the superscript 2?  Does it square \gamma_T(B) or does it square its argument B?  (In the former case, many people would write (\gamma_T(B))^2 or \gamma_T^2(B), in the latter case, most people would write \gamma_T(B^2))

**Details Of Ethics Concerns:**

--

---

> ### Author Response · Authors · 2023-11-17
> **Response to Reviewer rSjm: Part 1**
>
> We thank the reviewer for the review. The “technical flaw” pointed out by the reviewer is an incorrect assessment, our original argument is correct (details below). We urge the reviewer to thoroughly re-read and review the paper again for a fair and unbiased assessment of the merits of this paper.
>
> **[Technical flaw] “... for a finite privacy budget and an infinite (large) number of iterations, an infinite (large) amount of noise should be added in every step to the gradient”**
>
> This assessment is incorrect. We prove bounds in the streaming setting, where each data point is processed only once. Thus, the number $n$ of data points and the number of iterations $T$ are equal. To see why a finite amount of noise is sufficient even as $T \to \infty$, consider the uncorrelated noise case (DP-SGD): the privacy guarantee for all iterations can be obtained from parallel composition, so the noise multiplier is independent of $T$. Prior work has also studied this setting, e.g. [Chourasia et al. NeurIPS 2021](https://arxiv.org/pdf/2102.05855.pdf) and [Altschuler & Talwar, NeurIPS 2022](https://openreview.net/pdf?id=pDUYkwrx__w).
>
> For the more general correlated noise setting, Theorem 1.1 gives a rigorous privacy guarantee that holds any $T$, including in the limit $T \to \infty$. Note that the only dependence of the noise multiplier on $T$ is via the sensitivity $\gamma_T(\boldsymbol{B})$.
>
> **"While Eq (1) defines f(\theta, z) as objective function, Eq (6) seems to treat f as the derivative of the objective function."**
>
> The notation is correct — the objective for mean estimation is $f(\theta; z) = z^2 / 2 - z\theta$:
> $$
> \frac{1}{2} (\theta - z)^2 = \frac{z^2}{2} - z\theta + \frac{\theta^2}{2} = f(\theta; z) + r(\theta).
> $$
>
> **"The proof of theorem 2.1 cites Fourier analysis, but doesn't make explicit what is the derivation the authors have in mind."**
>
> The main paper only gives a short proof sketch (which provides an idea of why the statement is true), while the full proof and all its technical details are given in Appendix B.
>
> The technical details are based on standard results in Fourier analysis that we recall in Appendix F (specifically, Theorem F.2). The key idea of the proof is conveyed in that it is based on Fourier analysis, but we strongly feel that technical details regarding the Fourier transform would be a distraction in a short proof sketch.
>
> **"There is even no proof that the asymptotic optimality defined by Eq (4) exists (i.e., the limit converges), which is non-trivial as the more iterations are performed, the higher the amount of noise per iteration needs to be."**
>
> The reviewer is mistaken as we pointed out above. The limit exists in all cases we consider in the theoretical statements. The argument goes as follows:
> - For a finite noise variance, the quantity in Eq. (4) is well-defined. This requires the limiting sensitivity $\gamma_\infty(B)$ (defined in Eq. (5)) and the clip norm $G$ to both be finite; indeed, by Theorem 1.1, the required noise variance is $\gamma_\infty(B)^2 G^2 / (2 \rho)$.
> - The sensitivity $\gamma_\infty(B)^2 = log(1/\nu)$ is finite for $\nu$-DP-FTRL (see Table 2).
> - For mean estimation in Theorem 2.1, we take $G=1$ and for general strongly convex functions in Theorem 3.1, we take $G$ to be the Lipschitz constant of the loss function, which is assumed finite. Thus, the noise variance of DP-FTRL is finite and the limit in Eq. (4) is well-defined.
> - For quadratic functions, we analyze the asymptotic suboptimality of noisy-FTRL (i.e. DP-FTRL without gradient clipping) in Theorem 2.2; this asymptotic quantity is always well-defined. To get a DP bound in this case, (a) we argue for privacy that the gradient norm after $T$ iterations is bounded with high probability, and (b) the suboptimality after $T$ iterations is upper bounded by the asymptotic suboptimality at the same noise variance. The full privacy-utility bound of this case is given in Theorem D.13 in Appendix D.4.

---

> > ### Author Response · Authors · 2023-11-17
> > **Response to Reviewer rSjm: Part 2**
> >
> > **"insufficiently explicit.. Neighboring datasets … zero-out neighborhood."**
> >
> > Differential privacy is always defined with respect to a particular definition of neighboring datasets. The zero-out notion of neighborhood is standard in the DP-FTRL literature: e.g. [Kairouz et al. ICML 2021](https://arxiv.org/pdf/2103.00039.pdf) (Def. 1.1), [Denisov et al. NeurIPS 2022](https://arxiv.org/pdf/2202.08312.pdf)  (Thm. 4.1) or their numerous follow-ups [[1](https://arxiv.org/pdf/2211.06530.pdf), [2](https://arxiv.org/pdf/2306.08153.pdf), [3](https://openreview.net/forum?id=qCglMj6A4z)]. We stick to this notion of neighborhood consistently throughout the paper in order to be consistent with prior work on DP-FTRL. We will be more explicit about this in the revision.
> >
> > Also note that Sec. 2.1.1 of the practical guide of [Ponomareva et al. JAIR 2023](https://arxiv.org/pdf/2303.00654.pdf) says:
> >
> > > [add-or-remove one record] and [zero-out one record] have comparable semantics for a fixed $\varepsilon$
> >
> > Finally, here is a quick proof sketch about why the differential privacy guarantee holds in the learning setting. The reviewer’s specific concern that once a single gradient differs, the entire input stream afterward may differ is addressed by Theorem 4.1 of Denisov et al., which says that computing $CG+W$ with $G$ being a stream of adaptive gradients satisfies the same DP guarantees as $G$ being a stream differing in a single entry between adjacent datasets. The high-level idea is if $x$ is the sensitive example, we can split $G$ into $G_1 + G_2$, where $G_1$ is the contribution to $G$ of all examples except $x$, and $G_2$ is the contribution of $x$. We release all intermediate values, and we can assume the **adversary knows all examples except $x$ and which rounds they participate in**, so the adversary can compute $CG_1$ and subtract it from $CG+W$ to get $CG_2 + W$. $G_2$ will be 0 everywhere for both $D$ and $D’$ except for the single round that $x$ participates in, so this gives a reduction from gradient streams to streams differing in a single entry.
> >
> > **"The paper is not self-contained, and a lot of terms are not even explained in the appendix."**
> >
> > While we strongly disagree with this assessment, we would be happy to modify the paper as needed if the reviewer could provide concrete examples.
> >
> > **"Before Eq (3) the text says that B is a Toeplitz matrix. How shall we read B(\omega) after Eq (3)?"**
> >
> > Note that matrices are denoted with a bold $\boldsymbol{B}$, while non-bold $B$ is the discrete-time Fourier transform $B: [-\pi, \pi] \to \mathbb{C}$.
> >
> >
> > **"Does the series \beta_t need to satisfy any property to make the sums and limits converge?"**
> >
> > All the limits are well-defined as long as $\beta_t$’s are square-summable (denoted as $\boldsymbol{\beta} \in \ell^2$ in Theorem C.14, for instance). Some statements need the stronger assumption that $\beta_t$’s are absolutely summable (e.g. the lower bound in Theorem 2.2 or Theorem C.17), and these are explicitly pointed out (see e.g. the discussion at the top of p.28 right after Theorem C.17).
> >
> > **"How is \gamma_T defined?"**
> >
> > $\gamma_T(\boldsymbol{B})^2 = \max_{t} \Vert (\boldsymbol{B}^{-1})_{:,t} \Vert_2$ is the maximum column norm of $\boldsymbol{B}$. This is defined in the statement of Theorem 1.1.
> >
> > **"The text says that one can infer Eq (4) from this definition."**
> > Do you mean Eq. (5)? Please see Property C.1 or Theorem F.3 for the proof of Eq. (5).
> >
> > **"In Eq 4, how shall we read the superscript 2?"**
> >
> > There is no superscript in Eq. (4). Do you refer to Eq. (5)?
> > $\gamma_T(B)^2$ refers to $\big(\gamma_T(B)\big)^2$. Squaring the argument would be written as $\gamma_T(B^2)$. We believe there is no ambiguity between the two.

---

> > > ### Comment · Reviewer_rSjm · 2023-11-17
> > >
> > > A first potential improvement would be to make more explicit that you study the streaming setting where every example is processed only once.  It is true that the word "stream" occurs in the text in Section 1.1, but only once and without many other clues such as making explicit this is not merely a way in which the same dataset is repeatedly provided to the learner but referring to the setting where instances are only seen once.  For example, in Algorithm 1, after line 1 one could say "get the next instance $z_t$ from the stream", a phrase occurring in the pseudocode of many streaming algorithms to be more explicit.  It is possible that some of my comments are formulated in an unclear way as I may have only realized quite late we implications of this setting.  Nevertheless, I still believe the text is very hard to follow in general, and being more precise and explicit could help the reader a lot.
> > >
> > > So let me first get a few further minor elements right about the setup.
> > >
> > > * In Theorem 1.1, why do we have $\gamma_T(B) = \max_{t<T} ...$, i.e., why is the last $T$-th column of $B^{-1}$ not considered for $\gamma_T(B)$ ?
> > > * $B$ is called the "correlation matrix".  Normally, if $x$ is a random vector, then the correlation matrix of $x$ is defined as $\mathbb{E}[x x^\top]$, which is always a symmetric matrix.  However, the text says $b$ is a lower triangular matrix.  I conclude that $B$ is not the expectation of $w_t w_t^\top$ for $w_t$ the generated noise ?
> > >
> > >
> > > Making abstraction of these puzzling but less critical questions, it seems that the "stream" character of the problem makes some of my "impossible" comments changes into "trivial".    In particular, if there are an infinite number of instances, then whatever is the noise distribution, as long as the noise is zero-mean (given the instance) it is possible (certainly for linear regression) to get arbitrarily close to the optimum $\theta_*$.  The FTRL algorithms are suboptimal in this context, a simple approach is just to collect statistics for a sufficiently long time before establishing a model or a gradient.  For example, for linear regression one can just iterate over T instances $(z_i)_{i=1}^T$ and keep the running sums $V_t=\sum_{i=1}^t x_i y_i$ and $\Sigma_t=\sum_{i=1}^t z_i z_i^\top$, after which one can apply the classic formula $\theta_T = \Sigma_T^{-1} V_T$.  In this way $\lim_{T\to\infty} \|\theta_T - \theta_*\| = 0$.  For strongly convex problems one can apply a similar strategy to move in steps closer to the optimum, even in the presence of (arbitrary and unknown but fixed and zero-mean) noise.  I get that maybe you study these suboptimal algorithms (at least suboptimal for the settings you mention) to get a better insight in these algorithms while not aiming at direct applications, but in that case making such clear in the text would avoid a reader just implements your algorithm and thinks this is the best he can do.
> > >
> > > I now realize I misread Eq (6).  Usually when there is are two formulas separated by "with", the second formula defines something occurring in the first formula.  This is not the case here, Eq (6) just defines three independent functions and calls them all "objective function" while probably only the first one is the objective function and the other two are arbitrary other useful notations.
> > >
> > > The appendix contains useful material, but the proof of Theorem 2.1 doesn't refer to it, it would be convenient for a reader (and would not take much space in the main text) to include the pointers provided in your response.  Still, the statement of Thm 2.1 uses the symbol $\beta_{dpsgd}$ without defining it.  The first definition of $\beta$ doesn't mention it should be square-summable (as C.14 does), so as this is a requirement / part of $\beta$'s definition it may make sense to add this when first introducing $\beta$.   This would also avoid the reader not knowing why Eq (4) converges (as the reader at that point didn't see C.14 yet).
> > >
> > > In general, it is always preferrable to (a) include all conditions in claims made in the main text and (b) systematically cite precise sections in an appendix at points where the reader can look up more details, as reviewers/readers are not supposed to read the appendix systematically and hence may not be aware that more details are available.

---

> > > > ### Comment · Reviewer_rSjm · 2023-11-17
> > > >
> > > > Even if all the above would have been made clear already, it would still be useful for a reader to get the references you provide in your response to give context to the sudden introduction of "zero-out neighboring".
> > > >
> > > > Details:
> > > >
> > > > * I agree that $(\gamma_T (B))^2$ and $\gamma_T(B^2)$ are unambiguous, and there is also the clear $\gamma_T^2(B)$.  The variance $\gamma_T(B)^2$ is less common.  Especially given the definition symbol $:=$ things may get confusing since on the lefthand side of $:=$ one usually puts the symbol one defines, not some expression containing it (i.e., some people would prefer $\gamma_T(B)=(..)^{1/2}$ rather than the unambigous variants).
> > > >
> > > >
> > > > Conclusion:
> > > > It is important to understand that a reviewer/reader typically won't look at the appendix unless the main text tells them what information can be found there.  If the text would have referenced the appendix each time it omits details, many of my comments may not have been needed.  Remember the call for papers says "Authors may use as many pages of appendices (after the bibliography) as they wish, but reviewers are not required to read the appendix.".  Therefore, it is desirable that you systematically reference details in the appendix whenever you make a claim in the main text which is not fully supported with arguments/details/...
> > > > Some of the unclarities indeed can be disambiguated by minor details in the text, but given that at other points the text contains less rigorous notations it may be hard to expect a reader can infer everything from small details, it is preferably to make the important elements explicit (or otherwise be sufficiently redundant).

---

> ### Author Response · Authors · 2023-11-19
> **Response to Reviewer rSjm**
>
> We thank the reviewer for the clarifications. We urge the reviewer to please re-read and review the paper again for a fair assessment (including the significance and novelty) of the key contributions, which are the following:
> - **Theoretical**: We establish that DP-FTRL is _provably_ better than DP-SGD as a function of problem parameters such as the effective dimension and condition number. This is the _first_ clear theoretical separation between DP-SGD and DP-FTRL.
> - **Empirical/Practical**: We show empirically that the proposed $\nu$-DP-FTRL is substantially better than other efficient DP-FTRL variants for private deep image classification and language modeling. Further, it can nearly match the SoTA “multi-epoch” variant (which has a $O(T^3)$ cost; denoted “ME”).
>
>
> We would like to reiterate that all theorem statements in the main paper and appendix are fully self-contained (we are happy to fix any that are not). All relevant background and technical details are provided in the appendices. The appendices also include several navigational aids, such as a table of contents and outlines at the beginning of each section, to help a reader easily find the information they need.
>
> We would be more than happy to answer further questions or discuss any technical issues.
>
> **"The FTRL algorithms are suboptimal in this context"**
>
> You are right about potentially better algorithms for these problems. We are fully transparent about this fact at the start of Section 2:
>
> > We do not aim to achieve the best possible rates in these stylized models. Rather, our goal is to understand the noise dynamics of DP-FTRL and show a separation with DP-SGD.
>
> We also mention on page 1 that DP-FTRL is a practical algorithm that is deployed in industrial settings but a theoretical understanding is lacking. Our analysis is a major step in building a theoretical understanding of this algorithm.
>
> **Minor clarifications**:
>
> Thank you for the suggestions. We have made some changes (see uploaded revision) and clarified confusions as detailed below.
>
> - **Streaming setting**: Besides clearly mentioning that we consider a streaming setting, there are other clues in the setup. The dataset $\mathcal{D}$ has $T$ data points, and we run the algorithm for $T$ iterations (e.g. return $\theta_T$, the noise correlation matrix $\boldsymbol{B}$ is of size $T \times T$, etc.). We believe it is clear from the context that the size of the dataset grows with the number of iterations.
> - **max_{t<T}**: We do consider the last column. The matrices are 0-indexed — this can be inferred from Eq. (2), for instance. We have added a clarifying note in the revision.
> - **correlation matrix $\boldsymbol{B}$**: The term _correlation matrix_ comes from the cited line of work on DP-FTRL [[1](https://arxiv.org/pdf/2202.08312.pdf), [2](https://arxiv.org/pdf/2211.06530.pdf), [3](https://arxiv.org/pdf/2306.08153.pdf)]. We believe that it is unambiguously clear from Eq (2) that $\boldsymbol{B}$ is not the covariance matrix of the noise $\boldsymbol{w}_t$.
> - **Eq (6)**: The terms defined in Eq. (6) map back to the terms introduced in Eq. (1): they are the objective, loss, and regularizer respectively. Again, we believe that there is no ambiguity in the notation here.
> - **Pointer to the proof of Theorem 2.1**: We have added a pointer to the right appendix, following the reviewer's suggestion. All other theoretical statements already contain the right appendix pointers.
> - **statement of Thm 2.1 uses the symbol $\boldsymbol{\beta}_{\text{dpsgd}}$ without defining it**: Note that $\boldsymbol\beta_{\text{dpsgd}} = (1, 0, \ldots)$ are the noise weights that retrieve DP-SGD as a special case of DP-FTRL. The text says "... the asymptotic sub-optimality … of DP-SGD is $F_\infty(\boldsymbol{\beta}_{\text{dpsgd}}) = \cdots$", so we believe that it is clear from the context.
> - **doesn't mention [$\boldsymbol{\beta}$] should be square-summable**: Note that $\boldsymbol{\beta}$ being square-summable is not a part of the definition --- $F_\infty(\boldsymbol{\beta})$ is simply infinite if $\boldsymbol{\beta}$ is not square-summable. Furthermore, all theorem statements in both the main paper and appendix give appropriate qualifications on the summability of $\boldsymbol{\beta}$. Technical details about the convergence of various stochastic processes can also be found in the appendix.
> - **zero-out neighboring**: We have added a note in the revision pointing out the exact notion of neighborhood. We note that the results present an apples-to-apples comparison of DP-SGD vs. DP-FTRL under the same notion of neighborhood. We believe that the exact details of the neighborhood are technical detail that is not central to understanding the key results regarding the theoretical separation of DP-SGD and DP-FTRL.

---

> > ### Author Response · Authors · 2023-11-22
> >
> > Dear Reviewer rSjm,
> >
> > Thanks again for your feedback. To ensure a productive discussion, we kindly request that you take a moment to review our responses to your questions before the discussion period concludes in approximately one day.
> >
> > We hope that the additional background and context provided allow a clearer assessment of the novelty, significance, and rigor of the paper's theoretical and empirical contributions. If you have any further questions or concerns, we will be happy to address them.
> >
> > Best,
> > The authors

---

> > ### Comment · Reviewer_rSjm · 2023-11-23
> >
> > I didn't have time to comment each and every page of the text in detail.  Here, I provide comments for a short section, covering only the details up to part of the proof for Theorem 2.1.  In later parts of the appendix, the number of issues (both minor ones and points where a reader needs to spend a big amount of time to guess things from context) remains at least equally dense as in what I commented below.  In general, my impression still is that the results are quite interesting, but that the text makes only limited effort to help a reader understand the technical details.
> >
> >
> > Main text:
> >
> > Footnote 4 says "a $\rho$-zCDP guarantee can be readily translated into $(\epsilon,\delta)$-differential privacy" without giving the definition of either notion.  The conditions are hidden in appendix A.2, the main text could simply refer to them to get more self-contained.
> >
> > Theorem 2.1 uses the term "$\rho$-zCDP sequence" while usually only algorithms are called private (not sequences).  Maybe you mean that the DP-SGD from which you obtained the sequence should get parameters so it is $\rho$-zCDP, but the text doesn't seem to say what parameters to choose to realize that.
> >
> > Appendix A:
> >
> > This privacy of (11) can be seen a postprocessing -> ...  be seen as a postprocessing
> >
> > It is still unclear to me why you define neighboring datasets in terms of gradients rather than the more more common definition in terms of the underlying data (even if one can reason that if two instances have the same gradient, their difference isn't relevant for the algorithm).
> >
> > Appendix B:
> >
> > It is unclear why the statement of Theorem 2.1 says G=1, while the proof says $G\ge 1$.  Of course, $G\ge 1$ so it doesn't harm.
> >
> > Definition F.1. if satisfies -> if it satisfies
> >
> > Given that Definition F.1 talks about $\mathcal{X}$-valued sequences, it would be consistent for the sentence after Definition 2.1 to talk about $\mathcal{X}$-valued sequences rather than $\mathbb{R}^d$-valued sequences (or else that you irst say "From now on we will choose $\mathcal{D}=\mathbb{R}^d$." or use $\mathbb{R}^d$ already in the definition or ....).
> >
> > "LTI systems can be described in linear algebraic notation by the action of an infinite Toeplitz matrix H on an element of $\ell_{2e}$" -> do you mean $\ell_{2e}^d$?
> >
> > The text here doesn't use the original notation $\mathcal{A}_t$ anymore, I guess that you mean one can for such systems find a matrix $H$ such that $\mathcal{A}_t(x) = H_{t,:}x$.
> >
> > Of course, this is easier to see if $x\in\mathbb{R}$ rather than the originally used $\mathbb{R}^d$.  If $x_\tau\in \mathbb{R}^d$, then to let $y=Hx$ match we need $x\in\mathbb{R}^{\infty\times d}$ rather than its transpose.
> >
> > Later, before Theorem 9.2 we will discover this is in fact not correct, the authors are thinking of $y_t\in\mathbb{R}^p$, so $H$ will become an appropriate tensor (or maybe not, as Theorem F.2 says G (i.e., H) is in $\mathbb{R}^{p\times d}$, so I guess all components of $x_t$ and $y_t$ get the same linear transformation.
> >
> > It is natural to denote the DTFT of $x$ by $X(\omega)$, of $y$ by $Y(\omega)$, but why is the DTFT of $h$ denoted by $G(\omega)$?  I suppose it is more confusing for the reader, especially as in Theorem 2.1 another $G$ (real number, clip norm) is being used (even if not bold-faced there).
> >
> > In Theorem F.2, the product $\otimes$ isn't defined (but is hard to guess given the dimension confusion above).  Anyway, $\Sigma = \mathbb{E}[x_t \times x_{\tau}]$ seems to be a constant, not depending on $t$ and $\tau$, except perhaps that this may not be the correct interpretation as the integral over $G(\omega) \Sigma G(\omega)^*$ then shouldn't depend on $X(\omega)$ nor $Y(\omega)$.  So maybe $\Sigma$ does depend on $t$ and $\tau$ (a tensor with on the diagonal $d\times d$ matrices)?  In any case, this wouldn't write $\Sigma$ as a function of $\omega$ to fit in the integral.  The theorem doesn't say anything about $\mathbb{E}[y_t \otimes y_\tau]$ with $t\neq \tau$.

---

> ### Comment · Reviewer_rSjm · 2023-11-23
>
> Now we get to applying Theorem F.2 in the proof of Theorem 2.1.  We should think of the following:
>
> * $G$ in Thm 2.1 is not related to $\mathbf{G}(\omega)$ is Theorem F.2
> * I guess $x_t$ in Thm F.2 corresponds to $\delta_t$ or $\beta_t$ or $w_t$ in Thm 2.1.  Unfortunately, for $x_t$ in Thm F.2, $t$ ranges from $-\infty$ to $+\infty$ while both $\delta_t$, $\beta_t$ and $w_t$ in Thm 2.1 are sequences for which $t$ ranges from $t=0$ to $t=\infty$.  Maybe, let us guess that $x_t$, $y_t$ and $H$ in Thm F.2 correspond to $w_t$, $\beta_t$ and $\delta_t$ in Thm 2.1.
> * The text doesn't explain why the transformation from $\theta_t$ into $\delta_t$ was useful, Eq (11) looked much more as a linear transformation, but apparently it was a preparation for applying Thm F.2, and hence could provide a clue, even though only some of the symbols in the $\delta_t$ equation occur in the resulting bound.  For example, the newly introduced $w_{sgd}$ doesn't occur in the resulting $F_\infty(B)$ equation.  Maybe the term $\eta\sigma_{sgd} w_{sgd}$ is not important and we should only look at the term involving $w_t$, but in that term $\sigma_{sgd}$ does not occur, while this variable does occur in the $F_\infty(B)$ equation.
> * Applying Thm F.2 requires the LTI is asymptotically stable, but the text doesn't argue why this condition is fulfilled.  I guess this is a consequence of the assumption that the objective function is convex.
>
> Nevertheless, it is plausible that with adding half a dozen of additional intermediate steps and explanations this step can be shown to be correct.
>
> Next, the text says "Thus $F_\infty$ is a product of ...".  In fact, $F_\infty$ is a function, and $F_\infty(B)$ is an integral, which we can interpret as a sum, but not immediately as a product.  For every specific $\omega$, indeed we have as integrand $|B(\omega)|^2 \gamma_\infty(B)^2$ multiplied with other factors, where $\gamma_infty$ is an integral (a sum) with $|B(\omega)|^{-2}$ as one of its terms.  To write $F_\infty(B)$ as a product, an alternative strategy (not mentioned in the text) would be to first observe that $\sigma_{sgd}^2$ is a constant, and then to minimize the part remaining after removing $\sigma_{sgd}^2$, in that case one could write $F_\infty(B)$ as a product by getting $\gamma_\infty(B)$ out of the integral.
>
> Eq (14) then suggests that of the other factors in the product, including $G^2$ and $\sigma_{dp}^2$, only the denominator should be included when determining $B(\omega)$ to minimize $F_\infty(B)$, but the text doesn't give a reason.
>
> "elliptic integrals coming from the $\sigma_{dp}$ term" -> I guess you mean "factor" rather than "term" here.
>
> The text says the proof of the error bound now follows by applying a list of results it refers to.  However, the bound stated in Theorem 2.1 contains the variable $\rho$, but neither Lemma F.15 nor Corollary C.5 seem to introduce a $\rho$ into the equation we have so far.

---

### Official Review · Reviewer_NgBE · 2023-10-31

**Soundness:** 3 good
**Presentation:** 3 good
**Contribution:** 3 good
**Rating:** 8
**Confidence:** 2

**Summary:**

This paper investigates DP linear regression and strongly convex problems. The goal is to minimize $F(\theta) = \mathbb{E}\_z [f(\theta, z)] + r(\theta)$ while providing a DP guarantee. The authors introduce a new version of DP-FTRL named $\nu$-DP-FTRL. Instead of independent noises, $\nu$-DP-FTRL uses a correlated noise $\tilde w_t = \sum_{\tau=0}^t \boldsymbol{B}\_{t,\tau} w_\tau$, where each $w_\tau$ is an i.i.d. Gaussian and $\boldsymbol{B}$ is a Toeplitz lower-triangle matrix. The parameter is then privately updated as $\theta_{t+1} = \theta_t - \eta(g_t + \tilde w_t)$, with $g_t$ representing the non-private stochastic gradient (which may or may not be clipped).

The paper sets $\nu$-DP-FTRL against DP-SGD, comparing their theoretical guarantees and empirical performances. Theoretically, the authors provide upper bounds for the suboptimality gap $\mathbb{E}[F(\theta_T)] - \inf F(\theta)$ for both algorithms. Notably, $\nu$-DP-FTRL attains an asymptotic suboptimality gap of $\tilde O(d_\text{eff}\eta^2/\rho)$, where $d_\text{eff} \le d$ is the effective dimension. This matches the existing lower bound up to a logarithmic factor and improves upon the $O(d\eta/\rho)$ rate of DP-SGD. Empirically, $\nu$-DP-FTRL also demonstrates better experiment performance compared to DP-SGD.

**Strengths:**

- This paper provides a very detailed theoretical proof supporting the empirical observation that DP learning with correlated noise surpasses that with independent noise.
- The introduced algorithm, $\nu$-DP-FTRL, offers a notable improvement in the theoretical utility upper bound when compared to the leading bound of DP-SGD. Notably, this improved bound is dependent on the effective dimension $d\_\text{eff}$, which is often tighter than the vacuous dimension $d$ and thus adapts better to the problem difficulty. Furthermore, the bound demonstrates an improved dependence on the learning rate, improving from $O(\eta)$ to $O(\eta^2)$, which aligns with the existing lower bound.
- The authors utilize the Fourier transform as an instrumental analysis tool for bounding the suboptimality gap. This analytical approach is beyond my expertise, but it suggests an new perspective for analyzing the asymptotic behavior of optimization problems.

**Weaknesses:**

My main concern is about the privacy guarantee. The authors use a high probability bound to argue that most of the time, the stochastic gradients won't exceed the clipping norm. Consequently, $\nu$-DP-FTRL doesn't need any gradient clipping. But I'm not sure if this meets the standard DP definition, as sensitivity might be prohibitively high in rare cases. To me, the link between the high probability result and the standard definition isn't clear. Once this is addressed, this paper has very solid results.

**Questions:**

I'm curious about the role of $\nu$ in $\hat \beta_t^\nu$. Since $\binom{1/2}{t}$ in $\hat \beta_t^\nu$ already decreases rapidly (roughly $1/t$ if I'm correct), is it necessary to add an additional damping term $(1-\nu)^t$? Also, $\nu$ is currently set to some small value in the experiments, so $(1-\nu)^t$ decays significantly slower than $\binom{1/2}{t}$. Thus, I wonder if $\nu$ can be dropped to save some tuning effort. How would removing $\nu$ (or setting $\nu=0$) affect the theoretical bound and the empirical performance?

---

> ### Author Response · Authors · 2023-11-17
> **Response to Reviewer NgBE**
>
> We thank the reviewer for appreciating our theoretical innovations and the potential applicability of the proof technique. We addressed the concern about gradient clipping, as well as your other questions below. Please do not hesitate to reach out in case of further clarifications/questions. Thank you!
>
> **"$\nu$-DP-FTRL doesn't need any gradient clipping … sensitivity might be prohibitively high"**
>
> Great question! The key is that the statement "most of the time, the stochastic gradients won’t exceed the clipping norm" is made for the unclipped gradients. However, we still use the clipped gradients in the algorithm, and so the formal DP guarantee _always holds_ but the utility guarantee is conditional on the unclipped and clipped gradients being the same (i.e. the event $\mathcal{E}$). This is because DP-FTRL always includes gradient clipping as shown in Algorithm 1.
>
> Please see Theorem D.13 in Appendix D for the detailed theorem statement.
>
> **"role of $\nu$ in $\hat \beta_t^\nu$"**
>
> It turns out that ${½ \choose t} \approx t^{-3/2}$ (this can be seen from Stirling’s approximation). However, this is not enough for theory or practice, as we explain below.
>
> Firstly, a correction to the reviewer’s comment. The damping term $(1-\nu)^t \le \exp(-\nu t)$ is an exponential decay, which is asymptotically _faster_ (not slower as the reviewer notes) than the polynomial $t^{-3/2}$ (although the latter is faster for $t$ small).
>
> **Theory**: We need both these terms to get the right rate in theory. Note that $\nu=0$ corresponds to the weights of [Fichtenberger et al. (2023)](https://arxiv.org/pdf/2202.11205.pdf). As shown in Table 2, its sensitivity grows as $\log(T)$ — this is unbounded as $T \to \infty$. Thus, its asymptotic suboptimality is unbounded too. The exponential damping term is sufficient to achieve near-optimality for linear regression.
>
> **Experiments**: We compare to $\nu=0$ experimentally under the name “Optimal CC” (see Table 3 for details). Our experiments show that despite tuning a restart parameter for Optimal CC, it is around 3pp worse than $\nu$-DP-FTRL for all levels of privacy for CIFAR-10.
> Together, both these results show that the $(1-\nu)^t$ damping term is essential both in theory and practice.
>
> Hope we answered all your concerns. We would be happy to address further questions. Thank you for your time!

---

> > ### Author Response · Authors · 2023-11-20
> >
> > Dear Reviewer NgBE,
> >
> > Thank you again for your review! As the discussion period draws to a close, we kindly request that you take a moment to review our responses.
> >
> > Do you have any other questions or concerns we could address to merit a higher score? Thank you for your time and consideration!
> >
> > Best,
> > The Authors

---

> > > ### Comment · Reviewer_NgBE · 2023-11-20
> > > **Response to Authors**
> > >
> > > Thank you to the authors for the detailed response. Particularly, the main concern about the high probability arguments and its connection to differential privacy is resolved. Now I'm convinced by the DP guarantee of $\nu$-DP-FTRL. Personally, I think it's worth mentioning Thm D.13 (especially part part (b)) in the main text for clarity. Also, thank you for clarifying the role of the damping term in the algorithm. I have a better understanding of the algorithm design now.
> > >
> > > Overall, I think this paper has solid results and intriguing techniques. I will increase my score by one step since my main concern is resolved.

---

> > > > ### Author Response · Authors · 2023-11-21
> > > >
> > > > Thank you very much! We will include these updates in the next revision.

---

### Official Review · Reviewer_baaJ · 2023-10-31

**Soundness:** 3 good
**Presentation:** 3 good
**Contribution:** 3 good
**Rating:** 8
**Confidence:** 3

**Summary:**

This paper studies DP-SGD with correlated noise, which is called DP-FTRL. Under the assumptions of i.i.d. data samples, Toeplitz correlation matrix, and unique minimizer, the authors characterized the asymptotic suboptimality of DP-FTRL. This considers mean estimation and linear regression and provides analytical expressions for the asymptotic suboptimality in a function of learning rate and effective dimension. Throughout experiments, the authors show the effectiveness of the proposed methods in various way.

**Strengths:**

The paper is well-written and pleasing to read. The motivation for this investigation is very clear.
The analysis done in the paper is important in private ML, and the approaches to derive the results are interesting. I found that the closed-form expression for the optimal $\beta$ in mean estimation is interesting. It is quite surprising that the analysis of DP-FTRL on the simple mean estimation results in a better trade-off of DP-FTRL in experiments.

**Weaknesses:**

Most of the results are for Noisy-SGD and Noisy-FTRL, which are not DP as written in the paper.
This paper uses lots of assumptions, and it seems difficult to adopt the results in practice.
I understand it is a theory paper, but it would be useful to add more experimental results as they show the practical applicability.

**Questions:**

1) Is it possible to extend the analysis to general correlation matrices $B$ instead of Toeplitz?
2) What is the motivation to study mean estimation? Can Theorem 2.1. be extended to other problems?
3) What is $\mathcal{E}$ inside the expectation on page 7?
4) How to choose $\nu$ in general?

---

> ### Author Response · Authors · 2023-11-17
> **Response to Reviewer baaJ (1/2)**
>
> We thank the reviewer for appreciating the significance of the results. We have addressed the questions about Noisy-FTLR vs. DP-FTRL, and the practical applicability of the theory below. Please do not hesitate to reach out in case of further clarifications/questions. Thank you!
>
> **"Most of the results are for Noisy-SGD and Noisy-FTRL, which are not DP as written in the paper."**
>
> Theorem 2.1 for mean estimation and Theorem 3.1 for general strongly convex functions both hold for DP-FTRL and satisfy full DP guarantees.
>
> We give the main results for linear regression for noisy-FTRL for simplicity. The corresponding finite-time privacy-utility guarantees for DP-FTRL are given in Theorem D.13 of Appendix D, where DP-FTRL has a similar advantage over DP-SGD as Noisy-FTRL over Noisy-SGD (effective dimension dependence instead of dimension, etc.). Further, all our deep learning results compare our $\nu$-DP-FTRL to DP-SGD and prior works in a non-convex setting where all methods satisfy the stated privacy guarantees.
>
> **"This paper uses lots of assumptions, and it seems difficult to adopt the results in practice."**
>
> We respectfully disagree with this assessment. Our theoretical guarantees have two strengths:
> - This is the first rigorous evidence that DP-FTRL is provably better than DP-SGD. This explains empirical observations in prior works that lacked the theoretical backing.
> - Next, our theory is a guiding principle for further algorithmic developments. $\nu$-DP-FTRL, which we derived from our theoretical analysis, is **readily applicable in practice** for private deep learning.
>
> We empirically demonstrate that $\nu$-DP-FTRL outperforms all previous efficient DP-FTRL instantiations (Table 1) and DP-SGD for private image classification and federated language modeling. Furthermore, the simple structure of our correlated noise, which avoids the need to solve an SDP, actually makes it even easier to deploy in practice than the DP-FTRL instantiations proposed by [Denisov et al. (NeurIPS 2022)](https://arxiv.org/pdf/2202.08312.pdf) and [Choquette-Choo et al. (ICML 2023)](https://arxiv.org/abs/2211.06530).
>
> Thus, our theoretical analysis seamlessly leads to practical improvements.
>
> **"...add more experimental results..."**
>
> In Section 4, we compare our method on the benchmark tasks established in the prior literature ([Kairouz et al. ICML 2021](https://arxiv.org/pdf/2103.00039.pdf)), ([Choquette-Choo et al. ICML 2023](https://arxiv.org/pdf/2211.06530.pdf)), ([Choquette-Choo et al. NeurIPS 2023](https://arxiv.org/pdf/2306.08153.pdf), [Koloskova et al. NeurIPS 2023](https://openreview.net/forum?id=qCglMj6A4z)): example-level DP for image-classification on CIFAR-10 and user-level DP for language modeling (next word prediction) on Stack Overflow. Since the submission, we have updated our results on Stack Overflow to include all mechanisms and find that $\nu$-DP-FTRL outperforms all efficient baselines (listed in Table 3) and even matches the multi-epoch upper bound.
>
> If the reviewer has specific experiments they would like to see, we would be happy to add them.
>
> **"Is it possible to extend the analysis to general correlation matrices instead of Toeplitz?"**
>
> Analysis of DP-FTRL with non-Toeplitz matrices is likely to be very technically involved. It is an exciting direction for future work.
>
> From a theoretical perspective, the assumption that $\boldsymbol{B}$ is Toeplitz makes DP-FTRL a time-homogeneous stochastic process (i.e. $P(\theta_{t+1} | \theta_t, z_t=z) = P(\theta_t | \theta_{t-1}, z_{t-1} = z)$). This ensures the existence of a stationary distribution, whose properties we analyze. This is analogous to the classical topic of Markov chains, where most studies of stationarity assume that the system is time-homogeneous.
>
> In the absence of the Toeplitz assumption, $\mathbb{E}[F(\theta_T) - F(\theta_\star)]$ might not even have a limit as $T \to \infty$. We have to find and make appropriate assumptions so that this limit is well-defined. Investigating tree aggregation ([Kairouz et al. ICML 2021](https://arxiv.org/pdf/2103.00039.pdf)) can be a good starting point but the analysis will likely require much stronger technical tools.
>
> From a practical perspective, Toeplitz matrices are more computationally efficient than general correlation matrices. They have been proposed in the previous literature e.g. [Choquette-Choo et al. ICML 2023](https://arxiv.org/abs/2211.06530) for this reason.

---

> > ### Author Response · Authors · 2023-11-17
> > **Response to Reviewer baaJ (2/2)**
> >
> > **"What is the motivation to study mean estimation? Can Theorem 2.1. be extended to other problems?"**
> >
> > Mean estimation is the simplest unconstrained optimization problem with a quadratic objective in 1D. Even in this simple case, we show a non-trivial separation between DP-FTRL and DP-SGD in Theorem 2.1. We give two partial extensions of this result:
> > - Linear regression: Theorem 2.2 shows that $\nu$-DP-FTRL, although not exactly optimal, is near-optimal up to log factors.
> > - General smooth and strongly convex functions: Theorem 3.1 and the accompanying Figure 3 show that $\nu$-DP-FTRL always has a better performance bound than DP-SGD.
> >
> > Our empirical results on deep learning tasks show that the $\nu$-DP-FTRL algorithm is competitive in non-convex deep learning settings as well.
> >
> > **"What is $\mathcal{E}$ inside the expectation on page 7?"**
> >
> > $\mathcal{E}$ denotes the event when the norm of the gradients is under the clip norm $G$ (taken as in Eq. (9)) for all $T$ iterations. On this event, all the clipped gradients coincide exactly with the unclipped ones. As discussed in part (a) at the top of page 7, we have that $\mathbb{P}(\mathcal{E}) \ge 1-p$. The privacy guarantee always holds but the utility bound only holds under the event $\mathcal{E}$ (this approach is standard in the literature, see e.g. [Varshney et al. COLT 2022](https://arxiv.org/pdf/2207.04686.pdf)). We refer to Theorem D.13 in Appendix D for a full mathematical statement.
> >
> > **"How to choose $\nu$ in general?"**
> >
> > We treat the parameter $\nu$ as a hyperparameter which we tune together with the optimizer hyperparameters (e.g. learning rate). In our experimental comparisons, we tune a “restart” parameter for all the baselines (denoted as $\times S$ in the legend), so the overall hyperparameter tuning cost of $\nu$-DP-FTRL is the same as that of the baselines. With the tuning costs equalized, we still find that $\nu$-DP-FTRL outperforms all other efficient baselines by a large margin.

---

> > > ### Comment · Reviewer_baaJ · 2023-11-19
> > >
> > > Thank you for the detailed response to my questions. I have a better understanding of your work. One further question that came to me is about the explanation of the benefits of correlated noise. For me, it seems from figure 1 that the optimal coefficient sequence having negative values except for the first coefficient might compensate for the noise added to the gradient in the previous steps. Is there any intuitive explanation for why correlated noise gives better results than uncorrelated noise from the theoretical results?

---

> > > > ### Author Response · Authors · 2023-11-19
> > > > **Response to Reviewer baaJ**
> > > >
> > > > That’s a great observation! The (anti-)correlated noise indeed helps by "canceling out" the previously added noise. Thus, all but the first component of $\boldsymbol\beta_\star$ are negative.
> > > >
> > > > For learning problems, the gradients move the iterates toward the minimizer and "undo" the effect of the previously added noise. However, this effect is significant only for directions with a _large signal_ (e.g. for linear regression, this would be the directions with large eigenvalues of the Hessian). For _low signal_ directions, the (anti-)correlated noise is essential to preventing the past noise from accumulating.
> > > >
> > > > Have we answered all of your questions? Do you have any other concerns we could address to merit a higher score? Thank you.

---

> > > > > ### Comment · Reviewer_baaJ · 2023-11-21
> > > > >
> > > > > Thank you for the response. The explanation about the role of correlated noise is intuitive and helps me better understand the $\nu$-DP-FTRL. I think adding some discussion (e.g., the above example) of the theoretical results strengthens the paper.
> > > > > As the authors have answered my questions, I will increase my score accordingly.

---

> > > > > > ### Author Response · Authors · 2023-11-21
> > > > > >
> > > > > > Thank you very much! We will add this explanation to the next revision.

---

### Meta-Review · Area_Chair_9BRy · 2023-12-18

**Metareview:**

The paper studies the performance of DP-FTRL, the differentially private version of the follow the regularized leader algorithm for online learning and compares it with the vanilla DP-SGD, which adds independent Gaussian noise to achieve differential privacy. The main contribution is a new variant of DP-FTRL using a specific Toeplitz noise matrix along with the analysis of this algorithm for Gaussian inputs and the objective functions being linear regression and strongly convex functions. In the first setting, the paper proves theoretical bounds showing DP-FTRL improving over vanilla DP-SGD, though it should be noted that similar guarantees are known for more sophisticated versions of DP-SGD. For experiments, the paper shows that the new variant of DP-FTRL has much faster runtime than previous variants while achieving improved accuracy for high epsilon (high amount of privacy loss). All three reviewers appreciate the contribution of the paper but they have different opinions on the writing quality.

In terms of strength, the paper gives a simpler and more practical version of DP-FTRL with faster runtime (O(T) per iteration instead of large poly(T) for other DP-FTRL variants) while keeping the advantage over vanilla DP-SGD in the high epsilon regime. There are both theoretical contributions in the linear regression case as well as experimental validation on standard benchmarks.

In terms of weakness, the writing quality leaves much to be desired, especially when it manages to confuse some reviewer, who works in the area. While this is concerning, I lean toward accepting the paper due to its timely contribution to a very active topic. The paper contains techniques from several areas including differential privacy (zCDP), Fourier analysis, LTI systems, etc which is a plus to me since it introduces potentially new techniques to many in the audience. However, to do so, I urge the authors to improve the writing to remove typos, spell out the arguments similar to the updated proof for theorem 2.1,

**Justification For Why Not Higher Score:**

The writing quality seems low as it manages to confuse a diligent reviewer working in the area. A key theorem originally has an extremely terse proof and when spelled out in the discussion with the reviewer, the proof becomes significantly longer. The reviewer states that they see similarly many issues with the writing in the rest of the paper, which is over 50 pages.

**Justification For Why Not Lower Score:**

Two reviewers strongly support the paper. The topic has seen significant interest with a series of papers in the recent conferences. The results in the paper seem to bring both new theoretical understanding and practical improvements. That said, since the paper is already available on openreview for interested audience, I am also fine with the view of the third reviewer, who recommended reject so that the authors can revise and submit a better version to a later conference.

---

### Decision · Program_Chairs · 2024-01-16

Accept (poster)